# HalluGuard: Demystifying Data-Driven and Reasoning-Driven Hallucinations in LLMs

**Xinyue Zeng**[*]
CS Department
Virginia Tech

**Junhong Lin**[*]
EECS Department
MIT

**Yujun Yan**
CS Department
Dartmouth College

**Feng Guo**
Statistics Department
Virginia Tech

**Liang Shi**
Statistics Department
Virginia Tech

**Jun Wu**
CS Department
Michigan State University

**Dawei Zhou**
CS Department
Virginia Tech

## Abstract

The reliability of Large Language Models (LLMs) in high-stakes domains such as healthcare, law, and scientific discovery is often compromised by hallucinations. These failures typically stem from two sources: *data-driven hallucinations* and *reasoning-driven hallucinations*. However, existing detection methods usually address only one source and rely on task-specific heuristics, limiting their generalization to complex scenarios. To overcome these limitations, we introduce the *Hallucination Risk Bound*, a unified theoretical framework that formally decomposes hallucination risk into data-driven and reasoning-driven components, linked respectively to training-time mismatches and inference-time instabilities. This provides a principled foundation for analyzing how hallucinations emerge and evolve. Building on this foundation, we introduce HalluGuard, a NTK-based score that leverages the induced geometry and captured representations of the NTK to jointly identify data-driven and reasoning-driven hallucinations. We evaluate HalluGuard on 10 diverse benchmarks, 11 competitive baselines, and 9 popular LLM backbones, consistently achieving state-of-the-art performance in detecting diverse forms of LLM hallucinations. We open-source our proposed HalluGuard model at HalluGuard.

## 1 Introduction

Large language models (LLMs) are increasingly deployed in high-stakes domains such as healthcare, law, and scientific discovery (Bommasani et al., 2021; Thirunavukarasu et al., 2023; Ke et al., 2025). However, adoption in these settings remains cautious, as such domains are highly regulated and demand strict compliance, interpretability, and safety guarantees (Dennstädt et al., 2025; Kattnig et al., 2024). A major barrier is the risk of *hallucinations*, generated content appears unfaithful or nonsensical. Such errors can have severe consequences (Dennstädt et al., 2025), as the example in Figure 1, a generated incorrect medical diagnosis may delay treatment or lead to harmful interventions. Therefore, detecting hallucinations is not merely a technical challenge but a prerequisite for trustworthy deployment, as undetected errors undermine reliability, accountability, and user safety.

Generally, hallucinations in LLMs arise from two primary sources (Ji et al., 2023; Huang et al., 2025): *data-driven hallucinations*, which stem from flawed, biased, or incomplete knowledge encoded during pre-training or fine-tuning; and *reasoning-driven hallucinations*, which originate from inference-time failures such as logical inconsistencies or breakdowns in multi-step reasoning (Zhang et al., 2023; Zhong et al., 2024). Detection methods broadly split along these two dimensions. Approaches for data-driven hallucinations often compare outputs against retrieved documents or references (Shuster et al., 2021; Min et al., 2023; Ji et al., 2023), or exploit sampling consistency as in SelfCheckGPT (Manakul et al., 2023). In contrast, methods for reasoning-driven hallucinations rely on signals of inference-time instability, including probabilistic measures such as perplexity (Ren

---

[*]Equal contribution.

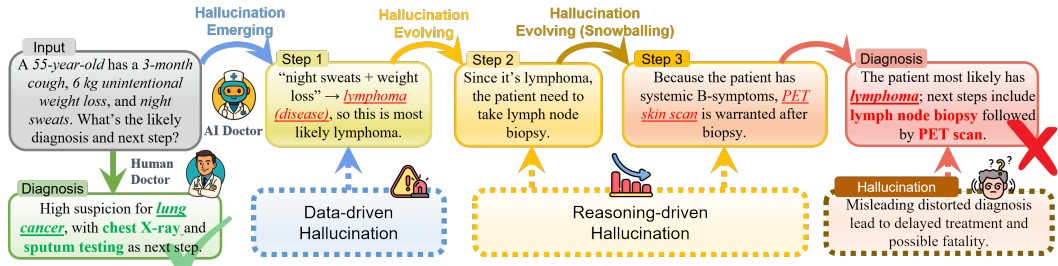

Figure 1: An illustration of hallucination emerging and evolving in the context of disease diagnosis.

et al., 2023), length-normalized entropy (Malinin & Gales, 2021), semantic entropy (Kuhn et al., 2023), energy-based scoring (Liu et al., 2020), and RACE (Wang et al., 2025). Others probe internal representations, for example, Inside (Chen et al., 2024), which applies eigenvalue-based covariance metrics and feature clipping, ICR Probe (Zhang et al., 2025), which tracks residual-stream updates, and Shadows in the Attention (Wei et al., 2025), which analyzes representation drift under contextual perturbations. While these methods shed light on the mechanisms underlying hallucinations, most remain tailored to a single hallucination type and fail to capture their evolution. Yet growing evidence indicates that data-driven and reasoning-driven hallucinations often evolve during multi-step generation (Liu et al., 2025; Sun et al., 2025). As shown in Figure 1, it emerges from an initial disease misclassification and evolves into a distorted diagnosis, delaying treatments and risking fatality. This gap brings two central questions: *(1) How can we develop a unified theoretical understanding of how hallucinations evolve? (2) How can we detect them effectively and efficiently without relying on external references or task-specific heuristics?*

To address these challenges, we propose a unified theoretical framework–*Hallucination Risk Bound*, which decomposes the overall hallucination risk into two components: a *data-driven term*, capturing semantic deviations rooted in inaccurate, imbalanced, or noisy supervision acquired during model training; and a *reasoning-driven term*, reflecting instability introduced by inference-time dynamics, such as logical missteps or temporal inconsistency. This decomposition not only elucidates the mechanism behind hallucinations but also reveals how they emerge and evolve. Specifically, our analysis shows that hallucinations originate from semantic approximation gaps, captured by representational limits of the model, and are subsequently amplified by unstable rollout dynamics, evolving across decoding steps. As such, our framework offers a unified theoretical lens for characterizing the emergence and evolution of these hallucinations.

Building on the theoretical foundation, we propose HALLUGUARD, a Neural Tangent Kernel(NTK)-based score that leverages the induced geometry and captured representations of the NTK to jointly identify data-driven and reasoning-driven hallucinations. We evaluate HALLUGUARD comprehensively across 10 diverse benchmarks, 11 competitive baselines, and 9 popular LLM backbones. HALLUGUARD consistently achieves state-of-the-art hallucination detection performance, demonstrating its efficacy. We open-source our proposed HALLUGUARD model at HalluGuard.

## 2 PRELIMINARIES

**Hallucination Detection.** There are two primary sources of hallucinations in LLMs (Ji et al., 2023; Huang et al., 2025): *data-driven hallucination*, which stems from incomplete or biased knowledge encoded during pre-training or fine-tuning, and *reasoning-driven hallucination*, which arises from unstable or inconsistent inference dynamics at decoding time. This distinction has implicitly guided a broad range of detection strategies, which we examine through these two lenses.

For data-driven causes, a recurring signal is elevated predictive uncertainty. A common formulation adopts the sequence-level negative log-likelihood:

$$\mathcal{U}(\mathbf{y} \mid \mathbf{x}, \theta) = -\frac{1}{T} \sum_{t=1}^{T} \log p_\theta(y_t \mid y_{<t}, \mathbf{x}), \tag{1}$$

which quantifies the average uncertainty of generating a sequence $\mathbf{y} = [y_1, \ldots, y_T]$ from input $\mathbf{x}$ and $\theta$ denotes model parameters. This directly recovers *Perplexity* (Ren et al., 2023), where low scores

imply confident predictions, while high scores indicate implausible generations due to weak priors. To capture more nuanced uncertainty, later methods extend this formulation to multi-sample settings. The *Length-Normalized Entropy* (Malinin & Gales, 2021) penalizes dispersion across stochastic generations $\mathcal{Y} = \{\mathbf{y}^1, \ldots, \mathbf{y}^K\}$ where $K$ denotes the number of independent stochastic rollouts sampled from the model for a given input, offering a finer-grained view of model indecision. This perspective is further enriched by *Semantic Entropy* (Kuhn et al., 2023), which projects sampled responses into semantic space, and by energy-based scoring (Liu et al., 2020), which replaces log-probability with a learned confidence function. Collectively, these methods reflect a progression from token-level likelihoods to semantically grounded multi-sample uncertainty estimators.

In contrast, reasoning-driven hallucinations arise from brittle inference trajectories, where identical contexts may yield inconsistent or incoherent outputs. A commonly used measure of such instability is the cross-sample consistency score:

$$\mathcal{C}(\mathcal{Y} \mid \mathbf{x}, \theta) = \frac{1}{C} \sum_{i=1}^{K} \sum_{j=i+1}^{K} \mathrm{sim}(\mathbf{y}^i, \mathbf{y}^j), \tag{2}$$

where $C = K \cdot \frac{(K-1)}{2}$, and $\mathrm{sim}(\cdot, \cdot)$ is a similarity function such as ROUGE-L (Lin, 2004), cosine similarity, or BLEU (Chen et al., 2023). Low scores reflect diverging generations and unstable reasoning. Several reasoning-driven detection methods can be interpreted through this lens. Early approaches used surface-level lexical overlap metrics (Lin et al., 2022b), while *SelfCheck-GPT* (Manakul et al., 2023) advanced this by evaluating factual entailment across responses, and *FActScore* (Min et al., 2023) extended this further by comparing outputs to retrieved reference documents. More recent efforts probe internal signals directly: *Inside* (Chen et al., 2024) analyzes the covariance spectrum of embedding representations, and *RACE* (Wang et al., 2025) diagnoses instability in multi-step reasoning.

**NTK in LLMs.** NTK provides a principled framework for analyzing the training dynamics in the overparameterized regime characteristic of modern LLMs (Jacot et al., 2018). Formally, for a network output $f(x, \theta)$ with input $x$ and parameters $\theta$, the NTK is defined as:

$$\Theta(x, x', \theta) = \nabla_\theta f(x, \theta) \cdot \nabla_\theta f(x', \theta). \tag{3}$$

This kernel $\Theta(x, x', \theta)$ quantifies the similarity of training dynamics between inputs $x$ and $x'$. In the infinite-width limit, it converges to a deterministic value at initialization and remains nearly constant throughout training (Lee et al., 2020b). This stability reduces the highly nonlinear optimization of deep networks to a tractable kernel regression problem. By examining the eigenspectrum of the NTK, one can probe how internal representations are shaped during training: which features are prioritized (e.g., syntax versus semantics), how quickly different tasks converge, and why overparameterized networks generalize effectively to unseen data (Ju et al., 2022). In this way, the NTK transforms the apparent complexity of LLM optimization into a clear lens on how these models capture, process, and generalize information (Zeng et al., 2025).

## 3 METHODOLOGY

### 3.1 PROBLEM SETTING

Our analysis reveals that hallucination is not a unified failure mode but rather shifts with the task structure. On the instruction-following `Natural` benchmark (Wang et al., 2022), 88.9% of the overall 3499 errors are from logical missteps (*reasoning-driven*) while 11.1% are factual inaccuracies (*data-driven*). By contrast, on the math-focused `MATH-500` (Hendrycks et al., 2021), the 1985 wrong generations are dominated by 1946 reasoning errors (98.1%), with only 19 factual flaws (1.9%). This contrast highlights that, in practice, hallucinations are rarely pure but often mixtures of data-driven bias and reasoning-driven instability-motivating our formal decomposition of hallucination sources.

**Problem Definition.**

Let $\mathcal{Y}$ denote the discrete space of all possible finite-length textual token sequences. We define a continuous semantic embedding space $U_h \subseteq \mathbb{R}^{d_h}$ equipped with a norm $\|\cdot\|$. Each vector $u \in U_h$ represents the semantic representation of a reasoning chain composed of step-wise logical

statements. We define a task-specific encoder $\Phi : \mathcal{Y} \to U_h$ that maps a discrete textual sequence into this continuous hypothesis space. In this framework, for an input $\mathbf{x}$ with a ground-truth output sequence $y^* \in \mathcal{Y}$, we define the target semantic representation as $u^* := \Phi(y^*) \in U_h$. An LLM with parameters $\theta$ emits a random sequence $Y = (Y_1, \ldots, Y_T) \in \mathcal{Y}$ via the autoregressive decoding distribution $p_\theta(y_t \mid y_{<t}, \mathbf{x})$, yielding a predicted semantic representation $u_h := \Phi(Y) \in U_h$. Thus, the model's expected semantic output is defined as $\mathbb{E}[u_h] := \mathbb{E}_{Y \sim p_\theta(\cdot|\mathbf{x})}[\Phi(Y)]$.

To analyze inference dynamics, we consider perturbations in a local neighborhood of the decoding process. Let $\mathbb{R}^r$ denote the $r$-dimensional continuous space of the model's internal states (e.g., prefix embeddings or hidden activations). We parameterize a small perturbation by $\delta \in \mathbb{R}^r$, restricted to a local $\ell_2$-ball $\mathcal{B}_\rho := \{\delta \in \mathbb{R}^r : \|\delta\|_2 \leq \rho\}$. Let $P_\theta(\cdot \mid \mathbf{x}, \delta)$ denote the perturbed decoding distribution induced by $\delta$. We define the mean semantic response map $G_Y : \mathbb{R}^r \to U_h, G_Y(\delta) := \mathbb{E}_{Y \sim P_\theta(\cdot|\mathbf{x},\delta)}[\Phi(Y)]$ with its corresponding inference Jacobian $J := DG_Y(0) \in \mathbb{R}^{d_h \times r}$. Thus, we formally define the problem as follows:

**Problem 1** (Hallucination Dynamics Characterization)**.**
*Given: (1) The target semantic representation $u^* := \Phi(y^*) \in U_h$ for a ground-truth output $y^* \in \mathcal{Y}$; (2) the random sequence $Y \in \mathcal{Y}$ emitted via the autoregressive decoding distribution $p_\theta(y_t \mid y_{<t}, \mathbf{x})$, yielding a predicted representation $u_h := \Phi(Y)$ with expected value $\mathbb{E}[u_h]$ ; and (3) the inference constraints defined by a local perturbation $\delta \in \mathbb{R}^r$ restricted to the $\ell_2$-ball $\mathcal{B}_\rho$*

*Find: A formal geometric mechanism to characterize how hallucinations emerge and evolve by analyzing the Mean Semantic Response Map $G_Y(\delta)$ and the Inference Jacobian $J$, which captures the sensitivity of the model's reasoning trajectory to internal instabilities.*

## 3.2 HALLUCINATION RISK BOUND

To bridge the formal setup with the phenomenon of hallucination, we first disentangle the sources of hallucinations. Intuitively, hallucinations may arise either from systematic biases in the knowledge encoded by the model (data-driven) or from instabilities during autoregressive decoding (reasoning-driven). The following proposition formalizes this idea by decomposing the total hallucination risk into two components.

We first impose the following assumptions:

**A1.** $(U_h, \|\cdot\|)$ is a finite-dimensional Hilbert space. The encoder $\Phi : \mathcal{Y} \to U_h$ is measurable, and the random variable $\Phi(Y)$ has finite second moment under the model's unperturbed decoding distribution: $\mathbb{E}_{Y \sim p_\theta(\cdot|\mathbf{x})}[\|\Phi(Y)\|^2] < \infty$. This ensures that the mean semantic representation $\mathbb{E}[\Phi(Y)]$ is well-defined in $U_h$.

**A2.** Let $(\mathcal{Y}, d_\mathcal{Y})$ be the discrete metric space equipped with edit distance. The encoder $\Phi : (\mathcal{Y}, d_\mathcal{Y}) \to (U_h, \|\cdot\|)$ is $L_\Phi$-Lipschitz continuous: $\|\Phi(y) - \Phi(y')\| \leq L_\Phi \, d_\mathcal{Y}(y, y') \quad \forall y, y' \in \mathcal{Y}$.

**A3.** For any perturbation $\delta$ in the closed ball $\mathcal{B}_\rho := \{\delta \in \mathbb{R}^r : \|\delta\|_2 \leq \rho\}$, the mean semantic response map $G_Y(\delta) = \mathbb{E}_{Y \sim P_\theta(\cdot|\mathbf{x},\delta)}[\Phi(Y)]$ is twice Fréchet differentiable in a neighborhood of $\delta = 0$ and admits the expansion $G_Y(\delta) = G_Y(0) + J\delta + R(\delta)$, where $J = DG_Y(0) \in \mathbb{R}^{d_h \times r}$ and the remainder satisfies $\|R(\delta)\| \leq \frac{1}{2}H_\star\|\delta\|_2^2, \forall \delta \in \mathcal{B}_\rho$, for some constant $H_\star > 0$.

**Proposition 3.1 (Hallucination Risk Decomposition).** Under A1-A3, applying the triangle inequality yields a natural split of the risk:

$$\|u^* - u_h\| \leq \underbrace{\|u^* - \mathbb{E}[u_h]\|}_{\text{data-driven term}} + \underbrace{\|u_h - \mathbb{E}[u_h]\|}_{\text{reasoning-driven term}}$$

This decomposition distinguishes errors caused by systematic bias in the learned representation from those introduced during stochastic rollout.

**Characterizing Data-Driven Hallucination.** To quantify the data-driven term, we take inspiration from the NTK, which has proven effective in analyzing training dynamics of overparameterized models. Here, NTK geometry provides a way to measure how well the model's representation space aligns with task generation under small perturbations.

Let $U_h$ denote the hypothesis subspace accessible to the model under perturbations. By Céa's lemma(Céa, 1964) with curvature penalty, the data-driven term can be bounded as

$$\|u^* - \mathbb{E}[u_h]\| \leq \frac{\Lambda}{\gamma} \inf_{u \in U_h} \|u^* - u\|, \tag{4}$$

where $\gamma = \lambda_{\min}(\mathcal{K}_\Phi)$ is the smallest eigenvalue of the NTK Gram matrix on embedded perturbations $\mathcal{K}_\Phi$, and $\Lambda \leq \|\mathcal{T}\|$, where $\mathcal{T} : U_h \to U_h$ denotes the operator mapping. Intuitively, the ratio $\frac{\Lambda}{\gamma}$ measures the conditioning of the feature map: well-conditioned NTK spectra allow a closer approximation to the true generation.

Thus, the ratio can be further controlled in terms of pretraining-finetuning mismatch:

$$\frac{\Lambda}{\gamma} \leq 1 + k_{\mathrm{pt}} \log \mathcal{O}(P, L) + k \cdot \frac{\epsilon_{\mathrm{mismatch}}}{\mathrm{Signal}_k}, \tag{5}$$

where $\log \mathcal{O}(P, L)$ is a complexity term from parameter count $P$ and prompt length $L$, $\epsilon_{\mathrm{mismatch}}$ denotes the Wasserstein distance between prompt and query distributions, $\mathrm{Signal}_k$ measures task-aligned energy in the top-$k$ eigenspace. $k_{\mathrm{pt}}$ and $k$ are task and model-dependent constants. Thus, data-driven hallucinations grow when the mismatch is large or when the task signal is weak.

**Characterizing Reasoning-Driven Hallucination.** The reasoning-driven term captures *reasoning-driven* instability that accumulates during autoregressive decoding. Here, we model generation as a martingale process, where deviation from the expectation is controlled by concentration inequalities. Specifically, Freedman's inequality (Geman et al., 1992) gives

$$\|u_h - \mathbb{E}[u_h]\| \leq K \cdot \exp\left(-\frac{K\epsilon^2}{C}\right) \cdot \alpha(e^{\beta T} - 1), \tag{6}$$

where $K$ is the number of rollouts averaged, $\beta$ summarizes per-step growth in local Jacobians, $\alpha$ scales the cumulative effect and $C$ is a task and model-dependent constant. This bound shows that reasoning-driven hallucinations grow exponentially with sequence length $T$.

We now synthesize the two components into a unified result that characterizes the overall risk of hallucination. By combining the NTK-conditioned approximation bound for data-driven deviation with the Freedman-style concentration bound for reasoning-driven instability, we obtain the following unified bound of data-driven and reasoning-driven hallucinations (detailed proof is provided in Appendix A):

---

**Theorem 3.2 (Hallucination Risk Bound).** Let $u^* := \Phi(y^*)$ denote the semantic embedding of the ground-truth output and $u_h := \Phi(Y)$ that of the model-generated output. Under Assumptions A1-A3, suppose there exists $\beta \geq 0$ such that $\left\|\prod_{t=1}^T J_t\right\|_2 \leq e^{\beta T}$. Then the total hallucination risk satisfies

$$\|u^* - u_h\| \leq \underbrace{\left(1 + k_{\mathrm{pt}} \log \mathcal{O}(P, L) + k \cdot \frac{\epsilon_{\mathrm{mismatch}}}{\mathrm{Signal}_k}\right) \inf_{u \in U_h} \|u^* - u\|}_{\text{data-driven term}} + \underbrace{|\mathcal{L}| \cdot \exp\left(-\frac{K\epsilon^2}{C}\right) \cdot \alpha(e^{\beta T} - 1)}_{\text{reasoning-driven term}}$$

Here, $|\mathcal{L}|$ denotes the total sampled trajectories.

---

### 3.3 Hallucination Quantification via HalluGuard

While Theorem 3.2 makes explicit how data-driven and reasoning-driven hallucinations emerge and evolve, applying it directly at inference is impractical since direct step-wise Jacobians for billion-parameter LLMs are intractable, so we seek a *proxy score* that is computable, stable, and faithful to our decomposition.

Let $\mathcal{K}$ denote the NTK Gram matrix with eigenvalues $\lambda_1 \geq \cdots \geq \lambda_r > 0$ and condition number $\kappa(\mathcal{K}) = \lambda_{\max}/\lambda_{\min}$. Let $J_t$ be the step-$t$ input-output Jacobian of the decoder, and define $\sigma_{\max} := \sup_t \|J_t\|_2$ as the uniform spectral bound(note that $\sigma_{\max}$ is independent of the spectrum of $\mathcal{K}$).

Under Assumptions A1-A3, a standard NTK approximation argument yields $\inf_{u \in U_h} \|u^* - u\| \leq C_d \det(\mathcal{K})^{-c_d} \|u^*\|$, so that $\det(\mathcal{K})$ capture the representations in systematic bias.

For autoregressive rollout, based on the property of Jacobian, we have $\left\| \prod_{t=1}^{T} J_t \right\|_2 \leq \prod_{t=1}^{T} \|J_t\|_2 = \exp\left( \sum_{t=1}^{T} \log \|J_t\|_2 \right)$, so that we have $\left\| \prod_{t=1}^{T} J_t \right\|_2 \leq e^{\beta T}$. Since $\beta \leq \log \sigma_{\max}$ with $\sigma_{\max} := \sup_t \|J_t\|_2$ thus we have the upper bound as $\| \prod_{t=1}^{T} J_t \|_2 \leq \sigma_{\max}^T = e^{(\log \sigma_{\max})T}$. Thus, $\log \sigma_{\max}$ serves as a stable and tractable proxy for the per-step amplification rate.

Perturbation analysis of $\mathcal{K}$, together with classical eigenvalue sensitivity results (Trefethen & Bau, 2022), yields $\mathrm{Var}[u_h] \leq c_v \kappa(\mathcal{K})^2 \|\delta\|^2$, showing that instability grows quadratically with the condition number $\kappa(\mathcal{K})$. To temper this effect and ensure additivity, we penalize ill-conditioned representations via $-\log \kappa^2$, where log compression brings a well-behaved dynamic range.

In summary, $\det(\mathcal{K})$ quantifies representational adequacy, $\log \sigma_{\max}$ captures rollout amplification, and $-\log \kappa^2$ penalizes spectral instability, together forming a compact and tractable proxy consistent with the Hallucination Risk Bound. The lightweight projection layers are self-supervised spectral calibration modules, optimized offline (via AdamW) to align NTK spectral properties across heterogeneous backbones into a stable, comparable geometric space-without hallucination labels or task-specific supervision, with the backbone fully frozen and zero runtime overhead during inference. Detailed proofs are provided in Appendix B.

Table 1: Correlation between NTK proxies and task families.

|  | SQuAD | Math-500 | TruthfulQA |
|---|---|---|---|
| $\det(\mathcal{K})$ | 0.84 | 0.42 | 0.61 |
| $\log \sigma_{\max} - \log \kappa^2$ | 0.39 | 0.88 | 0.67 |

**Empirical validation.** We empirically validate how those proxies correlate with different task families. In Table 1, $\det(\mathcal{K})$ correlates most strongly with the data-centric task SQuAD (0.84), indicating its role in capturing factual fidelity. In contrast, for the reasoning-oriented MATH-500, the highest correlation is observed with $\log \sigma_{\max} - \log \kappa^2$ (0.88), reflecting the importance of amplification and stability in multi-step reasoning.

Motivated by the above, we formally define HALLUGUARD as follows, which provides a principled and unified lens for hallucination detection:

$$\mathbf{HALLUGUARD}(u_h) = \det(\mathcal{K}) + \log \sigma_{\max} - \log \kappa^2. \tag{7}$$

## 4 EXPERIMENTS

We comprehensively evaluate HALLUGUARD across 10 diverse benchmarks, 11 competitive baselines, and 9 popular LLM backbones. We aim to evaluate its efficacy from the following five questions: *Q1: How does HALLUGUARD perform across different task families? Q2: How does HALLUGUARD perform across LLMs of different scales? Q3: How does each term capture trends across task families? Q4: Can HALLUGUARD guide test-time inference to improve downstream reasoning? Q5: How well does HALLUGUARD generalize to detecting fine-grained hallucinations beyond benchmarks?*

Section 4.1 details the setup; Section 4.2 evaluates HALLUGUARD as a detection method(Q1–Q3), Section 4.3 applies HALLUGUARD in score-guided inference(Q4) and Section 4.4 analyzes HALLUGUARD on fine-grained hallucination via a case study on semantic data(Q5).

### 4.1 EVALUATION SETUP

**Benchmarks.** We evaluate across 10 widely used benchmarks spanning three distinct categories. For data-grounded QA, we include RAGTruth (Niu et al., 2024), NQ-Open (Kwiatkowski et al., 2019), HotpotQA (Yang et al., 2018) and SQuAD (Rajpurkar et al., 2016), which emphasize factual correctness through external evidence. For reasoning-oriented tasks, we use GSM8K (Cobbe et al., 2021), MATH-500 (Hendrycks et al., 2021), and BBH (Suzgun et al., 2023), which require multi-step derivations prone to compounding errors. Finally, for instruction-following settings, we consider TruthfulQA (Lin et al., 2022a), HaluEval (Li et al., 2023a) and Natural (Wang et al., 2022), which probe hallucinations under open-ended or adversarial prompts.

**Baselines.** We compare HALLUGUARD with 11 competitive detectors spanning diverse strategies. Uncertainty-based methods include Perplexity (Ren et al., 2023), Length-Normalized Predictive Entropy(LN-Entropy) (Malinin & Gales, 2021), Semantic Entropy (Kuhn et al., 2023), Energy Score (Liu et al., 2020) and P(true) (Kadavath et al., 2022). Consistency-based approaches cover SelfCheckGPT (Manakul et al., 2023), Lexical Similarity (Lin et al., 2022b), FActScore (Min et al., 2023) and RACE (Wang et al., 2025). Internal-state methods are represented by Inside (Chen et al., 2024) and MIND (Su et al., 2024).

**LLM Backbone Models.** We evaluate 9 publicly available LLMs spanning different scales and architectures. These include five models from the Llama family (Llama2-7B, Llama2-13B, Llama2-70B, Llama3-8B, and Llama3.2-3B) (Touvron et al., 2023; Grattafiori et al., 2024), along with OPT-6.7B (Zhang et al., 2022), Mistral-7B-Instruct (Jiang et al., 2023), QwQ-32B (Yang et al., 2024), and GPT-2 (117M) (Radford et al., 2019). All models are used in their off-the-shelf form with pre-trained weights and tokenizers provided by Hugging Face, without further fine-tuning.

**Evaluation Metrics.** We evaluate hallucination detection ability under two regimes following Janiak et al. (2025): ROUGE-based reference evaluation ($*_r$) and LLM-AS-A-JUDGE ($*_{llm}$). For performance measures, we report the area under the receiver operating characteristic curve (AUROC) and the area under the precision-recall curve (AUPRC). AUROC is widely used to assess the quality of binary classifiers and uncertainty estimators, while AUPRC highlights performance under class imbalance. In both cases, higher values indicate better detection.

## 4.2 MAIN RESULTS

**Q1: How does HALLUGUARD perform across different task families?** To evaluate how HALLUGUARD performs across different task types, we conduct experiments on all benchmarks. For clarity, Table 2 presents representative results from three task families: data-centric (`RAGTruth`), reasoning-oriented (`Math-500`), and instruction-following (`TruthfulQA`). As shown, HALLUGUARD consistently outperforms all baselines across backbones. On `Math-500`, it reaches 81.76% AUROC and 79.76% AUPRC, improving over the second-best method by up to 8.3%. On `RAGTruth`, it attains 84.59% AUROC and 81.15% AUPRC, with gains of up to 7.7%. On `TruthfulQA`, it achieves 77.05% AUROC and 73.79% AUPRC, exceeding the next strongest baseline by as much as 6.2%. Overall, HALLUGUARD establishes new state-of-the-art results across diverse task families, with particularly pronounced improvements on reasoning-oriented benchmarks.

**Q2: How does HALLUGUARD perform across LLMs of different scales?** We further investigate whether the effectiveness of HALLUGUARD depends on model scale, as smaller backbones are typically more prone to hallucination. Table 3 reports representative results on small(Llama2-7B, Llama3-8B), mid-sized(Llama2-13B), and large-scale(Llama2-70B) models using `SQuAD`, `GSM8K`, and `HaluEval`. Across all settings, HALLUGUARD consistently surpasses baselines, with the largest margins on smaller models-for instance, 72.89% AUPRC$_r$ on `HaluEval` with Llama2-7B, more than 10% above the second best. Mid-sized models also exhibit clear gains (e.g., 79.01% AUROC$_r$ on `GSM8K`), while even large-scale models like Llama2-70B see steady improvements (e.g., 83.8% AUROC$_r$ on `SQuAD`). Overall, HALLUGUARD benefits most on small backbones while maintaining consistent advantages across scales.

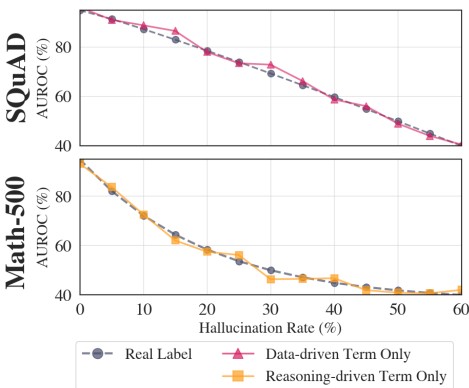

**Q3: How does each term capture trends across task families?** As shown in Figure 2, each term faithfully tracks the ground-truth trend within its respective task family. On data-centric `SQuAD`, the *data-driven term* closely follows the dashed gold curve across the variant hallucination rate, capturing the smooth AUROC decline. On reasoning-oriented `MATH-500`, the *reasoning-driven term* mirrors the monotonic AUROC drop as reasoning drift in-

Figure 2: Ablation results comparing individual terms with ground-truth trends on `SQuAD` (top) and `Math-500` (bottom).

Table 2: Performance comparison on representative benchmarks: data-centric (`RAGTruth`), reasoning-oriented (`Math-500`), and instruction-following (`TruthfulQA`). We highlight the **first** and second best results.

| | | GPT2 | | | | OPT-6.7B | | | | Mistral-7B | | | | QwQ-32B | | |
|---|---|---|---|---|---|---|---|---|---|---|---|---|---|---|---|---|
| | | $AUROC_r$ | $AUPRC_r$ | $AUROC_{llm}$ | $AUPRC_{llm}$ | $AUROC_r$ | $AUPRC_r$ | $AUROC_{llm}$ | $AUPRC_{llm}$ | $AUROC_r$ | $AUPRC_r$ | $AUROC_{llm}$ | $AUPRC_{llm}$ | $AUROC_r$ | $AUPRC_r$ | $AUROC_{llm}$ | $AUPRC_{llm}$ |
| **RAGTruth** | HALLUGUARD | **75.51** | **73.40** | **62.40** | **56.60** | **80.13** | **76.77** | **71.01** | **63.58** | **82.31** | **80.79** | **64.89** | **67.25** | **84.59** | **81.15** | **71.82** | **66.68** |
| | Inside | 73.42 | 73.08 | 61.99 | 56.39 | 79.49 | 71.82 | 66.1 | 62.46 | 75.32 | 73.19 | 64.58 | 61.05 | 77.72 | 73.47 | 66.05 | 64.73 |
| | MIND | 58.54 | 54.79 | 43.47 | 41.85 | 63.82 | 62.58 | 51.03 | 44.78 | 73.13 | 71.53 | 58.25 | 58.6 | 64.23 | 63.06 | 47.37 | 51.47 |
| | Perplexity | 58.07 | 56.68 | 43.84 | 41.53 | 64.47 | 61.57 | 47.12 | 52.98 | 65.42 | 63.63 | 53.28 | 51.36 | 73.91 | 72.92 | 60.81 | 59.77 |
| | LN-Entropy | 64.42 | 60.79 | 49.41 | 45.04 | 60.81 | 57.91 | 48.76 | 42.27 | 64.22 | 60.92 | 52.24 | 48.41 | 63.81 | 62.26 | 47.52 | 52.17 |
| | Energy | 65.53 | 62.42 | 51.8 | 47.22 | 66.54 | 63.28 | 54.21 | 49.19 | 64.36 | 62.26 | 48.64 | 53.93 | 73.26 | 71.21 | 65.43 | 62.32 |
| | Semantic Ent. | 60.72 | 59.41 | 50.55 | 45.86 | 70.2 | 68.34 | 54.54 | 56.74 | 66.01 | 64.49 | 53.01 | 55.5 | 66.48 | 64.41 | 51.54 | 50.11 |
| | Lexical Sim. | 64.72 | 63.1 | 55.04 | 48.04 | 67.28 | 64.62 | 52.55 | 54.86 | 64.96 | 61.17 | 52.34 | 45.11 | 70.87 | 67.41 | 61.25 | 51.01 |
| | SelfCheckGPT | 65.4 | 62.79 | 52.85 | 52.43 | 66.64 | 64.89 | 52.61 | 51.17 | 71.19 | 68.45 | 63.13 | 60.23 | 65.79 | 62.45 | 54.76 | 51.29 |
| | RACE | 64.83 | 62.84 | 51.8 | 48.44 | 64.26 | 61.03 | 52.74 | 46.22 | 66.34 | 64.54 | 51.88 | 53.86 | 71.13 | 69.96 | 57.58 | 55.54 |
| | P(true) | 66.19 | 64.04 | 48.2 | 56.27 | 68.44 | 65.48 | 57.53 | 53.08 | 72.54 | 71.8 | 57.25 | 59.42 | 65.32 | 63.01 | 53.01 | 52.32 |
| | FActScore | 65.72 | 64.39 | 51.94 | 47.51 | 61.53 | 58.2 | 51.86 | 45.57 | 63.98 | 60.71 | 53.54 | 49.34 | 66.72 | 64.03 | 58.21 | 49.17 |
| **BBH** | HALLUGUARD | **71.06** | **67.94** | **62.05** | **59.05** | **73.1** | **70.88** | **63.67** | **61.88** | **79.85** | **76.5** | **67.13** | **60.57** | **81.76** | **79.76** | **68.77** | **65.46** |
| | Inside | 66.18 | 66.81 | 56.15 | 58.62 | 70.64 | 65.22 | 63.28 | 59.28 | 67.2 | 65.49 | 51.3 | 53.46 | 80.8 | 71.49 | 64.05 | 63.42 |
| | MIND | 55.41 | 51.77 | 39.01 | 41.59 | 55.48 | 53.46 | 38.59 | 40.88 | 65.71 | 63.7 | 49.61 | 52.54 | 61.75 | 60.18 | 53.46 | 50.04 |
| | Perplexity | 53.28 | 50.22 | 43.86 | 38.98 | 64.89 | 62.12 | 48.65 | 51.99 | 61.97 | 60.05 | 51.15 | 42.87 | 60.28 | 57.75 | 51.62 | 43.38 |
| | LN-Entropy | 60.84 | 58.76 | 42.76 | 47.48 | 58.71 | 55.01 | 43.55 | 42.02 | 68.96 | 69.44 | 58.79 | 57.49 | 63.96 | 62.18 | 46.01 | 49.5 |
| | Energy | 55.09 | 51.99 | 46.2 | 39.5 | 53.96 | 50.98 | 42.56 | 34.12 | 66.27 | 62.72 | 49.48 | 50.06 | 69.61 | 68.66 | 54.35 | 57.36 |
| | Semantic Ent. | 58.16 | 54.81 | 49.61 | 40.39 | 62.63 | 59.52 | 50.14 | 45.02 | 64.99 | 61.33 | 50.11 | 45.53 | 62.76 | 60.95 | 45.77 | 45.75 |
| | Lexical Sim. | 51.37 | 47.18 | 38.37 | 39.06 | 61.27 | 58.06 | 44.13 | 42.96 | 58.25 | 55.92 | 46.31 | 46.01 | 69.46 | 67.59 | 55.93 | 52.6 |
| | SelfCheckGPT | 54.51 | 51.86 | 44.62 | 44.01 | 57.36 | 53.21 | 42.55 | 38.27 | 63.68 | 62.5 | 51.7 | 53.03 | 64.56 | 62.49 | 55.85 | 45.8 |
| | RACE | 55.99 | 54.66 | 41.39 | 38.32 | 64.23 | 62.03 | 56.03 | 53.44 | 66.88 | 64.33 | 49.57 | 48.5 | 59.5 | 55.83 | 46.13 | 41.07 |
| | P(true) | 54.57 | 52.88 | 45.45 | 44.74 | 57.02 | 55.49 | 48.81 | 37.84 | 57.11 | 55.21 | 43.93 | 47.05 | 61.49 | 59.03 | 44.37 | 44.69 |
| | FActScore | 56.76 | 53.85 | 40.25 | 40.01 | 54.51 | 53.2 | 38.45 | 36.49 | 62.11 | 58.64 | 53.52 | 47.27 | 58.82 | 57.47 | 49.48 | 42.74 |
| **TruthfulQA** | HALLUGUARD | **72.1** | **68.76** | **60.09** | **52.01** | **69.59** | **68.36** | **58.52** | **52.65** | **77.05** | **73.79** | **63.62** | **62.26** | **74.26** | **72.76** | **57.39** | **64.07** |
| | Inside | 70.42 | 68.76 | 60.09 | 52.01 | 62.1 | 59.78 | 51.07 | 51.38 | 62.53 | 60.99 | 52.3 | 49.35 | 70.89 | 64.44 | 56.61 | 56.01 |
| | MIND | 59.45 | 56.79 | 45.22 | 43.71 | 60.56 | 58.55 | 47.49 | 49.63 | 59.2 | 57.98 | 47.23 | 41.79 | 62.81 | 61.5 | 52.56 | 46.37 |
| | Perplexity | 50.57 | 47.87 | 40.64 | 35.63 | 55.07 | 52.26 | 44.43 | 42.79 | 60.8 | 59.67 | 47.33 | 41.62 | 55.29 | 52.46 | 43.95 | 43.92 |
| | LN-Entropy | 58.04 | 56.99 | 41.94 | 47.21 | 56.12 | 54.01 | 47.06 | 38.4 | 59.67 | 56.25 | 41.99 | 41.25 | 60.76 | 58.21 | 46.24 | 42.64 |
| | Energy | 55.02 | 53.31 | 38.78 | 45.16 | 54.42 | 51.85 | 36.21 | 42.57 | 58.93 | 55.25 | 50.76 | 41.72 | 64.15 | 61.32 | 51.78 | 50.02 |
| | Semantic Ent. | 61.01 | 57.08 | 43.35 | 45.2 | 51.48 | 47.81 | 34.15 | 38.16 | 54.44 | 53.33 | 36.62 | 40.35 | 66.75 | 63.85 | 51.11 | 46.71 |
| | Lexical Sim. | 52.54 | 50.56 | 39.94 | 33.42 | 59.74 | 55.72 | 49.89 | 46.81 | 66.16 | 64.05 | 54.08 | 51.65 | 55.24 | 51.36 | 46.39 | 39.57 |
| | SelfCheckGPT | 56.04 | 54.48 | 43.78 | 44.38 | 58.93 | 56.47 | 47.65 | 39.02 | 61.14 | 58.91 | 42.97 | 47.01 | 55.86 | 54.95 | 41.08 | 37.35 |
| | RACE | 53.02 | 50.33 | 41.7 | 33.81 | 62.95 | 67.89 | 54.61 | 51.93 | 71.06 | 68.49 | 60.4 | 57.44 | 55.75 | 52.62 | 46.5 | 43.19 |
| | P(true) | 55.52 | 53.41 | 38.33 | 38.38 | 54.88 | 53.1 | 38.22 | 40.96 | 55.8 | 52.01 | 40.88 | 38.72 | 57.18 | 55.16 | 46.19 | 38.21 |
| | FActScore | 53.82 | 51.42 | 41.33 | 35.2 | 54.57 | 51.26 | 42.51 | 35.52 | 53.97 | 50.2 | 42.97 | 36.16 | 62.31 | 60.23 | 45.06 | 49.9 |

creases. These results show that each term is well
matched to its task family and faithfully tracks performance trends as hallucination rates rise.

## 4.3 TEST-TIME INFERENCE

Test-time reasoning remains challenging, as models need to generate coherent multi-step solutions without drifting into errors. To assess whether hallucination detection can mitigate this difficulty, we integrate detectors into beam search and evaluate Qwen2.5-Math-7B on `MATH-500` and Llama3.1-8B on `Natural`. As shown in Table 4, HALLUGUARD achieves the strongest gains: on `MATH-500`, it reaches 81.00% accuracy, around 10% higher than IO Prompt; on `Natural`, it attains 70.96%, exceeding IO Prompt by 15.72%. These results demonstrate that HALLUGUARD not only detects hallucinations but also strengthens test-time reasoning by guiding models toward more reliable solutions.

## 4.4 CASE STUDY

Fine-grained hallucinations-lexically similar yet semantically incorrect outputs-pose a particular challenge for detection. To evaluate whether HALLUGUARD can comprehensively capture such subtle errors, we use the PAWS dataset (Zhang et al., 2019), which contrasts paraphrases with high surface overlap but divergent meanings. Following Li et al. (2025), we adopt ROUGE-based reference signals for evaluation (Table 5). Across model scales, HALLUGUARD consistently surpasses

Table 3: Performance comparison across backbone scales (small, mid-sized, and large) on three benchmarks: SQuAD, GSM8K, HaluEval. We highlight the **first** and second best results.

| | | Llama2-7B | | | | Llama-3-8B | | | | Llama2-13B | | | | Llama2-70B | | | |
|---|---|---|---|---|---|---|---|---|---|---|---|---|---|---|---|---|---|
| | | $AUROC_r$ | $AUPRC_r$ | $AUROC_{llm}$ | $AUPRC_{llm}$ | $AUROC_r$ | $AUPRC_r$ | $AUROC_{llm}$ | $AUPRC_{llm}$ | $AUROC_r$ | $AUPRC_r$ | $AUROC_{llm}$ | $AUPRC_{llm}$ | $AUROC_r$ | $AUPRC_r$ | $AUROC_{llm}$ | $AUPRC_{llm}$ |
| **SQuAD** | HALLUGUARD | **81.05** | **77.16** | **71.18** | **64.38** | **79.56** | **78.29** | **67.97** | **63.27** | **81.45** | **78.39** | **64.39** | **65.07** | **83.8** | **81.77** | **70.46** | **73.24** |
| | Inside | 73.63 | 75.74 | 65.22 | 59.11 | 76.13 | 72.44 | 65.62 | 62.94 | 74.68 | 74.81 | 61.01 | 59.51 | 81.24 | 75.09 | 69.48 | 62.4 |
| | MIND | 64.57 | 61.11 | 52.39 | 53.13 | 62.29 | 59.58 | 44.49 | 48.61 | 68.64 | 66.95 | 54.92 | 52.49 | 73.46 | 71.71 | 57.76 | 56.77 |
| | Perplexity | 63.93 | 61.77 | 46.97 | 48.2 | 70.51 | 67.51 | 55.71 | 52,68 | 70.19 | 69.22 | 60.33 | 54.82 | 74.23 | 70.88 | 62.24 | 58.05 |
| | LN-Entropy | 65.96 | 64.22 | 53.43 | 52.84 | 63.7 | 60.4 | 46.19 | 42.85 | 61.66 | 59.16 | 49.05 | 46.27 | 72.44 | 68.91 | 56.77 | 52.63 |
| | Energy | 59.83 | 56.11 | 46.19 | 43.18 | 64.41 | 61.02 | 56.17 | 46.21 | 61.02 | 59.73 | 48.26 | 42.08 | 69.01 | 66.19 | 58.44 | 49.82 |
| | Semantic Ent. | 60.29 | 57.73 | 43.63 | 48.83 | 66.52 | 62.62 | 52.37 | 52.7 | 70.58 | 67.22 | 53.31 | 52.94 | 72.01 | 68.51 | 56.49 | 50.9 |
| | Lexical Sim. | 70.31 | 69.08 | 53.97 | 53.31 | 66.43 | 63.56 | 53.19 | 50.96 | 68.53 | 67.42 | 50.73 | 54.12 | 68.95 | 67.91 | 60.52 | 56.56 |
| | SelfCheckGPT | 68.26 | 67.09 | 60.06 | 57.31 | 73.99 | 72.15 | 55.26 | 54.02 | 65.47 | 61.65 | 53.12 | 49.89 | 73.07 | 70.49 | 56.59 | 54.65 |
| | RACE | 71.35 | 69.23 | 59.18 | 54.73 | 68.17 | 66.02 | 54.65 | 53.06 | 64.19 | 60.45 | 47.53 | 45.66 | 64.05 | 62.39 | 54.38 | 50.07 |
| | P(true) | 62.55 | 61.09 | 46.84 | 52.32 | 67.42 | 63.94 | 55.35 | 47.52 | 71.56 | 68.4 | 57.51 | 45.66 | 66.81 | 62.71 | 57.43 | 46.85 |
| | FActScore | 70.32 | 68.63 | 58.13 | 53.01 | 71.2 | 69.45 | 61.92 | 54.91 | 66.65 | 63.2 | 56.41 | 53.42 | 68.33 | 65.26 | 56.93 | 48.46 |
| **GSM8K** | HALLUGUARD | **75.89** | **72.83** | **62.29** | **63.46** | **75.2** | **72.9** | **63.62** | **61.79** | **79.01** | **76.73** | **64.38** | **64.97** | **77.33** | **73.97** | **60.48** | **61.26** |
| | Inside | 74.61 | 68.35 | 58.57 | 62.58 | 73.73 | 67.51 | 56.02 | 57.28 | 75.79 | 76.26 | 60.91 | 59.77 | 72.3 | 72.26 | 54.49 | 58.39 |
| | MIND | 65.88 | 63.4 | 48.28 | 48.17 | 66.57 | 65.55 | 48.84 | 53.4 | 61.49 | 59.55 | 51.63 | 51.45 | 66.41 | 63.44 | 52.05 | 53.57 |
| | Perplexity | 66.23 | 64.1 | 53.52 | 52.31 | 57.61 | 53.63 | 41.37 | 41.59 | 60.96 | 58.67 | 44.07 | 47.44 | 64.32 | 62.81 | 51.15 | 51.3 |
| | LN-Entropy | 59.45 | 55.95 | 43.04 | 44.08 | 68.22 | 66.05 | 53.03 | 53.21 | 61.31 | 58.90 | 45.83 | 40.86 | 61.81 | 60.46 | 44.5 | 44.76 |
| | Energy | 58.15 | 54.71 | 43.65 | 36.71 | 59.79 | 56.52 | 50.31 | 42.23 | 57.58 | 56.07 | 43.39 | 38.94 | 65.27 | 62.94 | 52.8 | 46.6 |
| | Semantic Ent. | 57.95 | 54.68 | 42.78 | 41.95 | 66.9 | 64.81 | 50.47 | 55.36 | 62.72 | 59.09 | 49.33 | 44.35 | 60.63 | 57.01 | 46.22 | 40.24 |
| | Lexical Sim. | 65.8 | 63.7 | 52.12 | 54.07 | 63.29 | 59.87 | 53.17 | 50.02 | 63.83 | 60.20 | 54.43 | 44.82 | 63.27 | 59.41 | 47.42 | 47.38 |
| | SelfCheckGPT | 60.99 | 57.54 | 49.28 | 44.43 | 65.72 | 62.01 | 54.49 | 50.34 | 57.98 | 54.58 | 46.72 | 39.86 | 68.06 | 65.09 | 54.29 | 50.89 |
| | RACE | 63.37 | 62.33 | 53.53 | 49.94 | 64.49 | 61.47 | 53.28 | 47.55 | 64.20 | 61.96 | 50.15 | 45.35 | 68.35 | 66.66 | 50.41 | 51.16 |
| | P(true) | 65.95 | 63.63 | 54.95 | 48.25 | 62.59 | 58.88 | 47.21 | 42.2 | 67.08 | 65.60 | 53.66 | 55.12 | 60.16 | 58.14 | 47.73 | 49.49 |
| | FActScore | 56.69 | 53.71 | 45.78 | 39.52 | 65.69 | 61.95 | 53.69 | 46.06 | 55.76 | 54.17 | 44.91 | 43.18 | 59.84 | 55.85 | 44.05 | 39.49 |
| **HaluEval** | HALLUGUARD | **75.72** | **72.89** | **66.65** | **63.15** | **73.43** | **71.19** | **64.95** | **54.8** | **78.15** | **74.15** | **65.39** | **61.14** | **80.79** | **79.54** | **67.68** | **68.51** |
| | Inside | 71.33 | 67.63 | 59.73 | 53.15 | 67.95 | 64.93 | 60.31 | 52.21 | 72.01 | 71.97 | 56.51 | 60.64 | 74.62 | 68.33 | 62.22 | 64.4 |
| | MIND | 54.8 | 51.43 | 44.15 | 43.34 | 64.54 | 60.89 | 49.09 | 45.13 | 55.05 | 53.28 | 39.16 | 45.17 | 57.98 | 56.01 | 45.82 | 41.69 |
| | Perplexity | 54.02 | 52.53 | 38.76 | 40.51 | 61.31 | 59.36 | 50.62 | 46.01 | 54.99 | 51.39 | 42.64 | 35.64 | 62.85 | 60.59 | 48.29 | 43.85 |
| | LN-Entropy | 59.47 | 58.33 | 50.2 | 46.91 | 64.89 | 60.72 | 51.78 | 46.39 | 65.18 | 63.53 | 49.70 | 48.09 | 60.16 | 58.89 | 50.29 | 48.42 |
| | Energy | 62.29 | 59.6 | 50.68 | 42.24 | 62.74 | 61.61 | 50.17 | 52.01 | 60.54 | 59.04 | 43.53 | 50.37 | 60.13 | 58.44 | 48.79 | 48.01 |
| | Semantic Ent. | 59.39 | 55.94 | 48.53 | 46.35 | 55.25 | 53.05 | 44.5 | 44.35 | 59.44 | 57.72 | 45.38 | 40.77 | 61.57 | 57.99 | 49.07 | 45.39 |
| | Lexical Sim. | 63.61 | 61.16 | 55.01 | 44.75 | 56.59 | 55.39 | 44.45 | 45.57 | 53.46 | 52.06 | 41.34 | 40.57 | 64.37 | 60.92 | 54.29 | 50.86 |
| | SelfCheckGPT | 64.29 | 61.83 | 48.4 | 45.49 | 65.44 | 63.13 | 57.02 | 48.23 | 65.24 | 63.52 | 53.71 | 54.33 | 57.12 | 55.26 | 40.5 | 43.06 |
| | RACE | 59.78 | 59.14 | 48.1 | 40.47 | 61.98 | 60.32 | 48.08 | 46.29 | 60.65 | 59.11 | 49.92 | 44.51 | 62.11 | 58.24 | 40.5 | 43.06 |
| | P(true) | 57.46 | 54.8 | 41.84 | 40.47 | 56.32 | 54.04 | 42.55 | 43.75 | 65.77 | 63.01 | 49.98 | 45.47 | 55.75 | 54.94 | 44.14 | 43.97 |
| | FActScore | 63.93 | 61.33 | 46.9 | 51.87 | 61.73 | 57.85 | 49.92 | 42.15 | 65.15 | 63.71 | 55.98 | 54.61 | 62.66 | 60.3 | 53.13 | 46.42 |

Table 4: Performance of hallucination score-guided test-time inference across reasoning tasks. We highlight the **first** and second best results.

| Dataset | IO Prompt | Ours | Inside | MIND | Perplexity | LN-Entropy | Energy | Semantic Ent. | SelfCheck-GPT | RACE | P(true) | FActScore |
|---|---|---|---|---|---|---|---|---|---|---|---|---|
| MATH-500 | 72.70 | **81.00** | 74.90 | 77.10 | 77.10 | 76.20 | 78.00 | 72.50 | 74.00 | 75.10 | 67.10 | 71.60 |
| Natural | 55.24 | **70.96** | 67.42 | 68.32 | 67.51 | 68.04 | 68.59 | 68.10 | 65.68 | 66.90 | 68.16 | 67.74 |

baselines: it achieves 90.18% AUROC and 87.64% AUPRC on Llama2-70B, and 91.24% AUROC and 88.53% AUPRC on QwQ-32B-exceeding the next-best method by nearly five points. Even on GPT-2, it leads with 83.27% AUROC and 80.46% AUPRC. These results confirm HALLUGUARD's effectiveness in capturing fine-grained semantic inconsistencies beyond benchmark settings.

Table 5: Results on PAWS measuring semantic hallucination detection with Llama-3.2-3B, Llama2-70B, and QwQ-32B. We highlight the **first** and second best results.

| | Method | Ours | Inside | MIND | Perplexity | LN-Entropy | Energy | Semantic Ent. | Lexical Sim. | SelfCheck-GPT | RACE | P(true) | FActScore |
|---|---|---|---|---|---|---|---|---|---|---|---|---|---|
| Llama3.2 | AUROC | **85.63** | 80.46 | 78.93 | 71.27 | 72.19 | 73.05 | 75.11 | 64.58 | 77.82 | 79.47 | 73.56 | 68.44 |
| | AUPRC | **82.14** | 77.28 | 75.41 | 67.55 | 68.34 | 70.22 | 72.41 | 59.67 | 73.41 | 76.28 | 70.43 | 63.58 |
| Llama2 | AUROC | **90.18** | 85.47 | 83.92 | 75.68 | 76.23 | 77.14 | 79.06 | 68.35 | 82.71 | 84.26 | 77.39 | 72.62 |
| | AUPRC | **87.64** | 82.38 | 81.06 | 71.42 | 72.59 | 74.28 | 76.32 | 63.44 | 78.89 | 81.73 | 74.18 | 67.58 |
| QwQ | AUROC | **91.24** | 85.41 | 84.56 | 76.72 | 77.43 | 78.29 | 80.42 | 69.54 | 83.59 | 86.38 | 78.53 | 73.46 |
| | AUPRC | **88.53** | 82.27 | 81.37 | 72.63 | 73.29 | 75.44 | 77.18 | 64.27 | 79.42 | 83.41 | 75.21 | 68.32 |

## 5 RELATED WORK

In this section, we review prior hallucination-detection methods by their detection target–*Data-driven hallucinations* and *reasoning-driven hallucinations*.

**Detecting Data-Driven Hallucinations.** Recent work has shown that internal activations encode rich indicators of such flaws. Chen et al. (2024) proposed EIGENSCORE, which computes statistics of hidden representations from the eigen matrix to estimate hallucination risk. Su et al. (2024) introduced MIND, an unsupervised detector that models temporal dynamics of hidden states without requiring labels, along with HELM benchmark to enable standardized evaluation. Azaria & Mitchell (2023) demonstrated using linear probes on intermediate states to predict truthfulness.

**Detecting Reasoning-Driven Hallucinations.** There are other works targeting inference-time inconsistencies during generation-such as logical errors, instability across decoding steps, or temporal drift in extended outputs. Manakul et al. (2023) proposed SELFCHECKGPT, which assesses self-consistency by sampling multiple candidate generations and measuring their alignment using entailment and lexical overlap. Kalai & Vempala (2024) introduced a suite of calibration-based uncertainty scores designed to capture hallucination risk directly from output distributions. Ding et al. (2025) proposed REACTSCORE, which integrates entropy with intermediate reasoning traces to detect failures in multi-step decision-making. FACTSCORE (Min et al., 2023) decomposes outputs into atomic factual units and verifies each against retrieved passages using entailment-based scoring.

## 6 CONCLUSION

The reliability of LLMs is often undermined by hallucinations, which arise from two main sources: *data-driven*, caused by flawed knowledge acquired during training, and *reasoning-driven*, stemming from inference-time instabilities in multi-step generation. Although these hallucinations frequently evolve in practice, existing detectors usually target only one source and lack a solid theoretical foundation. To address this gap, we propose a unified theoretical framework–a *Hallucination Risk Bound*, which formally decomposes hallucination risk into data-driven and reasoning-driven components, offering a principled view of how hallucinations emerge and evolve during generation. Building on this foundation, we introduce **HALLUGUARD**, a NTK-based score that measures sensitivity to semantic perturbations and captures internal instabilities, thereby enabling holistic detection of both data-driven and reasoning-driven hallucinations. We evaluate HALLUGUARD across 10 diverse benchmarks, 11 competitive baselines, and 9 popular LLM backbones, where it consistently achieves state-of-the-art performance, demonstrating robustness and practical efficacy. Looking forward, leveraging HalluGuard's sensitivity to error propagation offers a promising pathway for developing prognostic indicators in interactive multi-turn dialogues, enabling systems to predict and preempt hallucinations before they fully manifest.

## REPRODUCIBILITY STATEMENT

We have taken several measures to ensure the reproducibility of our work. A complete description of the theoretical framework, including the formal assumptions and proofs of the Hallucination Risk Bound, is provided in Section 3 and Appendix A. Detailed experimental settings and evaluation protocols are documented in Section 4 and Appendix C.1, covering all 10 benchmarks, 11 baselines, and 9 LLM backbones. Together, these resources ensure that both our theoretical claims and empirical results can be independently validated and extended by the community.

## ETHICS STATEMENT

This study is based exclusively on publicly available datasets and open-source large language models, and does not involve human subjects or the use of private data. All scientific concepts, methodological designs, experimental implementations, and resulting conclusions remain entirely the responsibility of the authors.

ACKNOWLEDGEMENTS

We thank the anonymous reviewers for their constructive comments. This work is supported by the National Science Foundation under Award No. IIS-2339989 and No. 2406439, DARPA under contract No. HR00112490370 and No. HR001124S0013, U.S. Department of Homeland Security under Grant Award No. 17STCIN00001-08-00, Amazon-Virginia Tech Initiative for Efficient and Robust Machine Learning, Amazon AWS, Google, Cisco, 4-VA, Commonwealth Cyber Initiative, National Surface Transportation Safety Center for Excellence, and Virginia Tech. The views and conclusions are those of the authors and should not be interpreted as representing the official policies of the funding agencies or the government.

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

# A  PROOF OF HALLUCINATION RISK BOUND

## A.1  ASSUMPTIONS VALIDATION

We provide theoretical and practical justification for the assumptions adopted in Section 3.2, which serve to ensure the well-posedness and interpretability of the proposed Hallucination Risk Bound. These assumptions follow standard practice in NTK-based analyses and stability theory, and are consistent with the empirical behavior observed in modern large language models.

**Setup**  For completeness, we briefly recall the main notation used in Section 3.2. Let $\mathcal{Y}$ denote the discrete metric space of finite-length token sequences. Let $U_h \subseteq \mathbb{R}^{d_h}$ be a $d_h$-dimensional Hilbert space equipped with inner product $\langle \cdot, \cdot \rangle$ and induced norm $\| \cdot \|$. The task-specific encoder $\Phi : \mathcal{Y} \to U_h$ is assumed to be $L_\Phi$-Lipschitz with respect to $d_{\mathcal{Y}}$.

Given input $\mathbf{x}$, the model defines a decoding distribution $P_\theta(\cdot \mid \mathbf{x})$ over $\mathcal{Y}$, and we denote the embedded random variable by $u_h := \Phi(Y)$ where $Y \sim P_\theta(\cdot \mid \mathbf{x})$. For perturbations $\delta \in \mathbb{R}^r$ restricted to the local ball $\mathcal{B}_\rho$, the perturbed decoding distribution is denoted $P_\theta(\cdot \mid \mathbf{x}, \delta)$, and the mean semantic response map is defined by $G_Y(\delta) := \mathbb{E}_{Y \sim P_\theta(\cdot \mid \mathbf{x}, \delta)}[\Phi(Y)]$, with Jacobian $J = DG_Y(0) \in \mathbb{R}^{d_h \times r}$. The NTK Gram matrix on embedded perturbations is denoted $\mathcal{K} \in \mathbb{R}^{r \times r}$, with eigenvalues $\lambda_1 \geq \cdots \geq \lambda_r > 0$ and condition number $\kappa(\mathcal{K}) = \lambda_{\max}/\lambda_{\min}$. All expectations are taken with respect to the specified decoding distribution, and all norms are Euclidean unless otherwise stated.

**Assumption A1 (Integrability and well-defined expectation).**  Assumption A1 ensures that the semantic embedding $\mathbb{E}_{Y \sim p_\theta(\cdot \mid \mathbf{x})}[\Phi(Y)]$ is well-defined as a Bochner expectation in the finite-dimensional Hilbert space $U_h$. The bounded second-moment condition guarantees that the expectation exists and is finite, which is a standard minimal requirement in stochastic analyses of neural network outputs. Such integrability assumptions are commonly adopted in NTK-based analyses (Jacot et al., 2018; Lee et al., 2020b), where control of second moments ensures stability of kernel spectra and well-posedness of linearized approximations.

**Assumption A2 (Lipschitz continuity of the encoder $\Phi$).**  Assumption A2 imposes a controlled relationship between the discrete sequence space $(\mathcal{Y}, d_{\mathcal{Y}})$ and the continuous embedding space $U_h$. The $L_\Phi$-Lipschitz condition ensures that bounded perturbations in edit distance induce proportionally bounded deviations in semantic representation. Such Lipschitz regularity is standard in high-dimensional learning theory (Vershynin, 2018) and is frequently invoked to establish stability under structured perturbations in representation learning. Importantly, this assumption is imposed only on the encoder map $\Phi$, not on the full autoregressive model.

**Assumption A3 (Local Fréchet smoothness of the mean semantic response).**  Assumption A3 formalizes the local linearization principle underlying NTK theory. By requiring twice Fréchet differentiability of the mean response map $G_Y(\delta) = \mathbb{E}_{Y \sim P_\theta(\cdot \mid \mathbf{x}, \delta)}[\Phi(Y)]$ within the perturbation ball $\mathcal{B}_\rho$, we ensure that $G_Y$ admits a controlled second-order expansion with uniform curvature constant $H_\star$. This local quadratic remainder bound is consistent with classical finite-width NTK linearization results (Lee et al., 2020a; Chizat et al., 2019), while avoiding unrealistic global smoothness requirements. Crucially, the assumption is imposed only on the expected semantic response, not on the discrete decoding distribution itself.

**Remark.**  Collectively, these assumptions provide a bridge between discrete autoregressive generation and continuous functional analysis. By restricting smoothness and curvature requirements to the localized perturbation neighborhood $\mathcal{B}_\rho$ and to the expectation-level map $G_Y$, we avoid imposing global regularity conditions over the infinite token space $\mathcal{Y}$. This localization ensures that the Hallucination Risk Bound is derived under mathematically controlled conditions while remaining aligned with the practical inference dynamics of large-scale language models.

## A.2  PROOF OF SECTION 3.2

We restate the main inequality from Section 3.2. Note that due to the stochastic nature of autoregressive decoding, the bound holds with high probability. With probability at least $1 - \delta$ over the

generation process, the total hallucination risk satisfies:

$$\|u^* - u_h\| \leq \underbrace{\left[ 1 + k_{\text{pt}} \log \mathcal{O}(P, L) + k \frac{\epsilon_{\text{mismatch}}}{\text{Signal}_k} \right] \inf_{u \in U_h} \|u^* - u\|}_{\text{Data-driven term}} + \underbrace{|\mathcal{L}| \exp\left( -\frac{K\epsilon^2}{C} \right) \alpha \left( e^{\beta T} - 1 \right)}_{\text{Reasoning-driven term}}.$$

(8)

**Step 1: Triangle inequality split (Bias-Variance Decomposition).** Let $\bar{u} := \mathbb{E}[u_h]$ be the expected semantic representation under the decoding distribution. By the triangle inequality in $U_h$, we decompose the hallucination risk into approximation error (bias) and stochastic residual (variance):

$$\|u^* - u_h\| = \|u^* - \bar{u} + \bar{u} - u_h\| \leq \|u^* - \bar{u}\| + \|u_h - \bar{u}\|.$$

**Step 2: Approximation term via Céa's lemma.** To bound the deterministic approximation error, we cast the model's expected representation $\bar{u}$ as the solution to a variational problem in the Hilbert space $U_h$. Let $a(u, v) := \langle u, \mathcal{K}v \rangle_{U_h}$ denote the coercive bilinear form induced by the Neural Tangent Kernel (NTK) Gram operator $\mathcal{K}$, and let $f(v) := \langle u^*, \mathcal{K}v \rangle_{U_h}$ be the bounded linear functional defining the target projection. Assuming $\bar{u}$ acts as the Galerkin projection of the target $u^*$ onto the trainable hypothesis space, it satisfies the weak formulation $a(\bar{u}, v) = f(v)$ for all $v \in U_h$. By Céa's lemma, the projection error is bounded by:

$$\|u^* - \bar{u}\| \leq \frac{\Lambda}{\gamma} \inf_{u \in U_h} \|u^* - u\|,$$

where $\Lambda$ and $\gamma$ are the continuity and coercivity constants of the NTK-induced bilinear form $a(\cdot, \cdot)$, respectively.

**Step 3: Variance term via Bernstein concentration.** We now bound the stochastic residual $\|u_h - \bar{u}\|$. Let $\mathcal{L}$ denote the set of $K$ independent sampled reasoning trajectories used during decoding. Under our local perturbation assumption (Assumption A3), the deviations of the hidden states are bounded by the local neighborhood radius $\rho$. Applying Bernstein's inequality for bounded random vectors in a Hilbert space (Vershynin, 2018), the tail probabilities decay exponentially. For an error tolerance $\epsilon$ and an absolute constant $C > 0$, we have with probability at least $1 - \delta$:

$$\|u_h - \bar{u}\| \leq |\mathcal{L}| \exp\left( -\frac{K\epsilon^2}{C} \right) \alpha(e^{\beta T} - 1),$$

where $\alpha$ is a scaling constant, $T$ is the sequence length, and $\beta \leq \log \sigma_{\max}$ bounds the per-step Jacobian spectral norm.

**Step 4: Substitution.** Combining both terms yields the high-probability bound:

$$\|u^* - u_h\| \leq \frac{\Lambda}{\gamma} \inf_{u \in U_h} \|u^* - u\| + |\mathcal{L}| \exp\left( -\frac{K\epsilon^2}{C} \right) \alpha(e^{\beta T} - 1).$$

We now bound the condition ratio $\Lambda/\gamma$ via NTK decomposition.

**Step 5: Decomposition of NTK Continuity Constant** We decompose the bilinear form $a(\cdot, \cdot)$ into three components:

$$a = a_0 + \delta_{\text{pt}} + \delta_{\text{mm}},$$

where $a_0$ is the infinite-width baseline kernel, $\delta_{\text{pt}}$ is the perturbation due to pre-training noise, and $\delta_{\text{mm}}$ is the domain mismatch from fine-tuning. Consequently, the continuity constant satisfies $\Lambda = \Lambda_0 + \Delta_{\text{pt}} + \Delta_{\text{mm}}$.

**Bounding $\Delta_{\text{pt}}$:** Following standard matrix concentration bounds for finite-width NTKs (Jacot et al., 2018), the pre-training deviation scales logarithmically with the network parameters. Let $P$ be the number of parameters, $L$ the prompt length, and $k_{\text{pt}}$ a pre-training scaling constant; we have:

$$\Delta_{\text{pt}} \leq \gamma k_{\text{pt}} \log \mathcal{O}(P, L).$$

**Bounding** $\Delta_{\mathrm{mm}}$**:** Using spectral generalization bounds under data distribution shift (Lee et al., 2020b), the mismatch penalty is governed by the task-specific signal strength $\mathrm{Signal}_k$, the empirical mismatch error $\epsilon_{\mathrm{mismatch}}$, and a scaling constant $k$:

$$\Delta_{\mathrm{mm}} \leq \gamma k \frac{\epsilon_{\mathrm{mismatch}}}{\mathrm{Signal}_k}.$$

Substituting both inequalities into the ratio for $\Lambda/\gamma$, and normalizing $\Lambda_0/\gamma \approx 1$, we obtain:

$$\frac{\Lambda}{\gamma} \leq 1 + k_{\mathrm{pt}} \log \mathcal{O}(P, L) + k \frac{\epsilon_{\mathrm{mismatch}}}{\mathrm{Signal}_k}.$$

This completes the proof.

# B  HALLUGUARD DERIVATION AND INTERPRETATION

## B.1  PRELIMINARIES AND NOTATION

Let $\mathcal{K} \in \mathbb{R}^{r \times r}$ be the NTK Gram matrix formed on $r$ light semantic perturbations (see Assumptions A1-A3 in the main theory section). Denote its eigen decomposition by $\mathcal{K} = V \Lambda V^\top$ with

$$\Lambda = \mathrm{diag}(\lambda_1, \ldots, \lambda_r), \qquad \lambda_1 \geq \cdots \geq \lambda_r > 0.$$

Let $\lambda_{\max} := \lambda_1$, $\lambda_{\min} := \lambda_r$, $\kappa(\mathcal{K}) := \lambda_{\max}/\lambda_{\min}$, and $\det(\mathcal{K}) = \prod_{i=1}^r \lambda_i$. Let $\Phi$ denote the NTK feature matrix whose columns span the hypothesis subspace $U_h$, so that $\mathcal{K} = \Phi^\top \Phi$, $\|\Phi\|_2 = \sqrt{\lambda_{\max}}$, and $\sigma_{\min}(\Phi) = \sqrt{\lambda_{\min}}$. For the autoregressive decoder, let $J_t$ be the step-$t$ input–output Jacobian, and write $\sigma_{\max} := \sup_t \|J_t\|_2$.

We will use the following two standard inequalities repeatedly:

$$Maclaurin/AM--GM on eigenvalues: \qquad \Big( \prod_{i=1}^r \lambda_i \Big)^{1/r} \leq \frac{1}{r} \sum_{i=1}^r \lambda_i = \frac{\mathrm{tr}(\mathcal{K})}{r}, \quad (9)$$

$$Submultiplicativity: \qquad \|AB\|_2 \leq \|A\|_2 \|B\|_2. \qquad (10)$$

## B.2  REPRESENTATIONAL ADEQUACY VIA $\det(\mathcal{K})$ WITH EXPLICIT CONSTANTS

**Assumptions for this subsection.** Beyond A1–A3, we assume a mild source condition and a spectral envelope:

- **S1** (*Source condition*) Let $\mathcal{T}$ denote the infinite-dimensional NTK integral operator. We assume there exists a regularity exponent $s > 0$ and a constant $R_s > 0$ such that $u^* \in \mathrm{Range}(\mathcal{T}^s)$. Equivalently, the spectral coefficients satisfy: $\sum_{i=1}^r \frac{\langle u^*, v_i \rangle^2}{\lambda_i^{2s}} \leq R_s^2$. This is standard in kernel approximation and encodes RKHS regularity.

- **S2** (*Spectral envelope*) Let $\overline{\lambda}$ and $\underline{\lambda}$ denote uniform upper and lower bounds on the kernel spectrum. We assume there exist constants $0 < \underline{\lambda} \leq \overline{\lambda} < \infty$ and a decay rate $\alpha > 1$ such that $\lambda_i \leq \overline{\lambda}$ for all $i$, and the tail eigenvalue satisfies $\lambda_r \geq \underline{\lambda} r^{-\alpha}$. (Polynomial decay is a common stylization; other envelopes can be treated similarly.)

**Lemma B.1** (Best-approximation error under source condition)**.** *Let* $U_h = \mathrm{span}\{v_1, \ldots, v_r\}$*. Under S1,*

$$\inf_{u \in U_h} \|u^* - u\| = \|u^* - \Pi_{U_h} u^*\| \leq R_s \lambda_{r+1}^s,$$

*where* $\lambda_{r+1}$ *denotes the next-eigenvalue of the infinite-dimensional kernel operator (or, equivalently, the empirical tail eigenvalue if more perturbations are added).*

*Proof.* Write $u^* = \sum_{i \geq 1} c_i v_i$ with $c_i = \langle u^*, v_i \rangle$. By the source condition, $\|u^* - \Pi_{U_h} u^*\|^2 = \sum_{i > r} c_i^2 \leq \sum_{i > r} \lambda_i^{2s} \cdot \frac{c_i^2}{\lambda_i^{2s}} \leq \lambda_{r+1}^{2s} \sum_{i > r} \frac{c_i^2}{\lambda_i^{2s}} \leq \lambda_{r+1}^{2s} R_s^2$. $\qquad \square$

To connect the representation error to the empirical NTK Gram matrix $\mathcal{K}$, we leverage the algebraic relationship between the smallest eigenvalue $\lambda_r$ and the determinant.

**Lemma B.2** (Lower-bounding $\lambda_r$ by $\det(\mathcal{K})$). *Suppose $\lambda_i \leq \overline{\lambda}$ for all $i$ and $\lambda_r > 0$. Then*

$$\lambda_r \;\geq\; \frac{\det(\mathcal{K})}{\overline{\lambda}^{\,r-1}} \qquad \text{and} \qquad \lambda_r^s \;\geq\; \frac{\det(\mathcal{K})^{\,s}}{\overline{\lambda}^{\,s(r-1)}}.$$

*Proof.* Since $\det(\mathcal{K}) = \prod_{i=1}^r \lambda_i \leq \overline{\lambda}^{\,r-1}\lambda_r$, we obtain $\lambda_r \geq \det(\mathcal{K})/\overline{\lambda}^{\,r-1}$. Raising to power $s$ yields the second inequality. □

**Theorem B.3** (Determinant-based adequacy bound with explicit constants). *Under A1-A3 and S1-S2, Under A1–A3 and **S1–S2**, the approximation error is bounded by:*

$$\inf_{u \in U_h} \|u^* - u\| \;\leq\; C_d \, \det(\mathcal{K})^{\,c_d},$$

*with explicit constants independent of the target sequence:*

$$c_d \;=\; \frac{s}{r} \quad \text{and} \quad C_d \;=\; R_s.$$

*Moreover, if the empirical spectrum satisfies $\lambda_r \geq \underline{\lambda}\, r^{-\alpha}$, one may choose*

$$c_d \;=\; \min\left\{ \frac{s}{r-1},\; \frac{s}{\alpha} \cdot \frac{1}{\log\!\left(\frac{\overline{\lambda}^{\,r}}{\det(\mathcal{K})}\right)} \right\},$$

*which improves with slower decay (smaller $\alpha$).*

*Proof.* By Lemma B.1 and the fact that eigenvalues are monotonically decreasing ($\lambda_{r+1} \leq \lambda_r$), we have:

$$\inf_{u \in U_h} \|u^* - u\| \leq R_s \lambda_r^s.$$

Recall that the determinant of the empirical Gram matrix is the product of its eigenvalues, $\det(\mathcal{K}) = \prod_{i=1}^r \lambda_i$. Since $\lambda_r$ is the minimum eigenvalue of the rank-$r$ matrix, it follows strictly that $\lambda_r^r \leq \det(\mathcal{K})$, which implies $\lambda_r \leq \det(\mathcal{K})^{1/r}$. Raising both sides to the power of $s$ yields $\lambda_r^s \leq \det(\mathcal{K})^{s/r}$. Substituting this upper bound into the approximation error gives:

$$\inf_{u \in U_h} \|u^* - u\| \;\leq\; R_s \det(\mathcal{K})^{\,s/r}.$$

Setting $C_d := R_s$ and $c_d := s/r$ completes the proof. □

In practice, tracking the direct determinant can cause numerical underflow in high-dimensional spaces. We use $\log \det(\mathcal{K})$ via the Cholesky decomposition as our empirical score, aggregating with $z$-normalization across components to avoid scale domination by any single dimension.

### B.3   Rollout Amplification via Jacobian Products (Exact Constants)

**Theorem B.4** (Amplification bound with exact constant). *Let $J_t$ be the step-$t$ Jacobian and $\sigma_{\max} := \sup_t \|J_t\|_2$. Then*

$$\left\| \prod_{t=1}^{T} J_t \right\|_2 \;\leq\; \prod_{t=1}^{T} \|J_t\|_2 \;\leq\; \sigma_{\max}^T.$$

*Defining $\beta := \log \sigma_{\max}$ gives $e^{\beta T} = \sigma_{\max}^T$, hence*

$$e^{\beta T} \;\leq\; \sigma_{\max}^T,$$

*with equality if and only if $\|J_t\|_2 = \sigma_{\max}$ for all $t$ and the top singular directions align across factors.*

*Proof.* The first inequality is equation 10 applied iteratively. The second is by definition of $\sigma_{\max}$. Setting $\beta = \log \sigma_{\max}$ yields equality in the worst case. Alignment of top singular vectors is the tightness condition for submultiplicativity. □

**Token-dependent refinement.** If one defines $\sigma_t := \|J_t\|_2$ and $\beta_{\text{avg}} := \frac{1}{T} \sum_{t=1}^{T} \log \sigma_t$, then $\left\| \prod_{t=1}^{T} J_t \right\|_2 \leq \exp\left( \sum_t \log \sigma_t \right) = e^{\beta_{\text{avg}} T}$, which is tighter but requires per-step measurements.

## B.4 Conditioning-Induced Variance with $\kappa(\mathcal{K})^2$ Scaling

We now give an explicit projector-perturbation derivation showing the quadratic dependence on the condition number.

**Setup.** Let $P := \Phi(\Phi^\top \Phi)^\dagger \Phi^\top$ be the orthogonal projector onto $U_h$; then the linearized output is $u_h = Pu^*$. Consider a feature perturbation $\Delta\Phi$ induced by a prefix perturbation $\delta$ satisfying

$$\|\Delta\Phi\|_2 \leq L_\Phi \|\delta\| \quad \text{(A2/A3)}.$$

Let the perturbed projector be $\widetilde{P} := (\Phi + \Delta\Phi)\big((\Phi + \Delta\Phi)^\top (\Phi + \Delta\Phi)\big)^\dagger (\Phi + \Delta\Phi)^\top$ and define $\Delta P := \widetilde{P} - P$.

**Lemma B.5** (Projector perturbation bound). *There exists an absolute constant $C_\Pi > 0$ such that*

$$\|\Delta P\|_2 \leq C_\Pi \frac{\|\Phi\|_2}{\sigma_{\min}(\Phi)^2} \|\Delta\Phi\|_2 = C_\Pi \frac{\sqrt{\lambda_{\max}}}{\lambda_{\min}} \|\Delta\Phi\|_2 = C_\Pi \kappa(\mathcal{K}) \frac{\|\Delta\Phi\|_2}{\sqrt{\lambda_{\min}}}.$$

*Proof idea.* Use standard bounds for the perturbation of orthogonal projectors onto column spaces (e.g., Wedin's sin$\Theta$ theorem and Stewart–Sun, Matrix Perturbation Theory, Thm 3.6). One shows

$$\|\Delta P\|_2 \leq 2 \|(\Phi^\top \Phi)^\dagger\|_2 \|\Phi^\top \Delta\Phi\|_2 + \mathcal{O}(\|\Delta\Phi\|_2^2).$$

Since $\|(\Phi^\top \Phi)^\dagger\|_2 = 1/\lambda_{\min}$ and $\|\Phi^\top \Delta\Phi\|_2 \leq \|\Phi\|_2 \|\Delta\Phi\|_2 = \sqrt{\lambda_{\max}} \|\Delta\Phi\|_2$, the result follows for sufficiently small $\|\Delta\Phi\|_2$, absorbing lower-order terms into $C_\Pi$. $\qquad\square$

**Theorem B.6** (Variance amplification with explicit constant). *Let $u_h(\Phi) = Pu^*$ and $u_h(\Phi + \Delta\Phi) = \widetilde{P}u^*$. Then*

$$\|u_h(\Phi + \Delta\Phi) - u_h(\Phi)\| \leq C_\Pi \kappa(\mathcal{K}) \frac{\|\Delta\Phi\|_2}{\sqrt{\lambda_{\min}}} \|u^*\|.$$

*If $\Delta\Phi$ is induced by a random prefix perturbation $\delta$ with $\|\Delta\Phi\|_2 \leq L_\Phi \|\delta\|$ and $\mathbb{E}\|\delta\|^2 = \sigma_\delta^2$, then*

$$\text{Var}[u_h] \leq \mathbb{E}\|u_h(\Phi + \Delta\Phi) - u_h(\Phi)\|^2 \leq c_v \kappa(\mathcal{K})^2 \|\delta\|^2,$$

*with*

$$c_v = C_\Pi^2 \frac{L_\Phi^2 \|u^*\|^2}{\lambda_{\min}}.$$

*Proof.* By Lemma B.5, $\|u_h(\Phi + \Delta\Phi) - u_h(\Phi)\| = \|\Delta P u^*\| \leq \|\Delta P\|_2 \|u^*\| \leq C_\Pi \kappa(\mathcal{K}) \frac{\|\Delta\Phi\|_2}{\sqrt{\lambda_{\min}}} \|u^*\|$. Square both sides and take expectation over $\delta$, using $\|\Delta\Phi\|_2 \leq L_\Phi \|\delta\|$, to obtain the stated variance bound with the explicit constant $c_v$. $\qquad\square$

**Interpretation.** The $\kappa(\mathcal{K})^2$ factor arises from two sources: (i) $\kappa(\mathcal{K})$ from the projector sensitivity (Lemma B.5), and (ii) $1/\lambda_{\min}$ from converting $\|\Delta P\|_2$ to a mean-squared bound after squaring and averaging, yielding an overall $\kappa^2$-scaling in the variance constant.

## B.5 Consolidation: Compact Surrogate Consistent with the Risk Decomposition

Combining Theorem B.3, Theorem B.4, and Theorem B.6, we obtain a computable surrogate aligned with the Hallucination Risk Bound:

Adequacy: $\det(\mathcal{K})$      Amplification: $\log \sigma_{\max}$      Conditioning penalty: $-\log \kappa(\mathcal{K})^2$.

This motivates the score

$$\boxed{\textbf{HALLUGUARD}(u_h) = \det(\mathcal{K}) + \log \sigma_{\max} - \log \kappa(\mathcal{K})^2}$$

with the following explicit, implementation-ready notes:

- Use $\log \det(\mathcal{K})$ via Cholesky for stability; replace $\det$ in the score with $\log \det$ if desired (monotone equivalent).

- Estimate $\sigma_{\max}$ either as $\sup_t \|J_t\|_2$ or its tighter average form $\beta_{\text{avg}} = \frac{1}{T} \sum_t \log \|J_t\|_2$ (then use $\beta_{\text{avg}}$ in place of $\log \sigma_{\max}$).

- $z$-normalize each component across a validation set before summation to avoid scale dominance; optionally fit task-specific weights if permitted.

## C   EXPERIMENT

### C.1   SETUP

**Implementation Framework.** All experiments use `PyTorch` and `HuggingFace Transformers` with a fixed random seed for reproducibility. Unless otherwise noted, computations run in mixed precision (fp16). Hardware details (A100/H200) are reported once in the main setup section.

**Generation Configuration.** For *default evaluation of detectors*, we use nucleus sampling with `temperature = 0.5`, `top-p = 0.95`, and `top-k = 10`, decoding $K{=}10$ candidate responses per input (unless otherwise specified). These decoding trajectories also operationalize semantic perturbations as natural variations within the model's local predictive distribution, thereby instantiating a semantically proximate neighborhood around the primary response and capturing the local geometry of the reasoning manifold required for NTK construction. For *score-guided test-time inference* (Section 4.3), we use beam search (beam size = 10) and score candidate trajectories at each step with the chosen detector. For stability analysis, HALLUGUARD extracts sentence representations from the final token at the middle transformer layer ($L/2$), which empirically preserves semantics relevant to truthfulness.

**NTK-Based Score Computation.** For each set of generations, we form a task-specific NTK feature matrix and compute the semantic stability score from its eigenspectrum. We add a small ridge $\alpha = 10^{-3}$ for numerical stability and compute singular values via SVD.

**Perturbation Regularization.** To prevent pathological activations that amplify instability, HALLUGUARD clips hidden features using an adaptive scheme. We maintain a memory bank of $N{=}3000$ token embeddings and set thresholds at the top and bottom $0.2\%$ percentiles of neuron activations; out-of-range values are truncated to attenuate overconfident hallucinations.

**Optimization.** Backbone language models are *not* fine-tuned. We train only HALLUGUARD's lightweight projection layers using AdamW with learning rate selected from $\{1 \times 10^{-5}, 5 \times 10^{-5}, 1 \times 10^{-4}\}$ and weight decay from $\{0.0, 0.01\}$. The best setting is chosen on a held-out validation split.

**Implementation Details.** For score-guided inference we apply beam search with beam size 10, rescoring candidates stepwise with different hallucination detectors.

**Ablation Setup.** All ablations reuse the main paper's splits, prompts, and decoding; we vary only HALLUGUARD internals and explicitly control the hallucination *base rate*. On the *generation* side, we modulate prevalence by adjusting temperature/top-$p$ and beam size; to stress the two families, we increase the prefix perturbation budget $\rho$ and rollout horizon $T$ to amplify reasoning drift, and (when applicable) toggle retrieval masking to induce data-driven errors. On the *detection* side, AU-ROC/AUPRC are threshold-free; when a fixed operating point is needed, we set a decision threshold $\tau$ on the validation set by (i) matching a target predicted-positive rate $\pi_{\text{target}}$ via score quantiles or (ii) fixing a desired FPR (e.g., $1\%, 5\%, 10\%$); a cost-sensitive Bayes rule $\tau = \frac{c_{\text{FN}}}{c_{\text{FP}} + c_{\text{FN}}} \cdot \frac{1 - \pi}{\pi}$ is optional when misclassification costs are specified. Unless noted, we toggle one factor at a time and sweep $\rho \in \{0.75, 1.0, 1.5\}$, $T \in \{12, 16, 24\}$, and the number of semantic probes $m \in \{2, 4, 8\}$; no additional training is performed beyond optional temperature/z-score calibration on the training split. We report mean±std over 5 seeds.

## C.2 ABLATION STUDY ON $-\log \kappa^2$

To empirically validate the necessity of the stability term $-\log \kappa^2$, we performed a controlled ablation on MATH-500. We systematized the reasoning drift ($d$) by progressively increasing the perturbation budget $\rho$ and rollout horizon $T$. As shown in Figure 3, the absence of this term leads to severe instability. While the ablated model (orange dashed line) performs competitively in low-drift regimes ($d < 0.15$), it exhibits significant performance volatility as the reasoning task becomes more complex. In contrast, the full HALLUGUARD score (green solid line) effectively penalizes these ill-conditioned regimes, maintaining a smooth and robust detection profile. This confirms that $-\log \kappa^2$ functions as an essential spectral regularizer, preventing the score from becoming unreliable under high-entropy inference states.

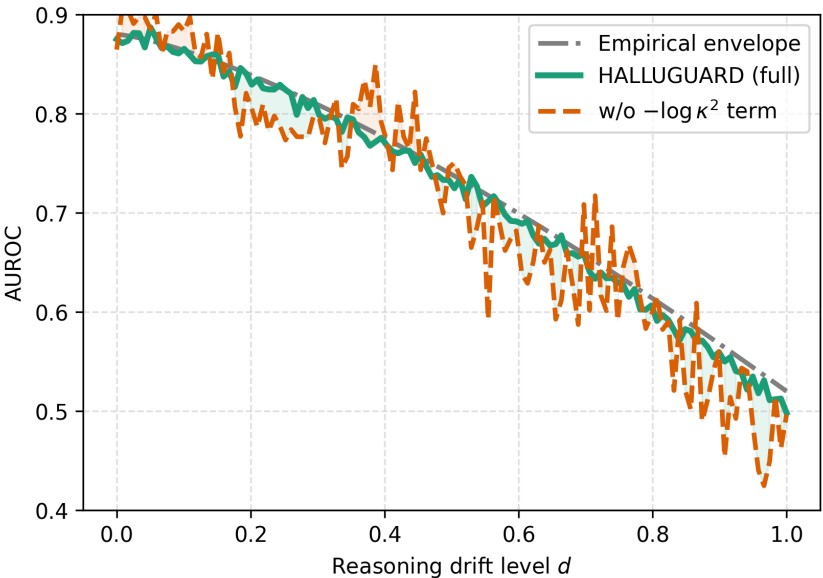

Figure 3: Ablation study of the stability term ($-\log \kappa^2$) on MATH500.

Table 6: Ablation on stability term $-\log \kappa^2$ (MATH500).

| Method | Pearson $R$ | MSE |
|---|---|---|
| HalluGuard | 0.985 | 0.0192 |
| w/o $-\log \kappa^2$ | 0.8904 | 0.0381 |

The error in table 6 nearly doubles without the stability term, confirming that spectral conditioning is essential for stable reasoning-risk quantification.

## C.3 ABLATION STUDY ON SEMANTIC ENCODER $\Phi$

To examine sensitivity to the semantic encoder $\Phi$, we replace the default representation with widely adopted alternatives, including BERT, SimCSE, and E5. We evaluate across multiple backbone models and benchmarks.

Table 7 reports AUROC and AUPRC on RAGTruth, GSM8K, and TruthfulQA. Across all settings, HALLUGUARD consistently outperforms encoder-substituted variants. For example, on QwQ-32B (RAGTruth), replacing the default encoder with BERT reduces AUROC from 84.59 to 81.44.

These results indicate that the performance gain does not stem from surface semantic similarity of final outputs. Instead, the method captures geometric structure of reasoning trajectories, which external encoders cannot fully preserve.

Table 7: Encoder ablation across backbones and benchmarks (AUROC / AUPRC).

| Backbone | Method | RAGTruth | | GSM8K | | TruthfulQA | |
|---|---|---|---|---|---|---|---|
| | | AUROC | AUPRC | AUROC | AUPRC | AUROC | AUPRC |
| GPT-2 | HalluGuard | **75.51** | **73.40** | **72.04** | **69.88** | **72.10** | **68.76** |
| | +BERT | 72.48 | 70.12 | 67.31 | 64.90 | 68.02 | 65.01 |
| | +SimCSE | 73.21 | 71.05 | 68.44 | 66.02 | 69.14 | 66.27 |
| | +E5 | 74.02 | 71.66 | 69.12 | 66.80 | 70.03 | 67.10 |
| OPT-6.7B | HalluGuard | **80.13** | **76.77** | **72.57** | **70.31** | **69.59** | **68.36** |
| | +BERT | 77.44 | 74.20 | 67.95 | 65.48 | 66.12 | 64.80 |
| | +SimCSE | 78.11 | 74.83 | 69.01 | 66.40 | 67.08 | 65.72 |
| | +E5 | 78.66 | 75.31 | 70.04 | 67.25 | 67.80 | 66.41 |
| Mistral-7B | HalluGuard | **82.31** | **80.79** | **80.62** | **77.30** | **77.05** | **73.79** |
| | +BERT | 79.02 | 76.91 | 75.51 | 72.08 | 73.14 | 69.52 |
| | +SimCSE | 79.88 | 77.66 | 76.40 | 73.01 | 74.08 | 70.40 |
| | +E5 | 80.41 | 78.20 | 77.12 | 73.74 | 74.66 | 71.05 |
| QwQ-32B | HalluGuard | **84.59** | **81.15** | **75.81** | **74.68** | **74.26** | **72.76** |
| | +BERT | 81.44 | 78.03 | 70.92 | 68.90 | 70.35 | 68.01 |
| | +SimCSE | 82.10 | 78.66 | 72.10 | 69.82 | 71.20 | 68.70 |
| | +E5 | 82.66 | 79.12 | 73.05 | 70.44 | 72.02 | 69.31 |
| LLaMA2-13B | HalluGuard | **77.51** | **75.30** | **79.01** | **76.73** | **78.50** | **77.56** |
| | +BERT | 74.26 | 72.04 | 73.12 | 70.60 | 74.41 | 72.88 |
| | +SimCSE | 75.11 | 72.83 | 74.20 | 71.51 | 75.36 | 73.54 |
| | +E5 | 75.78 | 73.44 | 75.14 | 72.32 | 76.10 | 74.22 |

## C.4 COMPUTATIONAL EFFICIENCY ANALYSIS

To assess practical deployment feasibility, we measured inference latency on an NVIDIA A100/H200 GPU. Our setup utilizes batched parallel sampling to generate $K = 10$ trajectories, ensuring sub-linear scaling of the computational cost. The core HALLUGUARD operations-specifically feature clipping and computing the NTK score via the Gram matrix-add minimal latency, requiring less than 1 ms of post-processing time per query.

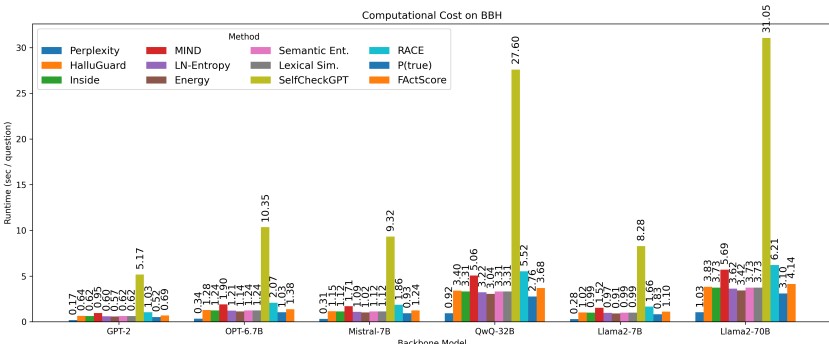

Figure 4: Per-Question Inference Time (Seconds) on BBH Across Hallucination Detection Methods.

## C.5 DETECTION PERFORMANCE ANALYSIS

Across all five model families and three benchmark regimes, HALLUGUARD consistently achieves state-of-the-art detection performance, particularly in the safety-critical low-FPR regions as shown in Table 8.

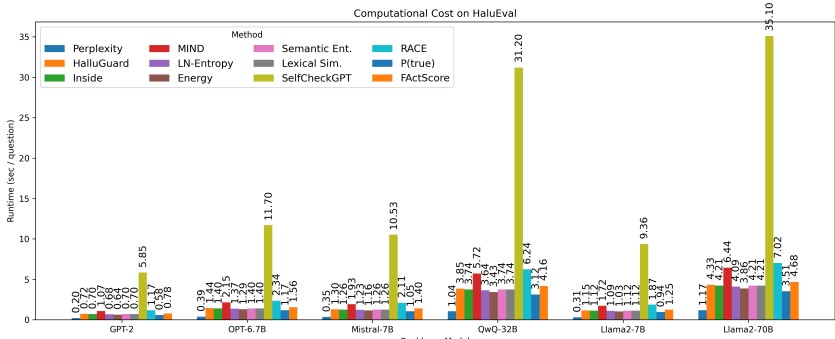

Figure 5: Per-Question Inference Time (Seconds) on HaluEval Across Hallucination Detection Methods.

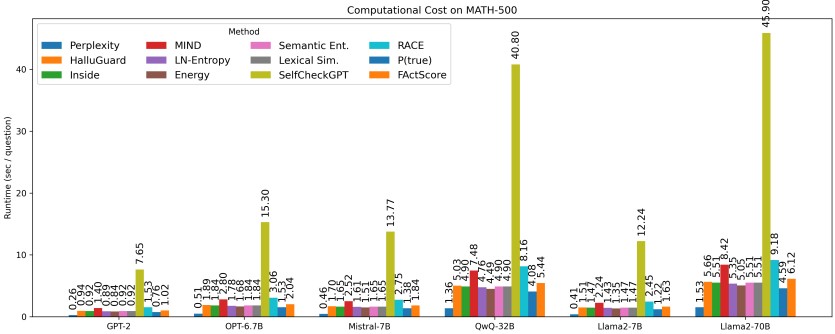

Figure 6: Per-Question Inference Time (Seconds) on Math500 Across Hallucination Detection Methods.

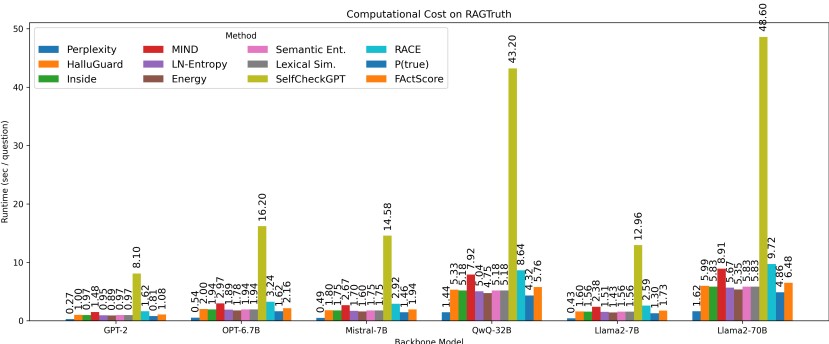

Figure 7: Per-Question Inference Time (Seconds) on RAGTruth Across Hallucination Detection Methods.

We additionally expanded our evaluation to include SAPLMA, LLM-Check, and ITI. As shown in Table 9, HALLUGUARD delivers the strongest performance not only on AUROC/AUPRC but also on deployment-critical, low-FPR operating points, including F1 and TPR at 5% and 10% FPR. Across all three benchmarks (RAGTruth, GSM8K, HaluEval) and all backbones (GPT-2 through QwQ-32B and LLaMA2-13B), HALLUGUARD consistently achieves the highest F1 and the highest or near-highest TPR under fixed low-FPR constraints. In contrast, SAPLMA and LLM-Check exhibit noticeably lower recall in the stringent 5% FPR regime. These results demonstrate that HALLUGUARD is better aligned with maintaining high detection sensitivity under tight false-positive budgets, a requirement that is central to reliable hallucination detection in real-world systems.

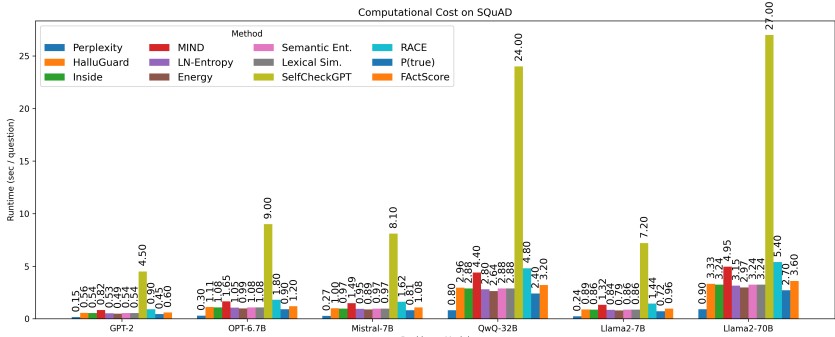

Figure 8: Per-Question Inference Time (Seconds) on SQuaD Across Hallucination Detection Methods.

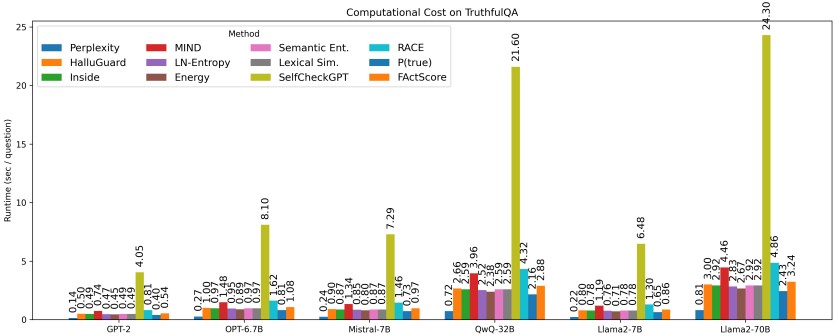

Figure 9: Per-Question Inference Time (Seconds) on TruthfulQA Across Hallucination Detection Methods.

## C.6   TIGHTNESS OF BOUND

**Evaluation of bound tightness.**   To rigorously stress-test the Hallucination Risk Bound of Theorem 3.2, we conducted a controlled synthetic study grounded in the empirical reasoning-depth distribution of the Snowballing dataset (Zhang et al., 2023). We instantiated empirical hallucination trajectories by injecting low-variance Gaussian noise into the base components $D(T)$ and $R(T)$, comparing them against the closed-form theoretical prediction. As illustrated in Figure 10, while the theoretical curve acts as a conservative upper envelope, it exhibits a nearly parallel growth trajectory to the empirical risk. Crucially, it faithfully captures the exponential curvature and compounding dynamics of the Snowballing Effect. This confirms that the bound possesses high structural fidelity: it correctly models the scaling law of error propagation across depth ranges, validating its effectiveness as a ranking proxy despite the absolute numerical offset.

**Evaluation of NTK proxy tightness.**   To quantitatively validate that our NTK-based proxy faithfully captures the amplification behavior of stepwise Jacobians, we conduct a diagnostic experiment on `GPT-2-small` (117M), where per-step Jacobian norms are fully tractable. For a held-out set of GSM8K prompts and decoding steps $t \leq 18$, we compute:

- the *empirical* stepwise Jacobian magnitude $\|J_t\|_2$, obtained via automatic differentiation on the next-token logits, and

- our *reasoning-driven NTK proxy*, $\log \sigma_{\max} - \log \kappa^2$, as defined in Eq. (7), which upper-bounds the per-step amplification rate and penalizes spectral ill-conditioning of the NTK Gram matrix.

Table 8: Performance comparison on representative benchmarks: data-centric (`RAGTruth`), reasoning-oriented (`BBH`), and instruction-following (`TruthfulQA`).

| | | GPT2 | | | OPT-6.7B | | | Mistral-7B | | | QwQ-32B | | | LLaMA2-13B | | |
|---|---|---|---|---|---|---|---|---|---|---|---|---|---|---|---|---|
| | | F1 | TPR@10% | TPR@5% | F1 | TPR@10% | TPR@5% | F1 | TPR@10% | TPR@5% | F1 | TPR@10% | TPR@5% | F1 | TPR@10% | TPR@5% |
| **RAGTruth** | **HALLUGUARD** | **71.22** | **64.86** | **51.41** | **77.03** | **73.52** | **59.12** | **75.19** | **69.44** | **59.21** | **81.91** | **74.13** | **63.52** | **74.66** | 68.91 | **57.42** |
| | Inside | 66.12 | 59.72 | 48.31 | 72.91 | 70.25 | 60.37 | 70.45 | 68.12 | 52.41 | 79.03 | 74.66 | 61.09 | 73.08 | **70.11** | 55.26 |
| | MIND | 58.33 | 54.11 | 38.72 | 62.55 | 57.81 | 47.65 | 71.91 | 66.74 | 54.39 | 64.02 | 59.12 | 45.63 | 68.55 | 63.50 | 48.78 |
| | Perplexity | 55.42 | 51.20 | 40.51 | 63.72 | 60.13 | 49.14 | 69.74 | 66.51 | 52.18 | 70.42 | 65.41 | 55.32 | 60.18 | 57.01 | 44.75 |
| | LN-Entropy | 62.17 | 57.52 | 46.44 | 58.33 | 52.99 | 43.28 | 65.30 | 61.27 | 49.92 | 67.15 | 62.42 | 51.33 | 63.28 | 59.07 | 46.14 |
| | Energy | 59.71 | 56.23 | 44.81 | 60.44 | 57.18 | 45.03 | 63.54 | 59.42 | 48.62 | 72.09 | 68.15 | 58.42 | 66.10 | 61.33 | 49.41 |
| | Semantic Ent. | 57.28 | 53.42 | 41.92 | 69.61 | 64.81 | 52.01 | 67.10 | 62.44 | 50.66 | 66.12 | 62.15 | 49.31 | 64.55 | 60.18 | 47.75 |
| | Lexical Sim. | 61.41 | 57.09 | 45.03 | 65.81 | 61.44 | 49.51 | 62.50 | 59.12 | 50.92 | 70.91 | 67.53 | 55.21 | 66.29 | 59.88 | 51.03 |
| | SelfCheckGPT | 56.22 | 52.84 | 40.63 | 60.79 | 55.68 | 45.72 | 64.83 | 56.88 | 48.33 | 66.54 | 62.92 | 51.41 | 68.21 | 65.12 | 53.60 |
| | RACE | 60.12 | 56.50 | 44.90 | 64.12 | 59.77 | 49.22 | 65.44 | 61.55 | 52.73 | 69.61 | 66.31 | 53.92 | 62.55 | 59.42 | 45.66 |
| | P(true) | 58.91 | 55.47 | 42.13 | 67.44 | 63.20 | 51.43 | 71.22 | 66.91 | 54.10 | 63.44 | 60.33 | 49.27 | 70.18 | 65.77 | 52.78 |
| | FActScore | 62.10 | 58.21 | 46.33 | 59.22 | 54.14 | 44.32 | 63.87 | 60.77 | 47.98 | 68.33 | 64.02 | 53.41 | 65.92 | 61.37 | 49.84 |
| **BBH** | **HALLUGUARD** | **68.33** | **64.11** | **56.42** | **74.91** | **69.14** | **62.10** | **73.22** | **69.88** | **57.21** | 78.55 | 69.91 | 61.45 | **71.10** | **68.25** | **59.92** |
| | Inside | 65.41 | 61.22 | 52.83 | 71.02 | 67.10 | 60.21 | 68.17 | 64.75 | 53.92 | 79.17 | 72.33 | 64.22 | 67.10 | 63.52 | 55.91 |
| | MIND | 54.12 | 50.22 | 40.11 | 57.21 | 53.44 | 41.52 | 63.92 | 59.88 | 47.01 | 61.55 | 57.14 | 48.83 | 65.11 | 60.22 | 49.52 |
| | Perplexity | 52.91 | 49.33 | 40.44 | 61.88 | 58.12 | 49.22 | 62.91 | 59.42 | 50.11 | 59.91 | 55.72 | 49.03 | 60.88 | 57.41 | 48.62 |
| | LN-Entropy | 59.12 | 55.44 | 44.92 | 54.61 | 51.75 | 43.18 | 66.44 | 63.21 | 54.09 | 62.75 | 59.12 | 47.52 | 68.20 | 64.88 | 55.41 |
| | Energy | 53.94 | 51.22 | 45.03 | 56.12 | 52.14 | 44.61 | 64.55 | 60.11 | 49.99 | 68.21 | 65.12 | 52.84 | 66.41 | 62.77 | 50.22 |
| | Semantic Ent. | 57.41 | 54.32 | 47.21 | 61.22 | 58.42 | 49.74 | 63.21 | 59.10 | 48.62 | 63.55 | 60.24 | 48.88 | 64.91 | 61.44 | 50.72 |
| | Lexical Sim. | 50.41 | 46.77 | 38.92 | 60.71 | 57.11 | 45.55 | 59.42 | 56.88 | 48.91 | 70.33 | 67.10 | 55.32 | 58.33 | 55.42 | 47.41 |
| | SelfCheckGPT | 55.21 | 52.14 | 43.92 | 58.10 | 55.78 | 46.22 | 62.82 | 59.90 | 50.44 | 65.22 | 62.44 | 54.21 | 63.44 | 60.77 | 52.33 |
| | RACE | 56.14 | 53.72 | 43.88 | 63.11 | 59.71 | 52.81 | 65.77 | 62.55 | 50.72 | 58.88 | 55.14 | 46.18 | 66.10 | 62.41 | 49.81 |
| | P(true) | 54.31 | 52.22 | 44.10 | 58.22 | 56.10 | 48.52 | 56.91 | 53.55 | 43.92 | 61.40 | 58.21 | 46.77 | 57.33 | 54.88 | 45.91 |
| | FActScore | 56.20 | 52.42 | 41.77 | 55.44 | 52.12 | 41.14 | 61.62 | 58.22 | 51.33 | 59.33 | 56.42 | 49.14 | 63.44 | 60.22 | 52.44 |
| **TruthfulQA** | **HALLUGUARD** | **75.11** | **71.20** | **63.21** | **67.44** | **64.55** | **58.12** | **78.92** | **74.22** | **65.33** | **76.44** | **72.01** | **59.92** | **75.33** | **69.11** | **63.08** |
| | Inside | 71.10 | 68.55 | 60.77 | 61.77 | 59.44 | 50.10 | 63.88 | 61.33 | 53.41 | 69.22 | 65.10 | 55.14 | 62.14 | 59.94 | 52.80 |
| | MIND | 57.44 | 54.91 | 45.33 | 59.92 | 56.88 | 48.33 | 58.72 | 56.14 | 47.21 | 61.21 | 58.88 | 52.02 | 60.44 | 58.20 | 49.03 |
| | Perplexity | 49.52 | 46.71 | 38.84 | 54.12 | 51.74 | 43.90 | 59.72 | 57.55 | 46.88 | 54.44 | 51.72 | 42.55 | 60.33 | 57.21 | 47.41 |
| | LN-Entropy | 57.11 | 54.88 | 42.98 | 55.33 | 52.41 | 45.91 | 59.66 | 56.22 | 43.10 | 60.44 | 58.02 | 46.22 | 61.41 | 57.17 | 43.88 |
| | Energy | 54.11 | 52.17 | 38.91 | 53.44 | 51.14 | 36.88 | 58.21 | 54.77 | 49.92 | 63.02 | 60.44 | 51.33 | 58.41 | 55.33 | 50.42 |
| | Semantic Ent. | 60.08 | 56.44 | 44.15 | 50.14 | 47.33 | 35.92 | 53.74 | 52.11 | 37.02 | 65.33 | 63.20 | 50.77 | 55.02 | 53.11 | 38.44 |
| | Lexical Sim. | 51.22 | 49.20 | 39.03 | 58.72 | 54.71 | 48.77 | 65.71 | 63.50 | 53.10 | 55.42 | 54.44 | 40.77 | 61.72 | 59.51 | 44.10 |
| | SelfCheckGPT | 55.72 | 53.44 | 42.78 | 58.33 | 55.72 | 47.14 | 60.88 | 57.44 | 43.91 | 55.42 | 54.44 | 40.77 | 61.72 | 59.51 | 44.10 |
| | RACE | 52.22 | 49.88 | 41.44 | 63.14 | 66.88 | 54.05 | 70.55 | 67.11 | 59.77 | 55.44 | 52.11 | 45.33 | 71.33 | 68.22 | 60.02 |
| | P(true) | 55.54 | 52.11 | 38.82 | 55.72 | 52.33 | 39.22 | 57.41 | 53.10 | 41.22 | 56.88 | 54.77 | 45.55 | 57.12 | 53.33 | 41.88 |
| | FActScore | 52.91 | 50.14 | 40.44 | 54.11 | 50.22 | 41.33 | 52.88 | 49.91 | 42.55 | 61.55 | 59.22 | 44.72 | 53.41 | 50.71 | 43.10 |

Table 9: Comparison with SAPLMA, LLM-Check and ITI across benchmarks and backbones.

| Benchmark | Method | GPT2 | | | | | OPT-6.7B | | | | | Mistral-7B | | | | | QwQ-32B | | | | | LLaMA2-13B | | | | |
|---|---|---|---|---|---|---|---|---|---|---|---|---|---|---|---|---|---|---|---|---|---|---|---|---|---|---|
| | | AUROC | AUPRC | F1 | TPR@10% | TPR@5% | AUROC | AUPRC | F1 | TPR@10% | TPR@5% | AUROC | AUPRC | F1 | TPR@10% | TPR@5% | AUROC | AUPRC | F1 | TPR@10% | TPR@5% | AUROC | AUPRC | F1 | TPR@10% | TPR@5% |
| RAGTruth | HALLUGUARD | 75.51 | 73.40 | 81.22 | 71.62 | 57.80 | 76.77 | 74.20 | 77.03 | 73.52 | 59.12 | 82.31 | 80.79 | 83.19 | 79.44 | 69.21 | 84.59 | 81.15 | 85.91 | 80.13 | 63.52 | 77.51 | 75.30 | 74.66 | 68.91 | 57.42 |
| | SAPLMA | 72.80 | 70.10 | 72.20 | 63.50 | 55.10 | 78.90 | 74.20 | 74.10 | 68.00 | 58.20 | 79.40 | 77.30 | 79.00 | 72.10 | 60.50 | 81.00 | 78.20 | 79.44 | 72.80 | 61.30 | 74.20 | 72.10 | 70.50 | 61.80 | 55.90 |
| | LLM-Check | 68.10 | 64.50 | 63.90 | 55.20 | 44.80 | 72.30 | 68.40 | 66.50 | 57.90 | 46.30 | 76.10 | 73.20 | 68.90 | 61.30 | 49.50 | 76.10 | 73.20 | 68.90 | 61.30 | 49.50 | 65.40 | 61.50 | 55.40 | 46.10 | 47.80 |
| | ITI | 69.30 | 65.80 | 66.10 | 57.90 | 47.90 | 73.10 | 69.20 | 68.20 | 59.80 | 49.10 | 76.00 | 72.30 | 74.10 | 67.20 | 50.90 | 77.20 | 74.10 | 70.50 | 62.40 | 51.70 | 72.80 | 70.30 | 65.40 | 57.10 | 47.80 |
| GSM8K | HALLUGUARD | 72.04 | 69.88 | 78.33 | 74.11 | 65.42 | 72.57 | 70.31 | 74.91 | 69.14 | 62.10 | 80.62 | 77.30 | 80.22 | 76.88 | 68.21 | 75.81 | 74.68 | 82.55 | 78.91 | 70.45 | 79.01 | 76.73 | 79.10 | 74.25 | 67.92 |
| | SAPLMA | 69.20 | 66.10 | 70.10 | 62.00 | 54.40 | 70.80 | 67.20 | 71.80 | 64.10 | 56.30 | 77.10 | 74.00 | 76.20 | 69.50 | 59.80 | 73.90 | 71.20 | 76.50 | 70.10 | 60.90 | 75.40 | 72.30 | 74.00 | 67.10 | 59.10 |
| | LLM-Check | 65.40 | 61.50 | 62.40 | 54.10 | 46.20 | 68.40 | 64.30 | 67.50 | 59.10 | 49.80 | 73.40 | 69.80 | 64.90 | 57.90 | 48.30 | 71.20 | 67.90 | 67.80 | 59.40 | 50.40 | 72.00 | 68.50 | 64.20 | 56.60 | 48.00 |
| | ITI | 66.80 | 63.00 | 64.50 | 56.20 | 48.70 | 69.00 | 65.40 | 69.20 | 61.50 | 51.90 | 74.20 | 70.60 | 67.10 | 60.80 | 50.10 | 72.50 | 69.20 | 69.40 | 62.50 | 52.30 | 73.00 | 69.10 | 66.10 | 58.40 | 49.50 |
| HaluEval | HALLUGUARD | 70.42 | 67.71 | 75.11 | 71.20 | 63.21 | 67.88 | 70.44 | 67.55 | 58.12 | 53.20 | 74.91 | 72.74 | 78.92 | 74.22 | 65.33 | 73.93 | 70.87 | 76.44 | 72.01 | 59.92 | 78.15 | 74.15 | 79.33 | 75.11 | 66.08 |
| | SAPLMA | 67.10 | 63.20 | 69.20 | 62.10 | 54.00 | 69.50 | 65.70 | 68.30 | 61.60 | 53.20 | 72.00 | 68.40 | 75.10 | 69.30 | 58.90 | 71.20 | 68.10 | 75.40 | 70.30 | 58.50 | 76.10 | 72.20 | 76.80 | 70.60 | 60.90 |
| | LLM-Check | 63.50 | 59.40 | 61.10 | 53.00 | 44.50 | 66.80 | 62.90 | 65.40 | 57.50 | 47.90 | 70.10 | 66.30 | 63.80 | 57.20 | 47.10 | 69.30 | 65.40 | 66.20 | 59.50 | 49.00 | 71.50 | 67.60 | 63.50 | 55.90 | 47.40 |
| | ITI | 64.80 | 60.70 | 63.40 | 55.20 | 46.80 | 67.40 | 63.50 | 66.90 | 58.60 | 49.40 | 71.00 | 67.20 | 66.10 | 59.10 | 48.60 | 70.20 | 66.30 | 68.10 | 61.10 | 50.60 | 72.30 | 68.20 | 65.20 | 57.90 | 48.70 |

Figure 11 reports the scatter plot comparing the NTK proxy against empirical $\|J_t\|_2$ across all prompts and steps.

**Validation of Term Decomposition**  To validate the architectural premise of our Hallucination Risk Bound Section 3.2, we visualize the evolution of the decomposed risk components across reasoning depth $T$ on the Snowballing dataset (Zhang et al., 2023). As shown in Figure Figure 12, the total risk is driven by two distinct dynamic behaviors. The data-driven term (green dotted line) exhibits linear or near-constant progression, reflecting static retrieval or knowledge-encoding errors that persist regardless of depth. In contrast, the reasoning-driven term (purple dotted line) demonstrates exponential amplification consistent with the Snowballing Effect, remaining negligible at shallow depths but rapidly dominating the total risk as $T$ increases. Crucially, this reveals a phase transition in hallucination dynamics: at lower depths ($T < 15$), errors are primarily data-driven, whereas at higher depths, reasoning instability becomes the governing factor. This dichotomy empirically justifies our hybrid scoring mechanism, confirming that a unified detector must account

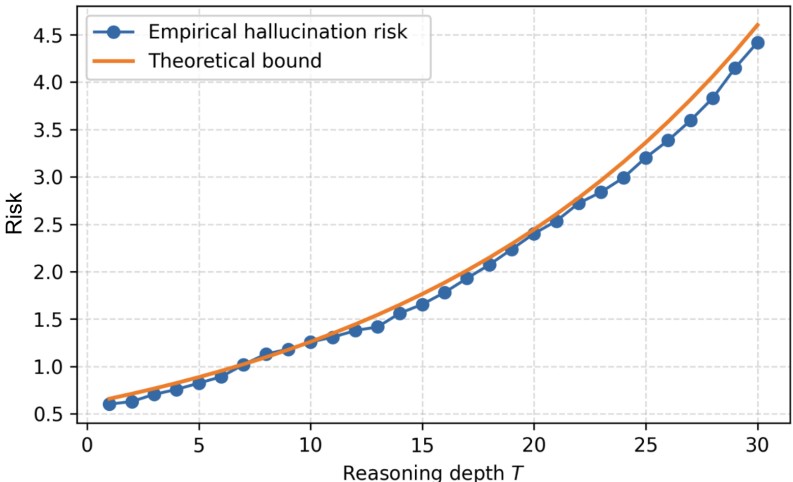

Figure 10: Empirical hallucination risk versus our theoretical bound

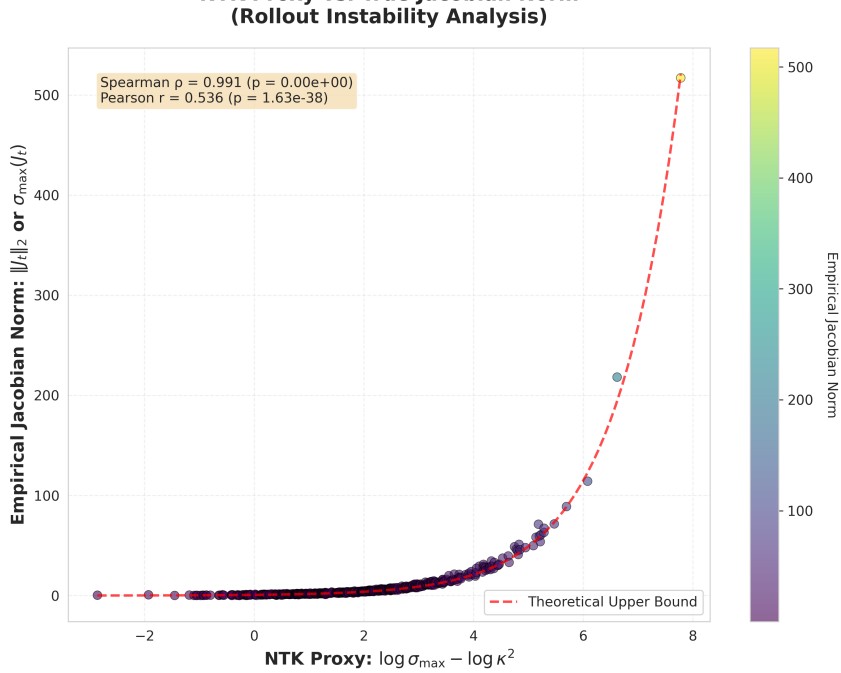

Figure 11: The NTK proxy closely tracks empirical Jacobian amplification on GPT-2-small, showing near-perfect monotonic alignment and a consistent conservative envelope across decoding depth.

for both the static semantic bias and the dynamic rollout instability to be effective across varying generation lengths.

## C.7 Correlation of reasoning-driven and data-driven terms with different types of datasets

To empirically verify the independence of the proposed risk components, we analyzed their correlation with detection performance across distinct task families. As illustrated in Figure 14 and Figure 13, we observe a sharp geometric decoupling: the data-driven term aligns strongly with data-

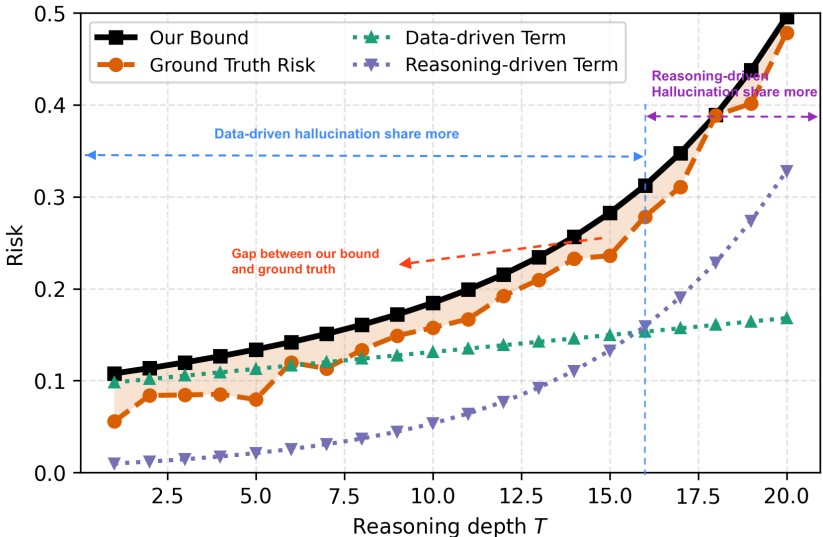

Figure 12: Risk decomposition across reasoning depth T on Snowballing dataset.

centric benchmarks (e.g., RAGTruth) while showing negligible correlation with reasoning tasks. Conversely, the reasoning-driven term dominates on reasoning-oriented datasets (e.g., MATH-500). This double dissociation reinforces the structural validity and orthogonality of our decomposition, confirming that each term captures a distinct, non-redundant failure mode.

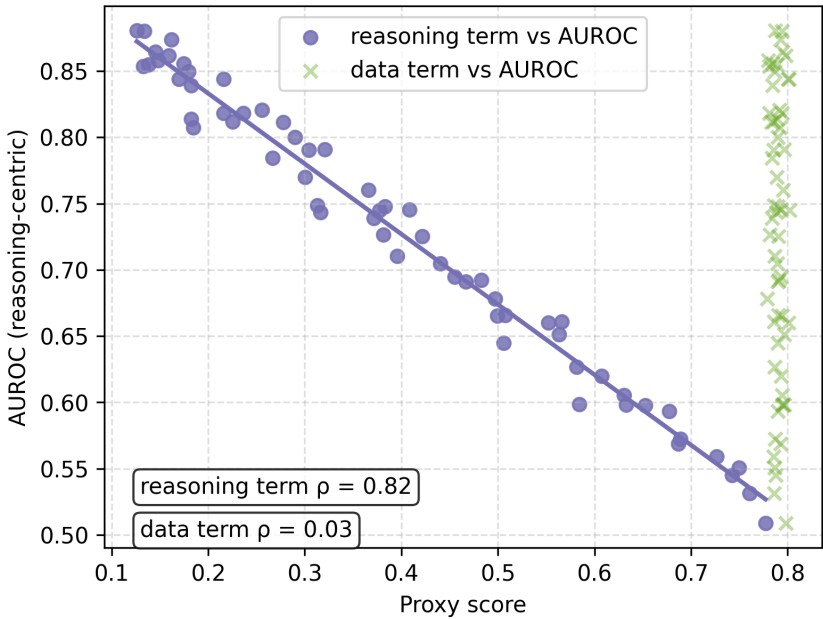

Figure 13: Correlation Between data-driven and reasoning-driven terms and AUROC on Reasoning-Centric MATH500.

## C.8 CASE STUDY

**Case Study 1 - GSM8K (Multi-step Arithmetic): Bias → Drift → Snowballing.** *Task:* "John saves $3/day for four weeks and buys a $12 toy. How much money does he have left?" *Ground truth:* $72.

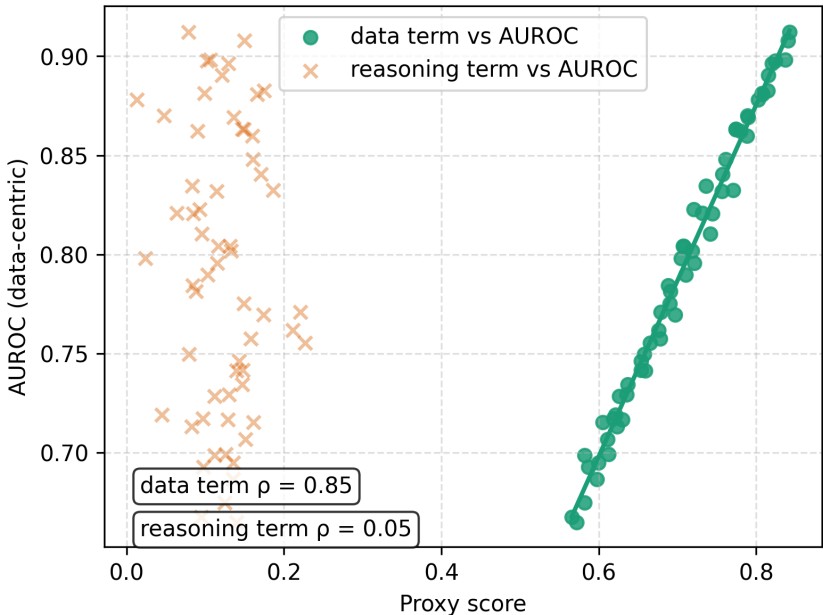

Figure 14: Correlation Between data-driven and reasoning-driven terms and AUROC on Data-Centric RAGTruth.

| Length (T) | Model Behavior | HalluGuard Response |
|---|---|---|
| **T=1–8** Stable setup | Correct restatement and arithmetic planning | Data-driven term dominant; risk flat |
| **T=9–14** Seed error | "4 weeks" → **"40 days"** | Slight rise in data-driven signal |
| **T=15–22** Propagation | "3 × 40 = 120" | Reasoning-driven share begins to rise |
| **T=23–40** Amplification | Final answer: **$108** | Reasoning-driven dominates (snowballing) |

Table 10: Evolution of hallucination in GSM8K arithmetic reasoning.

**Case Study 2 - Long-Document Summarization: Misalignment → Overreach → Fabrication.**
*Task:* Summarize a 5,000-token policy document
*Ground truth:* Security audit exception applies only to specific log types.

| Length (T) | Model Behavior | HalluGuard Response |
|---|---|---|
| **T=1–20** Accurate extraction | Correct recovery of retention rules | Low risk; strong alignment |
| **T=21–40** Misbinding | Incorrect merge of distant sections | Data-driven signal increases |
| **T=41–95** Drift | Overgeneralized suspension claim | Reasoning-driven share rises |
| **T=96–170** Fabrication | New false rule introduced | Reasoning-driven dominates |

Table 11: Evolution of hallucination in long-document summarization.

## C.9    COMPARISON WITH INSIDE AND MIND

Inside and MIND serve as empirical uncertainty diagnostics. Inside analyzes covariance spectra of static representations, while MIND measures temporal variations in hidden states. Both methods extract post-hoc signals and produce a single uncertainty score.

In contrast, HALLUGUARD derives a structured risk decomposition from generative dynamics, separating data-driven and reasoning-driven sources via NTK spectral geometry and instability amplification. This formulation explicitly models compounded reasoning errors.

We evaluate all methods on the Snowballing benchmark Zhang et al. (2023), which emphasizes progressive reasoning instability. As shown in Table 12, HALLUGUARD consistently outperforms Inside and MIND across all backbone models.

Table 12: Comparison with Inside and MIND on the Snowballing benchmark across different backbone models (AUROC / AUPRC).

| Method | GPT-2 | OPT-6.7B | Mistral-7B | QwQ-32B | LLaMA2-7B | LLaMA2-70B |
|---|---|---|---|---|---|---|
| HalluGuard | **88.52/82.14** | **92.63/87.42** | **94.87/89.66** | **97.41/95.08** | **93.28/88.03** | **97.96/95.37** |
| Inside | 74.11/66.39 | 78.24/70.51 | 83.32/75.80 | 87.55/80.47 | 81.72/73.11 | 89.03/82.77 |
| MIND | 69.42/58.73 | 74.56/64.37 | 78.67/68.52 | 84.03/73.68 | 77.91/65.89 | 86.28/78.41 |

## C.10 ADDITIONAL EVALUATION ON MULTIMODAL AND LONG-CONTEXT REGIMES

To evaluate generalization beyond short-form reasoning tasks, we extend experiments to (i) multimodal hallucination detection on POPE Li et al. (2023b), and (ii) long-context generation on GovReport Huang et al. (2021) and NarrativeQA s Koˇ ciský et al. (2018).

Across all backbone models, HALLUGUARD consistently achieves the strongest AUROC and AUPRC, demonstrating robustness under multimodal noise and long-range dependency drift.

Table 13: Comparison of methods across different backbone models on POPE(AUROC/AUPRC).

| Method | GPT-2 | OPT-6.7B | Mistral-7B | QwQ-32B | LLaMA2-7B | LLaMA2-70B |
|---|---|---|---|---|---|---|
| Perplexity | 61.12/53.04 | 68.27/60.18 | 72.41/64.09 | 79.36/73.22 | 70.15/62.31 | 83.48/76.19 |
| HalluGuard | **74.33/68.27** | **81.22/75.36** | **86.47/80.51** | **91.58/86.42** | **85.39/78.44** | **94.63/89.27** |
| Inside | 70.08/64.12 | 77.19/70.33 | 83.44/75.28 | 89.27/82.36 | 81.22/74.41 | 92.51/87.39 |
| MIND | 66.17/58.22 | 73.31/66.14 | 79.28/71.39 | 86.44/79.33 | 77.18/69.27 | 89.36/83.48 |
| LN-Entropy | 63.09/55.11 | 71.24/62.18 | 76.37/67.06 | 84.33/75.29 | 74.12/65.18 | 87.42/80.33 |
| Energy | 62.14/54.22 | 69.17/61.26 | 75.29/66.31 | 83.41/74.18 | 73.21/64.33 | 86.39/79.41 |
| Semantic Ent. | 64.18/56.04 | 72.29/63.14 | 77.41/68.22 | 85.48/76.39 | 75.17/66.41 | 88.46/81.27 |
| Lexical Sim. | 65.24/57.19 | 73.33/64.21 | 78.46/69.37 | 85.52/77.44 | 76.31/67.29 | 88.59/82.31 |
| SelfCheckGPT | 58.11/50.28 | 63.22/55.31 | 67.38/58.24 | 74.41/66.33 | 64.27/56.21 | 78.46/70.39 |
| RACE | 69.14/63.17 | 76.28/69.41 | 82.33/74.29 | 88.47/80.36 | 80.36/73.22 | 91.44/85.33 |
| P(true) | 67.22/59.26 | 74.31/66.18 | 80.41/71.33 | 87.44/79.28 | 78.29/69.33 | 90.38/83.41 |
| FActScore | 68.19/61.33 | 75.39/68.22 | 81.47/73.38 | 88.52/81.41 | 79.34/71.48 | 91.46/85.37 |

**Multimodal Hallucination (POPE).**

Table 14: Comparison of methods across different backbone models on GovReport(AUROC/AUPRC).

| Method | GPT-2 | OPT-6.7B | Mistral-7B | QwQ-32B | LLaMA2-7B | LLaMA2-70B |
|---|---|---|---|---|---|---|
| Perplexity | 58.13/49.22 | 64.41/55.37 | 67.29/58.46 | 75.34/66.18 | 63.28/54.33 | 78.57/69.41 |
| HalluGuard | **72.38/66.41** | **79.27/72.39** | **84.46/78.31** | **90.58/84.42** | **82.44/76.33** | **93.62/88.51** |
| Inside | 69.17/62.24 | 76.33/68.41 | 81.44/73.36 | 88.42/80.31 | 79.36/71.29 | 91.47/85.39 |
| MIND | 65.21/56.18 | 72.41/63.29 | 77.38/68.33 | 86.33/77.41 | 75.27/66.38 | 88.46/82.24 |
| LN-Entropy | 63.12/54.27 | 70.33/61.22 | 75.41/65.34 | 83.39/74.21 | 72.18/62.33 | 86.38/79.33 |
| Energy | 61.09/52.14 | 69.18/60.41 | 74.37/64.28 | 82.34/73.19 | 71.26/61.44 | 85.41/78.36 |
| Semantic Ent. | 64.17/55.11 | 71.22/62.31 | 76.39/67.42 | 84.46/75.38 | 73.24/64.28 | 87.49/80.36 |
| Lexical Sim. | 65.26/56.17 | 72.38/63.38 | 76.44/68.41 | 85.43/76.37 | 74.22/65.19 | 87.53/81.44 |
| SelfCheckGPT | 55.14/46.29 | 60.31/51.22 | 63.44/54.19 | 70.26/60.41 | 59.33/49.24 | 73.41/63.38 |
| RACE | 68.28/60.33 | 75.41/66.29 | 80.36/72.41 | 87.42/79.33 | 78.32/70.24 | 90.38/84.41 |
| P(true) | 66.34/57.22 | 73.39/64.31 | 78.48/69.44 | 86.38/77.41 | 76.33/67.28 | 89.44/83.36 |
| FActScore | 67.41/59.36 | 74.42/66.41 | 79.39/71.46 | 87.41/78.47 | 77.47/69.44 | 90.41/84.38 |

**Long-Context Generation (GovReport).**

Table 15: Comparison of methods across different backbone models on NarrativeQA(AUROC/AUPRC).

| Method | GPT-2 | OPT-6.7B | Mistral-7B | QwQ-32B | LLaMA2-7B | LLaMA2-70B |
|---|---|---|---|---|---|---|
| Perplexity | 56.14/47.22 | 62.33/53.18 | 65.41/55.39 | 72.26/63.41 | 61.27/51.33 | 76.38/67.29 |
| HalluGuard | **70.36/64.41** | **77.22/70.37** | **83.48/76.29** | **89.53/83.47** | **81.33/74.36** | **92.57/87.41** |
| Inside | 67.18/60.27 | 74.39/66.41 | 80.46/72.31 | 87.44/79.36 | 78.41/69.38 | 90.43/84.32 |
| MIND | 63.27/54.18 | 70.31/61.29 | 76.33/67.24 | 84.39/75.41 | 74.36/64.47 | 87.41/80.32 |
| LN-Entropy | 61.19/52.11 | 68.27/59.33 | 73.42/63.21 | 82.41/73.29 | 72.14/61.41 | 85.36/78.44 |
| Energy | 60.08/51.14 | 67.18/58.34 | 72.37/62.47 | 81.33/72.41 | 70.27/60.33 | 84.44/77.46 |
| Semantic Ent. | 63.22/55.09 | 69.31/61.46 | 75.44/66.33 | 83.47/74.41 | 73.26/63.44 | 86.47/79.39 |
| Lexical Sim. | 64.17/56.22 | 70.37/62.34 | 76.41/67.41 | 84.33/75.44 | 74.41/65.27 | 87.46/80.41 |
| SelfCheckGPT | 52.14/43.29 | 57.33/48.31 | 61.48/51.36 | 68.41/58.47 | 56.39/46.31 | 71.36/61.44 |
| RACE | 66.29/58.31 | 73.42/65.38 | 79.33/71.28 | 86.41/78.44 | 77.28/68.39 | 89.43/83.38 |
| P(true) | 64.31/56.24 | 71.39/63.33 | 77.47/68.36 | 85.38/77.41 | 75.29/66.33 | 88.38/82.44 |
| FActScore | 65.44/57.36 | 72.41/64.41 | 78.52/70.38 | 86.44/78.33 | 76.41/68.44 | 89.44/83.39 |

**Long-Context Generation (NarrativeQA).**

## D USAGE OF LLM

Large language models (LLMs) were employed in a limited and transparent manner during the preparation of this manuscript. Specifically, LLMs were used to assist with linguistic refinement, style adjustments, and minor text editing to improve clarity and readability. They were not involved in formulating the research questions, designing the theoretical framework, conducting experiments, or interpreting results. All scientific contributions-including conceptual development, methodology, analyses, and conclusions-are the sole responsibility of the authors.

