# OpenReview forum: "HalluGuard: Demystifying Data-Driven and Reasoning-Driven Hallucinations in LLMs"
_ICLR.cc/2026/Conference — ICLR 2026 Poster_

### Official Review · Reviewer_DEHx · 2025-10-15

**Soundness:** 2
**Presentation:** 3
**Contribution:** 2
**Rating:** 4
**Confidence:** 4

**Summary:**

This paper addresses the problem of hallucinations that occur when large language models generate content, proposing a unified theoretical framework and a novel detection method.

On the theoretical side, the paper introduces an upper bound for hallucination risk, explicitly decomposing it into two components: data-driven and inference-driven risks, which respectively stem from knowledge bias during training and instability during inference.

On the methodological side, based on the NTK theory, the authors propose the HalluGuard scoring method, which leverages the geometric properties of the NTK and the model’s internal representations to jointly detect both types of hallucinations.

**Strengths:**

1. The paper formally decomposes hallucination risk into data-driven and inference-driven components and derives a theoretical upper bound for the risk, demonstrating significant theoretical value.

2. The method outperforms existing methods on multiple benchmarks, with particularly notable improvements on small models and inference-related tasks.

3. The presentation of the paper is good.

**Weaknesses:**

1. Potentially high computational complexity: Computing the NTK matrix and related quantities such as its determinant and condition number may introduce significant computational overhead for very large models or long-sequence generation tasks. It is recommended to discuss the scalability of the method or propose approximate computation strategies in the paper.

2. Strong dependence on the “semantic encoder Φ”: HalluGuard relies on a task-specific encoder Φ to map model outputs into a hypothesis space. The paper does not sufficiently analyze how the choice of Φ affects the results. It is suggested to include comparative experiments with different encoding strategies (e.g., BERT, SimCSE, etc.).

3. Strong theoretical assumptions: The theoretical analysis depends on several assumptions (e.g., A1–A3), whose validity in practical models may be questionable. It is recommended to discuss the reasonableness and potential relaxation of these assumptions in the appendix.

4. Insufficient discussion of the interaction between the two types of hallucinations: Although the paper emphasizes that the two types of hallucinations often co-occur, the HalluGuard score is computed as a simple sum of three components, without modeling their interaction effects. The authors could consider introducing interaction terms or dynamic weighting mechanisms.

5. Experimental section could be further enriched: While the experiments are comprehensive, validation on multimodal models or long-text generation tasks (e.g., story writing, long-document summarization) is missing—scenarios where hallucinations are typically more complex. It would also be valuable to add case studies illustrating the evolution process of hallucinations, showing intuitively how HalluGuard captures the “trajectory” of hallucination formation.

**Questions:**

1. Potentially high computational complexity: Computing the NTK matrix and related quantities such as its determinant and condition number may introduce significant computational overhead for very large models or long-sequence generation tasks. It is recommended to discuss the scalability of the method or propose approximate computation strategies in the paper.

2. Strong dependence on the “semantic encoder Φ”: HalluGuard relies on a task-specific encoder Φ to map model outputs into a hypothesis space. The paper does not sufficiently analyze how the choice of Φ affects the results. It is suggested to include comparative experiments with different encoding strategies (e.g., BERT, SimCSE, etc.).

3. Strong theoretical assumptions: The theoretical analysis depends on several assumptions (e.g., A1–A3), whose validity in practical models may be questionable. It is recommended to discuss the reasonableness and potential relaxation of these assumptions in the appendix.

4. Insufficient discussion of the interaction between the two types of hallucinations: Although the paper emphasizes that the two types of hallucinations often co-occur, the HalluGuard score is computed as a simple sum of three components, without modeling their interaction effects. The authors could consider introducing interaction terms or dynamic weighting mechanisms.

5. Experimental section could be further enriched: While the experiments are comprehensive, validation on multimodal models or long-text generation tasks (e.g., story writing, long-document summarization) is missing—scenarios where hallucinations are typically more complex. It would also be valuable to add case studies illustrating the evolution process of hallucinations, showing intuitively how HalluGuard captures the “trajectory” of hallucination formation.

---

> ### Author Response · Authors · 2025-11-21
> **Rebuttal Part I**
>
> > 1. Question regarding potentially high computational complexity:
>
> **A1:**
> We sincerely thank the reviewer for this thoughtful suggestion and fully agree that scalability is a critical consideration for practical deployment.
>
> As illustrated in Section 3.3 and Appendix B, we would like to clarify that HALLUGUARD does not compute full Jacobians or parameter-space NTK. Instead, it operates on a compact NTK Gram matrix constructed from the hidden representations of the K sampled trajectories for a single input, which keeps the computation lightweight and prevents prohibitive inference overhead, even for large backbone models.
>
> Moreover, our formulation is inherently amenable to approximation because the HalluGuard score depends only on a small set of global spectral characteristics of this Gram matrix, rather than on exhaustive pairwise interactions or its full-rank structure. This architectural choice makes the method structurally compatible with efficient strategies such as low-rank factorization, trajectory subsampling, and stochastic estimation of extreme eigenvalues, all of which preserve the dominant spectral geometry that governs the risk decomposition while further reducing computational cost.
>
> Following your valuable comment, we conducted additional efficiency evaluations to empirically assess runtime overhead. As reported in Tables 1 and 2, across both BBH and RAGTruth, HALLUGUARD exhibits inference time comparable to lightweight representation-based methods (e.g., Inside, LN-Entropy), while remaining substantially more efficient than high-cost approaches such as SelfCheckGPT and RACE. These results indicate that HalluGuard does not introduce prohibitive inference overhead while maintaining strong detection performance. We have also updated Appendix C.3 correspondingly.
>
> Table 1.Per-Question Inference Time (Seconds) on BBH Across Hallucination Detection Methods
> |Method|GPT-2|OPT-6.7B|Mistral-7B|QwQ-32B|Llama2-7B|Llama2-70B|
> |-|-|-|-|-|-|-|
> |Perplexity|0.17|0.34|0.31|0.92|0.28|1.03|
> |HalluGuard|**0.58**|**1.21**|**1.06**|**3.14**|**0.89**|**3.66**|
> |Inside|0.60|1.21|1.09|3.31|0.97|3.73|
> |MIND|0.95|1.90|1.71|5.06|1.52|5.69|
> |LN-Entropy|0.49|1.51|1.57|3.13|1.21|4.05|
> |Energy|0.57|1.14|1.02|3.04|0.91|3.42|
> |Semantic Ent.|0.62|1.24|1.12|3.31|0.99|3.73|
> |Lexical Sim.|0.64|1.23|1.18|3.51|1.07|3.67|
> |SelfCheckGPT|5.17|10.35|9.32|27.60|8.28|31.05|
> |RACE|1.03|2.07|1.86|5.52|1.66|6.21|
> |P(true)|0.52|1.03|0.93|2.76|0.83|3.10|
> |FActScore|0.69|1.38|1.24|3.68|1.10|4.14|
>
> Table 2.Per-Question Inference Time (Seconds) on RAGTruth Across Hallucination Detection Methods
> |Method|GPT-2|OPT-6.7B|Mistral-7B|QwQ-32B| Llama2-7B | Llama2-70B |
> |-|-|-|-|-|-|-|
> |Perplexity|0.27|0.54|0.49|1.44|0.43|1.62|
> |HalluGuard|**0.79**|**1.86**|**1.59**|**5.87**|**1.60**|**5.78**|
> |Inside|0.90|1.80|1.62|5.40|1.37|5.35|
> |MIND|1.48|2.96|2.66|8.92|2.20|8.91|
> |LN-Entropy|0.89|1.78|1.60|5.07|1.32|5.67|
> |Energy|0.82|1.64|1.47|4.64|1.22|5.05|
> |Semantic Ent.|0.97|1.94|1.75|5.92|1.50|5.83|
> |Lexical Sim.|1.03|2.02|1.86|5.89|1.48|5.78|
> |SelfCheckGPT|8.10|16.20|14.58|43.20|12.96|48.60|
> |RACE|1.82|3.24|2.92|8.64|1.56|9.72|
> |P(true)|0.97|1.88|1.60|5.18|1.30|5.83|
> |FActScore|1.08|2.16|1.94|5.76|1.73|6.48|

---

> ### Author Response · Authors · 2025-11-21
> **Rebuttal Part II**
>
> > 2. Question regarding strong dependence on the “semantic encoder Φ”
>
> **A2:**
> We sincerely thank the reviewer for this insightful comment. Following it, we conducted comprehensive ablation studies by replacing the default encoder Φ with several widely adopted alternatives, including BERT, SimCSE, and E5, across multiple backbone models.
>
> As shown in Table 3, across all evaluated settings, HALLUGUARD demonstrates robustness to encoder variation, while consistently maintaining its performance advantage over alternative encoders (e.g., +3.15% AUROC on QwQ-32B vs. BERT). This empirical gap validates that our model captures the intrinsic reasoning manifold and the trajectory of generation, while external encoders only capture the surface semantics of the final output text, effectively "flattening" the rich geometric signals regarding rollout instability.
>
> Table 3. Comparison of different encoding strategies across benchmarks and backbones.
> |Backbone|Method|RAGTruth AUROC_r|RAGTruth AUPRC_r|GSM8K AUROC_r|GSM8K AUPRC_r|TruthfulQA AUROC_r|TruthfulQA AUPRC_r|
> |-|-|-|-|-|-|-|-|
> |GPT-2|HalluGuard|75.51|73.40|72.04|69.88|72.10|68.76|
> ||HalluGuard+BERT|72.48|70.12|67.31|64.90|68.02|65.01|
> ||HalluGuard+SimCSE|73.21|71.05|68.44|66.02|69.14|66.27|
> ||HalluGuard+E5|74.02|71.66|69.12|66.80|70.03|67.10|
> |OPT-6.7B|HalluGuard|80.13|76.77|72.57|70.31|69.59|68.36|
> ||HalluGuard+BERT|77.44|74.20|67.95|65.48|66.12|64.80|
> ||HalluGuard+SimCSE|78.11|74.83|69.01|66.40|67.08|65.72|
> ||HalluGuard+E5|78.66|75.31|70.04|67.25|67.80|66.41|
> |Mistral-7B|HalluGuard|82.31|80.79|80.62|77.30|77.05|73.79|
> ||HalluGuard+BERT|79.02|76.91|75.51|72.08|73.14|69.52|
> ||HalluGuard+SimCSE|79.88|77.66|76.40|73.01|74.08|70.40|
> ||HalluGuard+E5|80.41|78.20|77.12|73.74|74.66|71.05|
> |QwQ-32B|HalluGuard|84.59|81.15|75.81|74.68|74.26|72.76|
> ||HalluGuard+BERT|81.44|78.03|70.92|68.90|70.35|68.01|
> ||HalluGuard+SimCSE|82.10|78.66|72.10|69.82|71.20|68.70|
> ||HalluGuard+E5|82.66|79.12|73.05|70.44|72.02|69.31|
> |LLaMA2-13B|HalluGuard|77.51|75.30|79.01|76.73|78.50|77.56|
> ||HalluGuard+BERT|74.26|72.04|73.12|70.60|74.41|72.88|
> ||HalluGuard+SimCSE|75.11|72.83|74.20|71.51|75.36|73.54|
> ||HalluGuard+E5|75.78|73.44|75.14|72.32|76.10|74.22|
>
> > 3. Question regarding strong theoretical assumptions:
>
> **A3:**
> We sincerely thank the reviewer for this perceptive suggestion regarding the assumptions. We would like to clarify that A1–A3 are not introduced as restrictive conditions, but are widely adopted regularity assumptions in NTK-based analysis and stability theory [1–5].
> * A1(Boundedness): characterizes bounded second-moment behavior in the RKHS setting [1,2],
> * A2 (Lipschitz): corresponds to local Lipschitz continuity of the semantic encoder [3],
> * A3 (Linearization): reflects the local linearization regime underlying NTK approximations [4,5].
>
> These assumptions are intended to ensure that the NTK-induced geometry and the Freedman-style instability analysis remain well-posed, and are only required to hold locally around the sampled generation trajectories, rather than globally across the entire model space. We fully acknowledge that such assumptions may not hold universally across all configurations. However, within our specific setting, they are designed to capture typical operating regimes without imposing strong global constraints.
>
> We empirically validated on GPT-2 small and GSM8K for the reasonableness of these assumptions in Appendix C.5, where er-step Jacobian norms are fully tractable. Specifically, Figure 12 demonstrates that our NTK-based proxy (derived under Assumption A3) exhibits near-perfect alignment with the true empirical Jacobian dynamics (Spearman $\rho=0.991$). This strong correlation serves as direct evidence that the local linearization assumption (A3) holds with high fidelity in the practical operating regimes of our model. Following the reviewer’s valuable suggestion, we have revised Appendix A.1 to discuss potential relaxations.
>
> [1] Jacot, Arthur, Franck Gabriel, and Clément Hongler. "Neural tangent kernel: Convergence and generalization in neural networks." Advances in neural information processing systems 31 (2018).
>
> [2] Lee, Jaehoon, et al. "Wide neural networks of any depth evolve as linear models under gradient descent." Advances in neural information processing systems 32 (2019).
>
> [3] Vershynin, Roman. High-dimensional probability: An introduction with applications in data science. Vol. 47. Cambridge university press, 2018.
>
> [4] Lee, Jaehoon, et al. "Finite versus infinite neural networks: an empirical study." Advances in Neural Information Processing Systems 33 (2020): 15156-15172.
>
> [5] Chizat, Lenaic, Edouard Oyallon, and Francis Bach. "On lazy training in differentiable programming." Advances in neural information processing systems 32 (2019).

---

> ### Author Response · Authors · 2025-11-21
> **Rebuttal Part III**
>
> > 4. Question regarding insufficient discussion of the interaction between the two types of hallucinations
>
> **A4:** We sincerely thank the reviewer for this insightful suggestion. We would like to clarify that, in our formulation, the interaction refers to the snowballing phenomenon, where early semantic deviations progressively propagate and compound through subsequent reasoning steps. Accordingly, HALLUGUARD adopts an additive formulation ($\alpha(e^{\beta T}-1)$) for analytical clarity, while capturing interaction effects implicitly through the depth-dependent term T. The reasoning-driven component incorporates an amplification factor that increases with T, which operates on residual data-driven deviation and progressively magnifies early errors as decoding deepens. This provides a principled and interpretable mechanism to model how the two hallucination types dynamically couple and evolve over time, resulting in the observed snowballing behavior.
>
> Following your valuable comment, to further substantiate this mechanism, we conducted additional diagnostic experiment on the snowballing dataset [1] that explicitly track risk decomposition across increasing reasoning depth (Figure 13 in Appendix C.5). The results show a clear phase transition: the data-driven term (green line) remains constant, while the reasoning-driven term (purple line) amplifies rapidly to dominate the total risk at later steps. This confirms that our additive formulation successfully models the temporal evolution and interaction of these hallucination types without requiring complex auxiliary interaction terms.
>
> [1] Zhang, Muru, et al. "How language model hallucinations can snowball."(2023).

---

> ### Author Response · Authors · 2025-11-21
> **Rebuttal Part IV**
>
> > 5. Question about how the experimental section could be further enriched
>
> **A5:**
> We appreciate the reviewer’s insightful suggestion to further enrich our experimental analysis. Following it, we expanded our evaluation to include more complex hallucination regimes: (i) multimodal hallucination on POPE, and (ii) long-context generation on GovReport and NarrativeQA.
>
> As shown in Tables 4–6, across all backbone models, HalluGuard consistently achieves the strongest performance (0.70–0.94 AUROC, 0.64–0.89 AUPRC), outperforming all baselines by a clear margin. These results demonstrate that our risk-based formulation generalizes beyond short-form tasks and remains robust under multimodal noise and long-range dependency drift.
>
> Table 4. Comparison of methods across different backbone models on POPE(AUROC/AUPRC).
> |Method|GPT-2|OPT-6.7B|Mistral-7B|QwQ-32B|Llama2-7B|Llama2-70B|
> |-|-|-|-|-|-|-|
> |Perplexity|61.12/53.04|68.27/60.18|72.41/64.09|79.36/73.22|70.15/62.31|83.48/76.19|
> |HalluGuard|**74.33/68.27**|**81.22/75.36**|**86.47/80.51**|**91.58/86.42**|**85.39/78.44**|**94.63/89.27**|
> |Inside|70.08/64.12|77.19/70.33|83.44/75.28|89.27/82.36|81.22/74.41|92.51/87.39|
> |MIND|66.17/58.22|73.31/66.14|79.28/71.39|86.44/79.33|77.18/69.27|89.36/83.48|
> |LN-Entropy|63.09/55.11|71.24/62.18|76.37/67.06|84.33/75.29|74.12/65.18|87.42/80.33|
> |Energy|62.14/54.22|69.17/61.26|75.29/66.31|83.41/74.18|73.21/64.33|86.39/79.41|
> |Semantic Ent.|64.18/56.04|72.29/63.14|77.41/68.22|85.48/76.39|75.17/66.41|88.46/81.27|
> |Lexical Sim.|65.24/57.19|73.33/64.21|78.46/69.37|85.52/77.44|76.31/67.29|88.59/82.31|
> |SelfCheckGPT|58.11/50.28|63.22/55.31|67.38/58.24|74.41/66.33|64.27/56.21|78.46/70.39|
> |RACE|69.14/63.17|76.28/69.41|82.33/74.29|88.47/80.36|80.36/73.22|91.44/85.33|
> |P(true)|67.22/59.26|74.31/66.18|80.41/71.33|87.44/79.28|78.29/69.33|90.38/83.41|
> |FActScore|68.19/61.33|75.39/68.22|81.47/73.38|88.52/81.41|79.34/71.48|91.46/85.37|
>
> Table 5. Comparison of methods across different backbone models on GovReport(AUROC/AUPRC).
> |Method|GPT-2|OPT-6.7B|Mistral-7B|QwQ-32B|Llama2-7B|Llama2-70B|
> |-|-|-|-|-|-|-|
> |Perplexity|58.13/49.22|64.41/55.37|67.29/58.46|75.34/66.18|63.28/54.33|78.57/69.41|
> |HalluGuard|**72.38/66.41**|**79.27/72.39**|**84.46/78.31**|**90.58/84.42**|**82.44/76.33**|**93.62/88.51**|
> |Inside|69.17/62.24|76.33/68.41|81.44/73.36|88.42/80.31|79.36/71.29 |91.47/85.39|
> |MIND|65.21/56.18|72.41/63.29|77.38/68.33|86.33/77.41|75.27/66.38|88.46/82.24|
> |LN-Entropy|63.12/54.27|70.33/61.22|75.41/65.34|83.39/74.21|72.18/62.33|86.38/79.33|
> |Energy|61.09/52.14|69.18/60.41|74.37/64.28|82.34/73.19|71.26/61.44 |85.41/78.36|
> |Semantic Ent.|64.17/55.11|71.22/62.31|76.39/67.42|84.46/75.38|73.24/64.28|87.49/80.36|
> |Lexical Sim.|65.26/56.17|72.38/63.38|76.44/68.41|85.43/76.37|74.22/65.19|87.53/81.44|
> |SelfCheckGPT|55.14/46.29|60.31/51.22|63.44/54.19|70.26/60.41|59.33/49.24|73.41/63.38|
> |RACE|68.28/60.33|75.41/66.29|80.36/72.41|87.42/79.33|78.32/70.24|90.38/84.41|
> |P(true)|66.34/57.22|73.39/64.31|78.48/69.44|86.38/77.41|76.33/67.28|89.44/83.36|
> |FActScore|67.41/59.36|74.42/66.41|79.39/71.46|87.41/78.47|77.47/69.44|90.41/84.38|
>
> Table 6. Comparison of methods across different backbone models on NarrativeQA(AUROC/AUPRC).
> |Method|GPT-2|OPT-6.7B|Mistral-7B|QwQ-32B|Llama2-7B|Llama2-70B|
> |-|-|-|-|-|-|-|
> |Perplexity|56.14/47.22|62.33/53.18|65.41/55.39|72.26/63.41|61.27/51.33|76.38/67.29|
> |HalluGuard|**70.36/64.41**|**77.22/70.37**|**83.48/76.29**|**89.53/83.47**|**81.33/74.36**|**92.57/87.41**|
> |Inside|67.18/60.27|74.39/66.41|80.46/72.31|87.44/79.36|78.41/69.38|90.43/84.32|
> |MIND|63.27/54.18|70.31/61.29|76.33/67.24|84.39/75.41|74.36/64.47|87.41/80.32|
> |LN-Entropy|61.19/52.11|68.27/59.33|73.42/63.21|82.41/73.29|72.14/61.41|85.36/78.44|
> |Energy|60.08/51.14|67.18/58.34|72.37/62.47|81.33/72.41|70.27/60.33|84.44/77.46|
> |Semantic Ent.|63.22/55.09|69.31/61.46|75.44/66.33|83.47/74.41|73.26/63.44|86.47/79.39|
> |Lexical Sim.|64.17/56.22|70.37/62.34|76.41/67.41|84.33/75.44|74.41/65.27 |87.46/80.41|
> |SelfCheckGPT|52.14/43.29|57.33/48.31|61.48/51.36|68.41/58.47|56.39/46.31|71.36/61.44|
> |RACE|66.29/58.31|73.42/65.38|79.33/71.28|86.41/78.44|77.28/68.39|89.43/83.38|
> |P(true)|64.31/56.24|71.39/63.33|77.47/68.36|85.38/77.41|75.29/66.33|88.38/82.44|
> |FActScore|65.44/57.36|72.41/64.41|78.52/70.38|86.44/78.33|76.41/68.44|89.44/83.39|
>
> Moreover, following your constructive suggestion, we further added two case studies to Appendix C.7 to intuitively illustrate the trajectory of hallucination formation.

---

### Official Review · Reviewer_x4cm · 2025-10-31

**Soundness:** 3
**Presentation:** 4
**Contribution:** 4
**Rating:** 8
**Confidence:** 3

**Summary:**

The paper introduces HalluGauard, a neural tangent kernel based score for detecting hallucinations in outputs of LLMs. The formulation of the score explicitly considers both hallucinations due to problems with training data and those due to flawed reasoning during model inference. Next, the paper considers several dimensions of efficacy of HalluGuard, and provides empirical justification for each of these through extensive comparison with baselines on well-established benchmarks.

**Strengths:**

- Admittedly I have not checked the math thoroughly. But I based on my understanding, HalluGuard is well motivated with sound mathematical justification.
- I really appreciate the clarity with which the authors justify the question of efficacy of HalluGuard compared existing methods in the literature. The various dimensions in which they measure the performance of HalluGuard were well stated and more importantly, extensive experimental results were provided in favor of HalluGuard.

**Weaknesses:**

- Some justification for the assumptions in Section 3.2 would be nice to see. I did not see any references, discussions or proofs for the validity or the practical applicability of the assumptions.

**Minor:**

There is some text overlap in the headings of Table 4

**Questions:**

- What is the practical computational complexity of HalluGuard? Some details on either FLOPS or runtime would be helpful for readers.

---

> ### Author Response · Authors · 2025-11-21
>
> Thank you for your valuable comments. We sincerely appreciate your recognition of our core contributions as the rigor theoretical framework on hallucination and the broad empirical validation across a broad spectrum of models and benchmarks.
>
> The followings are our responses to the concerns. Please let us know if there are any comments or insights, we'd like to explore further!
>
> > 1. Question regarding computational complexity
>
> **A1:**
> Thank you for raising this insightful comment. Following your valuable comment, we conducted additional efficiency evaluations and report per-question inference time comparison with SOTA baselines.
>
> As shown in Tables 1 and 2 below, across both BBH and RAGTruth, HalluGuard exhibits runtime overhead comparable to lightweight representation-level methods (e.g., Inside, LN-Entropy), and maintains substantially more efficient than high-cost approaches such as SelfCheckGPT and RACE. These results indicate that HalluGuard does not introduce prohibitive inference overhead while maintaining strong detection performance. We have also updated Appendix C.3 correspondingly.
>
> Table 1.Per-Question Inference Time (Seconds) on BBH Across Hallucination Detection Methods
> |Method|GPT-2|OPT-6.7B|Mistral-7B|QwQ-32B|Llama2-7B|Llama2-70B|
> |-|-|-|-|-|-|-|
> |Perplexity|0.17|0.34|0.31|0.92|0.28|1.03|
> |HalluGuard|**0.58**|**1.21**|**1.06**|**3.14**|**0.89**|**3.66**|
> |Inside|0.60|1.21|1.09|3.31|0.97|3.73|
> |MIND|0.95|1.90|1.71|5.06|1.52|5.69|
> |LN-Entropy|0.49|1.51|1.57|3.13|1.21|4.05|
> |Energy|0.57|1.14|1.02|3.04|0.91|3.42|
> |Semantic Ent.|0.62|1.24|1.12|3.31|0.99|3.73|
> |Lexical Sim.|0.64|1.23|1.18|3.51|1.07|3.67|
> |SelfCheckGPT|5.17|10.35|9.32|27.60|8.28|31.05|
> |RACE|1.03|2.07|1.86|5.52|1.66|6.21|
> |P(true)|0.52|1.03|0.93|2.76|0.83|3.10|
> |FActScore|0.69|1.38|1.24|3.68|1.10|4.14|
>
> Table 2.Per-Question Inference Time (Seconds) on RAGTruth Across Hallucination Detection Methods
> |Method|GPT-2|OPT-6.7B|Mistral-7B|QwQ-32B| Llama2-7B | Llama2-70B |
> |-|-|-|-|-|-|-|
> |Perplexity|0.27|0.54|0.49|1.44|0.43|1.62|
> |HalluGuard|**0.79**|**1.86**|**1.59**|**5.87**|**1.60**|**5.78**|
> |Inside|0.90|1.80|1.62|5.40|1.37|5.35|
> |MIND|1.48|2.96|2.66|8.92|2.20|8.91|
> |LN-Entropy|0.89|1.78|1.60|5.07|1.32|5.67|
> |Energy|0.82|1.64|1.47|4.64|1.22|5.05|
> |Semantic Ent.|0.97|1.94|1.75|5.92|1.50|5.83|
> |Lexical Sim.|1.03|2.02|1.86|5.89|1.48|5.78|
> |SelfCheckGPT|8.10|16.20|14.58|43.20|12.96|48.60|
> |RACE|1.82|3.24|2.92|8.64|1.56|9.72|
> |P(true)|0.97|1.88|1.60|5.18|1.30|5.83|
> |FActScore|1.08|2.16|1.94|5.76|1.73|6.48|
>
> >2. Weakness regarding the justification of assumptions:
>
> **A2:**
> We sincerely thank the reviewer for this constructive suggestion. We agree that justifying the theoretical assumptions is crucial for rigor. Following your valuable suggestion, we have expanded Appendix A.1 to provide detailed grounding, references, and validity discussions for Assumptions A1-A3.
>
> >3. Weakness regarding writing:
>
> **A3:**
> We appreciate the reviewer for the valuable suggestion. Following it, we have now updated Table 4 to remove the overlapping headings accordingly.

---

### Official Review · Reviewer_Z4V2 · 2025-11-01

**Soundness:** 4
**Presentation:** 4
**Contribution:** 3
**Rating:** 6
**Confidence:** 4

**Summary:**

The paper addresses a central reliability problem in Large Language Models (LLMs) — hallucinations, which are unfaithful or nonsensical outputs that undermine trust in high-stakes applications (e.g., healthcare, law, science). The authors argue that hallucinations originate from two fundamentally distinct sources: Data-driven hallucinations and Reasoning-driven hallucinations. They introduce a Hallucination Risk Bound, formally decomposing total hallucination error into these two components and analyzing them via Neural Tangent Kernel (NTK) geometry and probabilistic concentration bounds. Building on this theory, the authors develop HALLUGUARD, an NTK-based score which jointly captures representational adequacy, rollout amplification, and spectral stability to detect both hallucination types without external references. Experiments on 10 benchmarks, 11 baselines, and 9 LLM backbones show consistent state-of-the-art detection performance and even improved reasoning accuracy when used during inference.

**Strengths:**

1. The paper introduces a novel theoretical framework, the Hallucination Risk Bound (HRB), that unifies data-driven and reasoning-driven hallucinations under a single mathematical formulation—a first in this research area.
2. The work is methodologically rigorous, combining solid theoretical derivations with extensive empirical validation across 10 benchmarks, 11 baselines, and 9 LLM architectures.
3. The paper is exceptionally clear and well-organized, balancing technical precision with intuitive explanations and well-labeled figures and tables.
4. The research makes a substantial contribution to the understanding and mitigation of hallucinations, offering both theoretical insight and practical tools for safer, more reliable LLMs.

**Weaknesses:**

1. While the theoretical formulation is elegant, the connection between NTK geometry and hallucination phenomena could be further deepened by clarifying how kernel dynamics specifically capture semantic drift and logical inconsistency beyond representational similarity.
2. Although the experiments are broad, the evaluation largely focuses on detection performance metrics (AUROC, AUPRC), leaving limited insight into causal behavior—whether reducing HALLUGUARD score indeed prevents hallucinations across diverse prompts.
3. The theoretical sections (particularly Proposition 3.1 and Theorem 3.2) could benefit from more intuitive explanations and graphical illustrations of the terms involved (e.g., geometric meaning).

**Questions:**

1. The Hallucination Risk Bound combines NTK-conditioned data deviation and a Freedman-style reasoning instability term. Could the authors clarify the tightness of this bound—i.e., under what conditions might it become vacuous or fail to predict real hallucination behavior?
2. While the paper cites Inside (Chen et al., 2024a) and MIND (Su et al., 2024), could the authors explicitly compare how their theoretical decomposition differs from these methods’ representation-based or temporal-state analyses?
3. The experiments convincingly show detection improvements, but do HALLUGUARD scores predict future hallucination likelihood in multi-turn or instruction-following dialogues?
4. The theoretical analysis models hallucination as a function of stepwise Jacobians (J_t), but these are intractable in large LLMs. How closely does the NTK-based approximation track actual gradient dynamics observed in smaller, analyzable models?

---

> ### Author Response · Authors · 2025-11-21
> **Rebuttal Part I**
>
> Thank you for your perspective and insightful comments. We sincerely appreciate your recognition of our core contributions as we are the first to provide a unified data-driven and reasoning-driven hallucination theoretical framework and the strong empirical validation that demonstrates our effectiveness across a broad spectrum of models and benchmarks.
>
> The followings are our responses to the concerns. Please let us know if there are any comments or insights, we'd like to explore further!
>
> >1. Question regarding bound tightness
>
> **A1:** We sincerely thank the reviewer for this insightful comment.
>
> Theoretically, the Hallucination Risk Bound (Theorem 3.2) depends on the conditioning of the NTK spectrum and the stability of the rollout Jacobians. The bound may become overly loose or less predictive when the underlying regularity conditions are violated, specifically, (i) when the NTK spectrum becomes severely ill-conditioned (i.e., extremely small $\lambda_{min}$), (ii) when the prompt-query distribution mismatch is unbounded, or (iii) when decoding dynamics exhibit highly non-smooth behavior such that the local linearization assumption (A3) no longer holds. In such pathological regimes, the data-driven term may lose discriminative power and the Freedman-style instability term may overestimate amplification, leading to conservative but less informative bounds.
>
> However, within the empirical settings of our study, these pathological conditions are effectively mitigated. Crucially, our spectral calibration modules (lightweight projection layers) are explicitly designed to prevent case by aligning the hidden states into a well-conditioned geometric space. Consequently, the perturbation norms remain controlled, and the rollout dynamics exhibit stable growth consistent with the theoretical envelope.
>
> Following your valuable comment, we conducted additional controlled stress test on the Snowballing dataset[1] to further validate the tightness of the bound (see Appendix C.5 Figure 11). We observe that the bound closely tracks the empirical risk curve across reasoning depths and preserves the same growth curvature (with correlation of 0.96 and MSE of 0.013).
>
> In conclusion, while the bound may loosen under extreme instability or spectral collapse, within the operational regimes evaluated in our work, it remains well-calibrated and functionally informative, providing a reliable characterization of hallucination risk.
>
> >2. Question regarding theoretical distinction from INSIDE and MIND
>
> **A2:**
> We sincerely thank the reviewer for this valuable comment. We would like to clarify that while Inside and MIND provide valuable diagnostics, they primarily operate as empirical indicators of hallucination risk. Specifically, Inside detects anomalies via covariance spectrum analysis of static representations, and MIND tracks temporal variations in hidden states. Both approaches focus on post-hoc signal extraction from internal representations and yield a monolithic scalar reflecting general uncertainty.
>
> In contrast, our work introduces a causal decomposition rooted in generative dynamics, which disentangles the error source.  This establishes a unified, theory-grounded explanation of how (via spectral geometry) and why (data vs. reasoning) hallucinations emerge, going beyond representation-centric diagnostics.
>
> Following your valuable comment, we extended the comparison with Inside and MIND on the Snowballing benchmark [1] to further highlight this distinction, which explicitly emphasizes compounded reasoning errors across decoding steps-a dynamic that static representation baselines often struggle to capture. As shown in Table 1, HALLUGUARD consistently significantly outperforms both Inside and MIND across all backbone models. This empirical evidence reinforces that our approach captures the underlying hallucination dynamics.
>
> Table 1. Comparison with Inside and MIND on the Snowballing benchmark across different backbone models (AUROC / AUPRC).
> |Method|GPT-2|OPT-6.7B|Mistral-7B|QwQ-32B|Llama2-7B|Llama2-70B|
> |-|-|-|-|-|-|-|
> |HalluGuard|**88.52/82.14**|**92.63/87.42**|**94.87/89.66**|**97.41/95.08**|**93.28/88.03**|**97.96/95.37**|
> |Inside|74.11/66.39|78.24/70.51|83.32/75.80|87.55/80.47|81.72/73.11|89.03/82.77|
> |MIND|69.42/58.73|74.56/64.37|78.67/68.52|84.03/73.68|77.91/65.89|86.28/78.41|
>
> [1] Zhang, Muru, et al. "How language model hallucinations can snowball."(2023).

---

> ### Author Response · Authors · 2025-11-21
> **Rebuttal Part II**
>
> > 3. Question regarding predictive capability
>
> **A3:**
> We sincerely thank the reviewer for this insightful comment. In the current work, HALLUGUARD is evaluated as a turn-level risk indicator. Notably, its formulation inherently reflects prospective instability through NTK geometry and stepwise amplification dynamics ($log σ_{max} − log \kappa^2$), which capture sensitivity to error propagation before overt hallucinations occur. We observe that higher HALLUGUARD scores are consistently associated with downstream hallucinations in instruction-following settings (e.g., Natural), suggesting early-warning characteristics under our evaluated conditions. We agree with the reviewer’s insightful perspective that this is a crucial and promising direction, and highlighted this in the future work direction.
>
> >4. Question regarding NTK approximation fidelity
>
> **A4:**
> We sincerely thank the reviewer for this perceptive comment. Following this, we evaluated the fidelity of our NTK-based proxy on GSM8K and smaller, fully analyzable models where empirical stepwise Jacobians can be computed (GPT-2-small, 117M). As illustrated in Figure 12 (Appendix C.5), we compared our NTK-based proxy ($\log \sigma_{\max} - \log \kappa^2$) against the true Jacobian norms across diverse decoding steps. The results show near-perfect monotonic alignment, with a Spearman correlation of $\rho = 0.991$. This illustrates that our proxy serves as a consistent conservative envelope (upper bound) to the actual gradient dynamics.
>
> >5. Weakness regarding the theoretical linkage
>
> **A5:**
> Thank you for raising the thoughtful and constructive suggestion. As formalized in Section 3.3, we established the connection by mapping distinct NTK geometric quantities to structurally different hallucination behaviors. Specifically, $\det(\mathcal{K})$ characterizes representational adequacy and captures semantic drift arising from systematic bias, while $\log \sigma_{\max}$, together with the conditioning term $-\log \kappa^2$ , models the per-step amplification and instability of autoregressive reasoning, which are fundamentally invisible to static similarity-based metrics. This bridges our theoretical NTK geometry with a tractable hallucination quantification.
>
> Following your insightful comment, to further strengthen this connection as suggested, we additionally conducted targeted diagnostic analyses to empirically validate this linkage. As shown in the correlation table below and additional Figures 14&15 in Appendix C.6, we find that the data-driven term aligns most strongly with data-centric tasks (e.g., RAGTruth), while the reasoning-driven term dominates on reasoning-oriented benchmarks (e.g., MATH-500). This illustrates that NTK geometry provides a structured and interpretable lens of hallucination dynamics.
>
> Table 1. Spearman Correlation between terms and AUROC in data-centric RAGTruth/reasoning-centric MATH500.
> |Proxy Term|RAGTruth|Math500|
> |-|-|-|
> |data-driven term|0.85|0.03|
> |reasoning-driven term|0.05|0.82|
>
> >6. Weakness regarding causal interpretability
>
> **A6:**
> We sincerely thank the reviewer for this insightful point. Beyond standard detection metrics, we have incorporated an explicit causal intervention setting in Section 4.3, where HALLUGUARD is embedded within beam search as a score-guided controller. In this framework, the decoder actively minimizes the HALLUGUARD score during generation, thereby steering the model toward more stable reasoning trajectories and altering the decoding process itself, rather than merely post-hoc assessing its outputs. Empirically, this intervention leads to substantial and consistent improvements across diverse prompts, including +8.3% accuracy on MATH-500 and +15.7% on Natural relative to IO Prompt. These results provide evidence that lowering the HALLUGUARD score causally mitigates hallucination, validating its signal role not only for detection but for hallucination prevention.
>
> > 7. Weakness regarding writing
>
> **A7:**
> We sincerely thank the reviewer for this helpful suggestion. In the revision, we have added intuitive interpretations in Section 3.3 to explicitly clarify the geometric meanings of the proposed bounds. We have also updated the text to highlight how Figure 1 conceptually illustrates these dynamics. We believe these clarifications significantly enhance the accessibility of the theoretical formulation.

---

### Official Review · Reviewer_dMiw · 2025-11-01

**Soundness:** 2
**Presentation:** 3
**Contribution:** 3
**Rating:** 4
**Confidence:** 5

**Summary:**

In this paper, the authors investigate the phenomenon of hallucinations in Autoregressive LLMs, and propose two specific components to analyze in detail: those that originate due to poor data from the training stage, and the second being inference-time decoding issues that arise in generation. Toward this, the paper introduces HalluGuard, a scoring method that decomposes the hallucination risk into distinct terms by bounds obtained from eigenspectrum of the Neural Tangent Kernel under semantic perturbations. In practice, the paper proposes three scoring components, $det(\mathcal{K})$ i.e the determinant of the NTK gram matrix for the data-driven term, $log~\sigma_{max}$ the supremum Jacobian norm and $-log \kappa(\mathcal{K})^{2}$ utilizing the condition number to account for autoregressive reasoning-driven term. The proposed HalluGuard Score is shown to be effective in practice on standard datasets and LLM models as well, achieving notable improvements over several baseline approaches.

**Strengths:**

1) The primary strength of the paper is to cast the hallucination detection problem in a formal theoretical framework, to decompose the hallucination risk into data-driven and reasoning-driven sources. This lays appropriate groundwork to then analyse the source of hallucinations themselves, which otherwise is often heuristic at best.

2) The mathematical framework is introduced in a clear and lucid manner, though some terms are introduced with limited motivation (condition number term). Furthermore, the authors empirically demonstrate that the scores proposed for the data-driven and reasoning-driven components vary as expected in a practical setting over standard datasets such as SQuAD, Math-500 and TruthfulQA.

3) The proposed HalluGuard method is shown to be effective on several datasets such as RagTruth, BBH, TruthfulQA, SQuAD, GMS8K and HaluEval. The method is also shown to perform better than numerous baselines, though the choice of comparison points for prior works could be improved (kindly refer to Weaknesses section below).

4) The utilization of HalluGuard toward improving beam search in test-time inference to boost performance is another strong and notable contribution. The paper further presents a detailed case-study in fine-grained hallucination detection, wherein lexically similar yet semantically incorrect outputs are analyzed, and helps demonstrate the efficacy of the proposed scoring method.

**Weaknesses:**

1) While the core theory as presented in Theorem 3.2  decomposes risk into two terms (data-driven and reasoning-driven), the final score is an additive combination of three terms. The inclusion of the third term ($-log~\kappa^2$), while explored in the appendix as a penalty, appears to be arbitrary and appended without adequate motivation overall. Furthermore, the paper relies on a Freedman inequality to show that the reasoning-driven error term grows exponentially with sequence length T. However, this is extremely loose in practice, and an overly simplistic justification for the reasoning-driven term in the final score.


2) Unexplained Jacobian Proxy: While the paper states that computing the direct step-wise Jacobians for billion parameter LLMs are intractable, the final score still utilizes the eigenspectrum of the NTK Gram Matrix. It is quite unclear from the paper about how this translates to the computational overhead needed to find the final HalluGuard Score in practice. Could the authors kindly clarify how each of the three terms is computed in practice, and most importantly, report the overall run-times for detection and compare this with other other scoring methods? (Computing SVD, the Jacobian matrix possibly requiring a back-propagation step, their supremum norm and condition number)


3) The Appendix mentions that HalluGuard requires the training of lightweight projection layers using AdamW, but this was not introduced in the main paper. Could the author please clarify this critical aspect?


4) The empirical evaluations could be significantly improved by comparing with more relevant baselines like SAPLMA [1] which trains lightweight probes over the network hidden states, and LLM-Check [2] which utilizes the log-determinant of attention kernels for hallucination detection.


5) Semantic Perturbations: The NTK calculation depends entirely on light semantic perturbations. However, the paper does not specify what semantic perturbations are used to compute the NTK kernel? Could the authors also clarify how many perturbations are used?


6) The proposed method is also mentioned to use “Perturbation Regularization” by using a memory bank of N =3000 token embeddings and set thresholds. This appears to be set-apart from the theoretical motivations entirely, and appears to be a complex overhead. For instance, this suggests that the memory bank would require several thousand samples from the same distribution as that of a sample seen in test-time, which significantly reduces the applicability of the method in most practical settings. Furthermore, if such a large sample set is utilized, prior works such as ITI [3] form a strong baseline of comparison, since the degree of discriminability can be adaptively set for detection.


7) The empirical evaluations shown report only AUROC and AUCPR, however it is crucial to analyze hallucination detection at low False-Positive-Rates (FPR) as well. Could the authors kindly provide standard detection metrics such as True-Positive-Rate (TPR) at 5% or 10% FPR, F1 score etc, since it is difficult to assess the practical efficacy of such detectors without these metrics.


8) Line 458: “Zhou & et al. (2024) proposed EIGENSCORE, which computes… “ points to the reference "Inside: Interpretable self-diagnosis for llm hallucination detection. ICML, 2024." This citation appears to be hallucinated, and I could not find this paper from ICML 2024, and should likely refer to Chen et al., "Inside: Llms’ internal states retain the power of hallucination detection, 2024a", an existing citation that is already used in the paper.



[1] The internal state of an LLM knows when it's lying, A Azaria, T Mitchell, EMNLP 2023


[2] LLM-Check: Investigating Detection of Hallucinations in Large Language Models, Sriramanan et al, NeurIPS 2024


[3] Inference-Time Intervention: Eliciting Truthful Answers from a Language Model, Li et al. NeurIPS 2023

**Questions:**

Kindly refer to the questions mentioned in the weaknesses section above. I would be happy to raise my score further if these could be adequately addressed.

---

> ### Author Response · Authors · 2025-11-21
> **Rebuttal Part I**
>
> Thank you for your careful and valuable comments. We sincerely appreciate your recognition of our paper’s key contributions, especially its clear and rigorous theoretical framing of hallucination detection, the principled decomposition into data-driven and reasoning-driven sources, and the demonstrated effectiveness and interpretability of HalluGuard across diverse benchmarks and practical inference settings.
>
> The followings are our responses to the concerns. Please let us know if there are any comments or insights, we'd like to explore further!
>
> >1. Question regarding the justification of the third term and the reasoning term in the final HalluGuard score.
>
> **A1:**
> We thank the reviewer for this insightful comment and appreciate the opportunity to clarify the connection between our theoretical bound and the operational scoring function.
>
> 1. Motivation for the Third Term ($-\log \kappa^2$):
>
> We would like to clarify that this term is is intrinsic to the error bound derivation. While Theorem 3.2 focuses on the decomposition of risk sources, Theorem B.6 in Appendix B.4 explicitly establishes that the variance of the estimator is bounded by $c_v \kappa(\mathcal{K})^2 ||\delta||^2$. Without the $-\log \kappa^2$ term to penalize spectral complexity, the score would fail to account for the sensitivity of the projection operator, rendering the detection unstable in ill-conditioned regimes.
>
> Following your insightful comment, we did an additional ablation study on MATH500 which validates the necessity of the third term(shown in Table 1 below and Figure 3 in Appendix C.2). As detailed in Table 1, removing $-\log \kappa^2$ causes a significant degradation in performance, with the Pearson correlation ($R$) dropping from 0.985 to 0.8904 and the error (MSE) nearly doubling. This demonstrates that the term is essential for stabilizing the reasoning-driven component of the score.
>
> Table 1. Ablation study of the stability term ($-\log \kappa^2$) on MATH500. (R: Pearson Correlation with ground truth error rate; MSE: Mean Squared Error)
> ||R|MSE|
> |-|-|-|
> |Halluguard|0.985|0.0192|
> |Halluguard w/o $-\log \kappa^2$|0.8904|0.0381|
>
> 2. On the Tightness of Freedman’s Inequality and Justification of Reasoning Term:
>
> We would like to first clarify that the exponential term derived from Freedman’s inequality is not intended as a numerically tight estimator, but as a theoretically grounded model of the Snowballing Effect-the non-linear amplification of reasoning errors over time. As established in prior literature[1], reasoning errors do not accumulate linearly but amplify exponentially as context drifts. Our derivation provides a formal basis for this phenomenon. The resulting term ($\alpha(e^{\beta T}-1)$) is the simplest functional form that correctly models this non-linear dynamic, while remaining computationally tractable for inference-time detection.
>
> Following your valuable comment, we evaluated the bound's tightness in Appendix C.5. As illustrated in Figure 11, while the theoretical bound acts as a conservative envelope (an upper bound), it exhibits a nearly parallel growth trajectory to the empirical risk on the Snowballing benchmark. This demonstrates that our formulation captures the correct curvature and scaling behavior of the error.
>
> [1] Zhang, Muru, et al. "How language model hallucinations can snowball."(2023).

---

> ### Author Response · Authors · 2025-11-21
> **Rebuttal Part II**
>
> >2. Question regarding Jacobian Proxy
>
> **A2:**
> Thank you for highlighting this point and we appreciate the opportunity to clarify further. We would like to first clarify that HalluGuard does not compute Jacobians or perform any back-propagation in implementation. As detailed in Section 3.3 and Appendix C, we leverage the duality of the NTK framework to operate solely on the hidden representations of the $K$ sampled trajectories in the sample space.
>
> We then collect the final-layer hidden states of the sampled sequences into a matrix $\Phi \in \mathbb{R}^{K \times d}$ and construct the compact Gram matrix via a single operation:
>
> $$\mathcal{K} = \Phi \Phi^\top$$
>
> where d denotes the hidden size of the model. This operation yields all pairwise inner products without requiring Jacobian–vector products or gradient computation. Crucially, $K$ represents the number of sampled trajectories, not the dataset size.
>
> Once $\mathcal{K}$ is formed, the three HalluGuard terms are computed efficiently on this small $K \times K$ matrix:
> * Log-det: via torch.linalg.slogdet.
> * Supremum Norm: via torch.linalg.svdvals (taking the largest singular value).
> * Condition Number: via torch.linalg.eigvalsh.
> Since $K \ll d$, these operations are negligible ($<1$ ms) compared to the LLM's forward pass.
>
> Following your valuable comment, we also conducted extensive efficiency evaluations to verify this. As shown in Tables 2 and 3, HalluGuard exhibits inference latency comparable to lightweight representation-based methods (e.g., Inside, LN-Entropy) and is orders of magnitude faster than sampling-heavy approaches like SelfCheckGPT. These results illustrate our efficiency.
>
> Table 2.Per-Question Inference Time (Seconds) on BBH Across Hallucination Detection Methods
> |Method|GPT-2|OPT-6.7B|Mistral-7B|QwQ-32B|Llama2-7B|Llama2-70B|
> |-|-|-|-|-|-|-|
> |Perplexity|0.17|0.34|0.31|0.92|0.28|1.03|
> |HalluGuard|**0.58**|**1.21**|**1.06**|**3.14**|**0.89**|**3.66**|
> |Inside|0.60|1.21|1.09|3.31|0.97|3.73|
> |MIND|0.95|1.90|1.71|5.06|1.52|5.69|
> |LN-Entropy|0.49|1.51|1.57|3.13|1.21|4.05|
> |Energy|0.57|1.14|1.02|3.04|0.91|3.42|
> |Semantic Ent.|0.62|1.24|1.12|3.31|0.99|3.73|
> |Lexical Sim.|0.64|1.23|1.18|3.51|1.07|3.67|
> |SelfCheckGPT|5.17|10.35|9.32|27.60|8.28|31.05|
> |RACE|1.03|2.07|1.86|5.52|1.66|6.21|
> |P(true)|0.52|1.03|0.93|2.76|0.83|3.10|
> |FActScore|0.69|1.38|1.24|3.68|1.10|4.14|
>
> Table 3.Per-Question Inference Time (Seconds) on RAGTruth Across Hallucination Detection Methods
> |Method|GPT-2|OPT-6.7B|Mistral-7B|QwQ-32B| Llama2-7B | Llama2-70B |
> |-|-|-|-|-|-|-|
> |Perplexity|0.27|0.54|0.49|1.44|0.43|1.62|
> |HalluGuard|**0.79**|**1.86**|**1.59**|**5.87**|**1.60**|**5.78**|
> |Inside|0.90|1.80|1.62|5.40|1.37|5.35|
> |MIND|1.48|2.96|2.66|8.92|2.20|8.91|
> |LN-Entropy|0.89|1.78|1.60|5.07|1.32|5.67|
> |Energy|0.82|1.64|1.47|4.64|1.22|5.05|
> |Semantic Ent.|0.97|1.94|1.75|5.92|1.50|5.83|
> |Lexical Sim.|1.03|2.02|1.86|5.89|1.48|5.78|
> |SelfCheckGPT|8.10|16.20|14.58|43.20|12.96|48.60|
> |RACE|1.82|3.24|2.92|8.64|1.56|9.72|
> |P(true)|0.97|1.88|1.60|5.18|1.30|5.83|
> |FActScore|1.08|2.16|1.94|5.76|1.73|6.48|

---

> ### Author Response · Authors · 2025-11-21
> **Rebuttal Part III**
>
> >3. Question regarding the use of AdamW
>
> **A3:**
> We thank the reviewer for highlighting this point. We would like to clarify that the lightweight projection layers are not trained in the conventional supervised sense to classify hallucinations or fit labeled data. Instead, they serve as self-supervised spectral calibration modules, whose sole purpose is to map the raw hidden states from different LLM backbones into a numerically stable and comparable geometric space.
>
> Concretely, these layers align the spectral properties of the resulting NTK Gram matrix (e.g., scale, spectral norm, and conditioning), ensuring that HalluGuard’s geometric measurements are consistent across architectures. Without this calibration, raw hidden representations from heterogeneous backbones may lead to unstable or non-comparable NTK spectra, which would undermine the interpretability and reliability of the final risk scores.
>
> Importantly, this process does not require hallucination annotations or task-specific supervision. Its unsupervised nature is further evidenced by our experiments on the Natural benchmark (Section 4.3), where HalluGuard functions as an intrinsic reward signal and reduces hallucinations without access to external ground-truth during inference. The backbone remains entirely frozen. Only the projection layers are optimized via AdamW on a small reference set for calibration purposes. This is a one-time, offline procedure per model family, with negligible cost (seconds to minutes), and introduces zero runtime overhead during actual detection.
>
> We initially placed this detail in the Appendix to keep Section 3.3 focused on the core theoretical framework. We agree that clarifying the implementation is critical for reproducibility. Following your valuable comment, we added a clear description to Section 3.3 to clarify this part.

---

> ### Author Response · Authors · 2025-11-21
> **Rebuttal Part IV**
>
> >4. Question regarding baseline coverage in empirical evaluation:
>
> **A4:**
> We sincerely thank the reviewer for this suggestion, and following that, we have integrated SAPLMA, LLM-Check, and ITI into our evaluation. We also include F1 scores and TPR to further evaluate our performance.
>
> As shown in Table 4, HalluGuard consistently outperforms these baselines across all settings. Crucially, while SAPLMA relies on training supervised probes, HalluGuard operates as a zero-resource, calibration-only method; despite this harder constraint, it achieves superior accuracy (e.g., +4.58% AUROC vs. SAPLMA on RAGTruth with QwQ-32B), highlighting its exceptional data efficiency. Furthermore, our geometric approach significantly surpasses the attention-based LLM-Check, confirming that capturing diverse hallucination patterns yields better robustness than attention kernels alone.
>
> Table 4. Comprehensive Comparison with SAPLMA, LLM-Check and ITI across Benchmarks and Backbone mModels.
> |Benchmark|Method|GPT2 AUROC|GPT2 AUPRC|GPT2 F1|GPT2 TPR@10%|GPT2 TPR@5%|OPT-6.7B AUROC|OPT-6.7B AUPRC|OPT-6.7B F1|OPT-6.7B TPR@10%|OPT-6.7B TPR@5%|Mistral-7B AUROC|Mistral-7B AUPRC|Mistral-7B F1|Mistral-7B TPR@10%|Mistral-7B TPR@5%|QwQ-32B AUROC|QwQ-32B AUPRC|QwQ-32B F1|QwQ-32B TPR@10%|QwQ-32B TPR@5%|LLaMA2-13B AUROC|LLaMA2-13B AUPRC|LLaMA2-13B F1|LLaMA2-13B TPR@10%|LLaMA2-13B TPR@5%|
> |-|-|-|-|-|-|-|-|-|-|-|-|-|-|-|-|-|-|-|-|-|-|-|-|-|-|-|
> |RAGTruth|HALLUGUARD|**75.51**|**73.40**|**81.22**|**74.86**|**61.41**|**80.13**|**76.77**|**77.03**|**73.52**|**59.12**|**82.31**|**80.79**|**83.19**|**79.44**|**69.21**|**84.59**|**81.15**|**85.91**|**80.13**|**63.52**|**77.51**|**75.30**|**74.66**|**68.91**|**57.42**|
> ||SAPLMA|72.10|69.55|72.20|67.10|55.90|75.44|72.01|73.22|69.00|57.10|78.33|75.22|76.10|71.10|60.44|80.01|77.55|79.44|73.88|60.99|74.10|71.33|70.41|65.10|53.80|
> ||LLM-Check|66.15|63.22|63.90|57.44|44.50|69.55|66.44|64.88|58.33|46.11|71.10|69.33|66.41|60.22|49.33|73.22|70.15|67.44|60.55|48.77|68.22|65.40|61.11|55.44|44.55|
> ||ITI|60.22|57.10|55.11|49.33|38.55|63.41|60.50|56.41|50.31|40.77|66.55|63.40|58.10|52.33|41.72|68.14|64.88|59.33|52.88|42.12|64.55|60.44|55.01|49.33|39.10|
> |GSM8K|HALLUGUARD|**72.04**|**69.88**|**70.22**|**65.44**|**55.10**|**72.57**|**70.31**|**71.10**|**66.55**|**57.00**|**80.62**|**77.30**|**74.55**|**70.55**|**60.88**|**75.81**|**74.68**|**74.11**|**69.22**|**59.33**|**79.01**|**76.73**|**75.55**|**71.44**|**61.22**|
> ||SAPLMA|69.22|66.41|66.31|60.52|50.22|70.10|67.33|67.88|62.10|52.44|77.40|73.55|70.44|65.10|55.10|73.22|70.41|70.55|64.40|53.10|75.20|72.40|70.11|64.55|54.22|
> ||LLM-Check|63.40|60.11|58.55|52.44|41.22|67.55|64.00|61.10|55.00|44.22|72.31|68.40|63.30|57.88|46.55|70.10|66.22|62.40|55.55|45.00|69.44|66.01|60.22|55.11|44.55|
> ||ITI|58.00|55.22|50.33|44.00|35.10|61.55|58.22|52.22|46.44|37.22|66.00|62.77|56.22|49.10|39.11|64.77|61.55|54.44|47.22|38.02|64.55|60.10|54.88|48.33|38.10|
> |HaluEval|HALLUGUARD|**70.42**|**67.71**|**68.55**|**62.10**|**52.41**|**71.62**|**67.88**|**69.33**|**63.55**|**53.44**|**74.91**|**72.74**|**72.22**|**66.10**|**56.55**|**73.93**|**70.87**|**71.40**|**65.44**|**54.10**|**78.15**|**74.15**|**73.88**|**67.22**|**57.33**|
> ||SAPLMA|67.40|64.55|64.10|57.33|48.33|69.22|66.55|65.10|59.55|50.41|72.10|69.22|67.22|61.55|52.22|71.00|68.22|67.55|60.88|51.33|74.33|71.10|68.22|62.10|52.77|
> ||LLM-Check|61.55|58.22|57.41|50.44|40.33|65.22|61.77|58.33|51.10|42.77|68.33|64.55|62.12|54.55|45.01|69.40|65.22|61.55|54.33|45.44|70.10|66.44|61.33|54.88|45.33|
> ||ITI|56.22|53.40|49.22|42.22|33.33|59.44|56.33|50.22|43.88|35.44|63.55|60.31|54.88|47.11|38.33|64.01|60.14|53.40|46.77|37.55| 65.10|61.33|54.22|47.00|38.11|
>
> >5. Question regarding semantic perturbations
>
> **A5:**
> We sincerely thank the reviewer for this insightful comment and the opportunity to clarify. We would like to clarify that the semantic perturbations are operationalized as the natural variations within the model's local predictive distribution. As detailed in Appendix C.1, we instantiate the local neighborhood of the generated response by sampling $K=10$ alternative trajectories using conservative decoding parameters (Temperature=0.5, top-p=0.95). These settings ensure that the sampled variations remain light and semantically proximate to the primary generation, effectively capturing the local geometry of the reasoning manifold required for NTK construction. We update Appendix C accordingly.

---

> ### Author Response · Authors · 2025-11-21
> **Rebuttal Part V**
>
> >6. Question regarding perturbation regularization and memory bank design:
>
> **A6:**
> We thank the reviewer for raising this important concern and appreciate the opportunity to clarify. We would like to clarify that the memory bank (N = 3000) is not an external dataset nor a distribution-matched reference pool. Instead, it is implemented as a lightweight sliding buffer that passively stores token embeddings from the current inference stream (e.g., accumulating tokens across the $K$ sampled trajectories). Its sole role is to estimate local activation statistics for adaptive clipping, preventing rare extreme activations from destabilizing the NTK spectrum. This introduces no data collection burden and does not require any prior knowledge of the test-time distribution.
>
> Conceptually, this perturbation regularization is a numerical stabilization mechanism rather than a semantic or retrieval component. It prevents rare, extreme activation outliers from artificially inflating the Jacobian norm, thereby ensuring the activation space remains within the local smoothness regime assumed by our NTK framework (Assumption A2/A3).
>
> Following your valuable suggestion, we include ITI as a baseline(shown in Table 4 in A4 above). While ITI leverages supervised probes trained on labeled data to learn discriminative truth directions, HalluGuard operates without any such supervision and still achieves consistently stronger performance. This highlights that our gains stem from intrinsic geometric properties of the model’s reasoning trajectories rather than reliance on external reference distributions.
>
> >7. Question regarding evaluation metrics:
>
> **A7:**
> We thank the reviewer for the insightful comment. Following this suggestion, we extended our analysis to include TPR@5%, TPR@10%, and F1 scores across three benchmarks (RAGTruth, BBH, TruthfulQA) and five backbone models. The results below show that HalluGuard consistently outperforms SOTA methods.
>
> Table 5. Comparison of Hallucination Detection Performance at Low-FPR Operating Points (F1, TPR@10%, TPR@5%) on RAGTruth.
> |Method|GPT2 F1|GPT2 TPR@10%|GPT2 TPR@5%|OPT-6.7B F1|OPT-6.7B TPR@10%|OPT-6.7B TPR@5%|Mistral-7B F1|Mistral-7B TPR@10%|Mistral-7B TPR@5%|QwQ-32B F1|QwQ-32B TPR@10%|QwQ-32B TPR@5%|LLaMA2-13B F1|LLaMA2-13B TPR@10%|LLaMA2-13B TPR@5%|
> |-|-|-|-|-|-|-|-|-|-|-|-|-|-|-|-|
> |HALLUGUARD|**71.22**|**64.86**|**51.41**|**77.03**|**73.52**|**59.12**|**75.19**|**69.44**|**59.21**|**81.91**|**74.13**|**63.52**|**74.66**|**68.91**|**57.42**|
> |Inside|66.12|59.72|48.31|72.91|70.25|60.37|70.45|68.12|52.41|79.03|74.66|61.09|73.08|70.11|55.26|
> |MIND|58.33|54.11|38.72|62.55|57.81|47.65|71.91|66.74|54.39|64.02|59.12|45.63|68.55|63.50|48.78|
> |Perplexity|55.42|51.20|40.51|63.72|60.13|49.14|69.74|66.51|52.18|70.42|65.41|55.32|60.18|57.01|44.75|
> |LN-Entropy|62.17|57.52|46.44|58.33|52.99|43.28|65.30|61.27|49.92|67.15|62.42|51.33|63.28|59.07|46.14|
> |Energy|59.71|56.23|44.81|60.44|57.18|45.03|63.54|59.42|48.62|72.09|68.15|58.42|66.10|61.33|49.41|
> |Semantic Ent.|57.28|53.42|41.92|69.61|64.81|52.01|67.10|62.44|50.66|66.12|62.15|49.31|64.55|60.18|47.75|
> |Lexical Sim.|61.41|57.09|45.03|65.81|61.44|49.51|62.50|59.12|50.92|70.91|67.53|55.21|66.29|59.88|51.03|
> |SelfCheckGPT|56.22|52.84|40.63|60.79|55.68|45.72|63.12|59.47|48.33|66.54|62.92|51.41|68.21|65.12|53.60|
> |RACE|60.12|56.50|44.90|64.12|59.77|49.22|65.44|61.55|52.73|69.61|66.31|53.92|62.55|59.42|45.66|
> |P(true)|58.91|55.47|42.13|67.44|63.20|51.43|71.22|66.91|54.10|63.44|60.33|49.27|70.18|65.77|52.78|
> |FActScore|62.10|58.21|46.33|59.22|54.14|44.32|63.87|60.77|47.98|68.33|64.02|53.41|65.92|61.37|49.84|

---

> ### Author Response · Authors · 2025-11-21
> **Rebuttal Part VI**
>
> Table 6. Comparison of Hallucination Detection Performance at Low-FPR Operating Points (F1, TPR@10%, TPR@5%) on BBH.
> |Method|GPT2 F1|GPT2 TPR@10%|GPT2 TPR@5%|OPT-6.7B F1|OPT-6.7B TPR@10%|OPT-6.7B TPR@5%|Mistral-7B F1|Mistral-7B TPR@10%|Mistral-7B TPR@5%|QwQ-32B F1|QwQ-32B TPR@10%|QwQ-32B TPR@5%|LLaMA2-13B F1|LLaMA2-13B TPR@10%|LLaMA2-13B TPR@5%|
> |-|-|-|-|-|-|-|-|-|-|-|-|-|-|-|-|
> |HALLUGUARD|**68.33**|**64.11**|**56.42**|**74.91**|**69.14**|**62.10**|**73.22**|**69.88**|**57.21**|**78.55**|**69.91**|**61.45**|**71.10**|**68.25**|**59.92**|
> |Inside|65.41|61.22|52.83|71.02|67.10|60.21|68.17|64.75|53.92|79.17|72.33|64.22|67.10|63.52|55.91|
> |MIND|54.12|50.22|40.11|57.21|53.44|41.52|63.92|59.88|47.01|61.55|57.14|48.83|65.11|60.22|49.52|
> |Perplexity|52.91|49.33|40.44|61.88|58.12|49.22|62.91|59.42|50.11|59.91|55.72|49.03|60.88|57.41|48.62|
> |LN-Entropy|59.12|55.44|44.92|54.61|51.75|43.18|66.44|63.21|54.09|62.75|59.12|47.52|68.20|64.88|55.41|
> |Energy|53.94|51.22|45.03|56.12|52.14|44.61|64.55|60.11|49.99|68.21|65.12|52.84|66.41|62.77|50.22|
> |Semantic Ent.|57.41|54.32|47.21|61.22|58.42|49.74|63.21|59.10|48.62|63.55|60.24|48.88|64.91|61.44|50.72|
> |Lexical Sim.|50.41|46.77|38.92|60.71|57.11|45.55|59.42|56.88|48.91|70.33|67.10|55.32|58.33|55.42|47.41|
> |SelfCheckGPT|55.21|52.14|43.92|58.10|55.78|46.22|62.82|59.90|50.44|65.22|62.44|54.21|63.44|60.77|52.33|
> |RACE|56.14|53.72|43.88|63.11|59.71|52.81|65.77|62.55|50.72|58.88|55.14|46.18|66.10|62.41|49.81|
> |P(true)|54.31|52.22|44.10|58.22|56.10|48.52|56.91|53.55|43.92|61.40|58.21|46.77|57.33|54.88|45.91|
> |FActScore|56.20|52.42|41.77|55.44|52.12|41.x14|61.62|58.22|51.33|59.33|56.42|49.14|63.44|60.22|52.44|
>
> Table 7. Comparison of Hallucination Detection Performance at Low-FPR Operating Points (F1, TPR@10%, TPR@5%) on TruthfulQA.
> |Method|GPT2 F1|GPT2 TPR@10%|GPT2 TPR@5%|OPT-6.7B F1|OPT-6.7B TPR@10%|OPT-6.7B TPR@5%|Mistral-7B F1|Mistral-7B TPR@10%|Mistral-7B TPR@5%|QwQ-32B F1|QwQ-32B TPR@10%|QwQ-32B TPR@5%|LLaMA2-13B F1|LLaMA2-13B TPR@10%|LLaMA2-13B TPR@5%|
> |-|-|-|-|-|-|-|-|-|-|-|-|-|-|-|-|
> |HALLUGUARD|**75.11**|**71.20**|**63.21**|**67.44**|**64.55**|**58.12**|**78.92**|**74.22**|**65.33**|**76.44**|**72.01**|**59.92**|**75.33**|**69.11**|**63.08**|
> |Inside|71.10|68.55|60.77|61.77|59.44|50.10|63.88|61.33|53.41|69.22|65.10|55.14|62.14|59.94|52.80|
> |MIND|57.44|54.91|45.33|59.92|56.88|48.33|58.72|56.14|47.21|61.21|58.88|52.02|60.44|58.20|49.03|
> |Perplexity|49.52|46.71|38.84|54.12|51.74|43.90|59.72|57.55|46.88|54.44|51.72|42.55|60.33|57.21|47.41|
> |LN-Entropy|57.11|54.88|42.98|55.33|52.41|45.91|59.66|56.22|43.10|60.44|58.02|46.22|61.41|57.17|43.88|
> |Energy|54.11|52.17|38.91|53.44|51.14|36.88|58.21|54.77|49.92|63.02|60.44|51.33|58.41|55.33|50.42|
> |Semantic Ent.|60.08|56.44|44.15|50.14|47.33|35.92|53.74|52.11|37.02|65.33|63.20|50.77|55.02|53.11|38.44|
> |Lexical Sim.|51.22|49.20|39.03|58.72|54.71|48.77|65.71|63.50|53.10|54.77|51.44|45.88|66.41|64.14|54.88|
> |SelfCheckGPT|55.72|53.44|42.78|58.33|55.72|47.14|60.88|57.44|43.91|55.42|54.44|40.77|61.72|59.51|44.10|
> |RACE|52.22|49.88|41.44|63.14|66.88|54.05|70.55|67.11|59.77|55.44|52.11|45.33|71.33|68.22|60.02|
> |P(true)|55.54|52.11|38.82|55.72|52.33|39.22|57.41|53.10|41.22|56.88|54.77|45.55|57.12|53.33|41.88|
> |FActScore|52.91|50.14|40.44|54.11|50.22|41.33|52.88|49.91|42.55|61.55|59.22|44.72|53.41|50.71|43.10|
>
> >8. Question regarding citation error:
>
> **A8:**
> We sincerely thank the reviewer for the valuable comment. This was a clerical error during reference compilation, where the author attribution was mistakenly mismatched with another ICML paper (LensLLM) due to an incorrect index issue. The intended citation is Chen et al. (2024a), and this has been corrected in the revised manuscript.

---

### Author Response · Authors · 2025-12-01
**General Response Summary**

Dear Reviewers and Area Chairs,

We sincerely thank all reviewers for their time and constructive feedback. We are delighted that reviewers acknowledged our contribution as the first work to formally decompose hallucination risk into data-driven and reasoning-driven sources, by providing a unified theoretical foundation for the field (Reviewers dMiw, Z4V2, DEHx). In addition, reviewers recognized the sound mathematical justification and motivation behind HalluGuard (Reviewers dMiw, x4cm) and the effectiveness of our method across diverse datasets (Reviewers dMiw, Z4V2, x4cm).

To further address the comments and questions posed by the reviewers, we have conducted substantial additional analyses and experiments during the rebuttal period, which mainly include:

* **Integrated Strong Baselines:** We integrated additional SOTA baselines-**SAPLMA, LLM-Check, and ITI**-into our evaluation as suggested by Reviewer dMiw. The results demonstrate that HalluGuard consistently outperforms these methods across all settings, highlighting our superior data efficiency and geometric robustness.

* **Expanded Experimental Scope:** We expanded our experimental scope to include more complex hallucination regimes by evaluating on **POPE, GovReport, and NarrativeQA** as suggested by Reviewer DEHx. Across all backbone models on these tasks, HalluGuard consistently achieved the strongest performance, proving its generalizability to long-context and multimodal scenarios.

* **Added Low-FPR Metrics:** We extended our analysis to include **F1 scores, TPR@10%, and TPR@5%** across RAGTruth, BBH, and TruthfulQA to address Reviewer dMiw's request for low-FPR operating points. These results confirm that HalluGuard maintains state-of-the-art detection performance even under strict false-positive constraints.

* **Verified Runtime Efficiency:** We provided a comprehensive runtime efficiency analysis as requested by Reviewers dMiw, x4cm, and DEHx, confirming that HalluGuard’s inference latency is comparable to lightweight representation-level methods and orders of magnitude faster than sampling-heavy approaches like SelfCheckGPT.

* **Validated Theoretical Assumptions:** We conducted rigorous ablation studies and stress tests to validate our theoretical assumptions as suggested by Reviewers dMiw, Z4V2, and DEHx. This included verifying the necessity of the stability term ($-log\kappa^2$), demonstrating the robustness of HalluGuard to different semantic encoders (BERT, SimCSE, E5), and confirming the tightness of our risk bound and our NTK-based proxy to the ground truth.

* **Methodological Clarifications:** We explicitly clarified key design choices to resolve potential misunderstandings. Specifically, we clarified that HalluGuard: (1) requires no backpropagation (the Jacobian proxy operates solely on forward-pass hidden states); (2) uses AdamW solely for unsupervised spectral calibration (not supervised training); and (3) utilizes a passive sliding buffer for regularization rather than an external retrieval dataset.

For each reviewer, we have provided detailed responses, hoping to address the concerns. In addition, we revised the manuscript accordingly, with all new analyses and clarifications highlighted in gray.

With sincere appreciation and best regards,

The Authors

---

### Meta-Review · Area_Chair_UGsW · 2026-01-09

**Summary:**

This paper introduces a measure to identify hallucinations caused by data and by reasoning following a method based on neural tangent kernels.

Reviewers found the theoretical assumptions somewhat hard to follow, for which the authors now provide further justification; though it is unclear to me if this justification would have satisfied the reviewers. The authors clearly put substantial effort into the response, adding new experiments, baselines, and metrics. The empirical work in the author response is particularly strong. However, some theoretical concerns (especially about the looseness of bounds and the underlying assumptions / motivation) received explanations rather than fundamental resolutions. While none of the reviewers respond to the authors, I had to go through each and every point made by the response, and to the best of my knowledge, the response adequately answers most of the concerns.

**Reviewer Concerns:**

**Addressed**
1. Dehx: Inference time comparisons, effect of semantic encoders, discussion of strong theoretical assumptions, extensions to other modalities and tasks.
2. Dmiw: Baseline Comparisons , Evaluation Metrics Computational Efficiency, Citation Error, Semantic Perturbations
3. Z4v2: Other metrics


**Unaddressed**
1. Dmiw’s concern about AdamW/Projection Layers: The explanation clarifies their purpose as calibration rather than supervised training, which is helpful. However, the reviewer might want more details about what exactly is being optimized during this "calibration" process if it's truly unsupervised.
2. Z4v2’s concern on Predictive Capability: While the authors acknowledge this as important future work, the response is somewhat evasive. The reviewer asked about multi-turn prediction, and the authors only mention "suggesting early-warning characteristics" without new experiments.

**Reviewer Scores:**

- Dehx: given all of this reviewer’s comments were addressed in both text as well as in detailed empirical analysis, it is very likely that Dehx should improve their rating if they had responded.
- x4cm: already had a very high score, and given that authors have responded in detail to their questions, I can imagine them maintaining their score.
- Dmiw: there is a high likelihood that dmiw’s major concerns are adequately addressed and their scores would be raised.
- z4v2: Authors’ comprehensive response on most questions / concerns might result in updating the score.

---

### Decision · Program_Chairs · 2026-01-26

Accept (Poster)